# PRIMAL-DUAL POLICY OPTIMIZATION FOR LINEAR CMDPS WITH ADVERSARIAL LOSSES

**Kihyun Yu, Seoungbin Bae**
Department of Industrial and Systems Engineering
KAIST
{khyu99,sbbae31}@kaist.ac.kr

**Dabeen Lee** *
Department of Mathematical Sciences
Seoul National University
dabeenl@snu.ac.kr

## ABSTRACT

Existing work on linear constrained Markov decision processes (CMDPs) has primarily focused on stochastic settings, where the losses and costs are either fixed or drawn from fixed distributions. However, such formulations are inherently vulnerable to adversarially changing environments. To overcome this limitation, we propose a primal-dual policy optimization algorithm for online finite-horizon adversarial linear CMDPs, where the losses are adversarially chosen under full-information feedback and the costs are stochastic under bandit feedback. Our algorithm is the *first* to achieve sublinear regret and constraint violation bounds in this setting, both bounded by $\widetilde{\mathcal{O}}(K^{3/4})$, where $K$ denotes the number of episodes. The algorithm introduces and runs with a new class of policies, which we call weighted LogSumExp softmax policies, designed to adapt to adversarially chosen loss functions. Our main result stems from the following key contributions: (i) a new covering number argument for the weighted LogSumExp softmax policies, and (ii) two novel algorithmic components—periodic policy mixing and a regularized dual update—which allow us to effectively control both the covering number and the dual variable. We also report numerical results that validate our theoretical findings on the performance of the algorithm.

## 1 INTRODUCTION

Safe reinforcement learning (RL) studies sequential decision-making under safety constraints through interaction with an unknown environment. Many real-world applications have been explored under the safe RL framework, including autonomous driving (Isele et al., 2018), robotics (Achiam et al., 2017), and healthcare (Coronato et al., 2020). A common modeling framework for safe RL is the online constrained Markov decision process (CMDP) formulation, where the agent seeks a policy that minimizes (or maximizes) cumulative expected loss (or reward), while ensuring that the expected cumulative cost does not exceed a given budget (Altman, 2021).

To better capture realistic scenarios, it is often necessary to model adversarial environments, where components may vary arbitrarily over time. For instance, in autonomous driving, the loss may reflect safety risks such as sudden braking, but it can also increase drastically due to unexpected traffic or hazardous weather. In service robotics, the loss may correspond to task failures or user dissatisfaction, which fluctuate with human preferences or rapidly changing tasks. In such applications, assuming a fixed loss signal is overly restrictive. Therefore, to model these scenarios, it is essential to consider CMDPs under adversarial settings.

Online adversarial CMDPs assume that the loss or cost functions can change arbitrarily across episodes, rather than being drawn from fixed stochastic distributions. Recently, adversarial CMDPs have been investigated under the tabular setting (Qiu et al., 2020; Stradi et al., 2024; 2025a;c; Zhu et al., 2025). Although these works achieved sublinear regret and constraint violation bounds, they focused only on environments where the state space is finite and relatively small. As a result, their algorithmic guarantees may not extend to settings with large state spaces. Such algorithms are often unsatisfactory in real-world applications, where the number of states is typically extremely large.

---

*Additional Affiliations: Research Institute of Mathematics, Seoul National University; Interdisciplinary Program in Artificial Intelligence, Seoul National University; Korea Institute for Advanced Study

To capture settings with a large state space, safe RL with linear function approximation has been studied. Ding et al. (2021) proposed a primal-dual policy optimization algorithm for linear mixture CMDPs, where the dynamics are expressed as a mixture of a finite set of basis kernels. For linear CMDPs, assuming linear structure in the loss and cost functions as well as in the dynamics, Ghosh et al. (2022; 2024) designed a primal-dual-type optimistic value iteration algorithm with a softmax policy. For the same setting, Kitamura et al. (2025) developed an algorithm achieving zero constraint violation under the assumption of a known safe policy. Although these algorithms can handle large state spaces, they considered only stochastic settings, where the loss and cost functions are fixed or drawn from underlying distributions. That said, they fail to capture the aforementioned applications, where taking into account adversarial environments is essential when modeling safe RL algorithms.

To overcome these limitations, this paper proposes an algorithm for adversarial linear CMDPs. To handle adversarial losses with constraints, as online primal-dual mirror descent type algorithms become a standard choice, it is natural to consider primal-dual policy optimization for our setting (Chen et al., 2021; Ding et al., 2021). However, when applying primal-dual policy optimization to adversarial linear CMDPs, additional challenges arise—most notably in bounding the covering number of the value function class (Jin et al., 2020).

To elaborate on this challenge, primal-dual policy optimization induces a more intricate policy class, necessitating a new covering number argument. In particular, slightly perturbing the primal variable before optimizing it—namely, the policy in our case—is commonly used in various settings (Wei et al., 2020; Qiu et al., 2020; Ding et al., 2021; Stradi et al., 2025c). The purpose of this step is to derive a compact dual variable bound, which is essential for regret and violation analyses. However, this step breaks the recursive structure of policy optimization, so the resulting policy cannot be represented as a typical softmax policy. As a consequence, the covering number argument becomes non-trivial. In other words, while policy mixing is simple and common, it poses a critical issue for covering number arguments in linear CMDPs. Despite these challenges, we aim to answer the following question:

*Can we design a primal-dual policy optimization algorithm for adversarial linear CMDPs that ensures sublinear regret and violation bounds?*

**Main Contributions** We answer the question affirmatively with Algorithm 1, designed for finite-horizon adversarial linear CMDPs, where the losses are adversarially chosen under full-information feedback, and the costs are stochastic under bandit feedback. We summarize our main contributions.

- We present a primal-dual policy optimization algorithm (Algorithm 1) for adversarial linear CMDPs that achieves regret and constraint violation upper bounds of $\widetilde{\mathcal{O}}(K^{3/4})$, where $K$ is the number of episodes. Our algorithm is the *first* algorithm that achieves sublinear regret and violation in the adversarial linear CMDP setting. Moreover, the algorithm develops a new class of policies, which we refer to as weighted LogSumExp softmax policies, designed to adapt to adversarially chosen loss functions.

- We establish a covering number argument for the novel class of weighted LogSumExp softmax policies, induced by primal-dual policy optimization algorithms. The main technical difficulty arises from the fact that the weight parameters across policies may differ, preventing direct application of standard properties of the LogSumExp function. Nevertheless, our analysis shows that the covering number under this policy class is bounded by $\widetilde{\mathcal{O}}(n^2 d^2)$, where $n$ is the maximum number of mixing steps, and $d$ is the feature dimension of the linear CMDP.

- Another challenge in designing a sublinear algorithm for adversarial linear CMDPs lies in the need to simultaneously control both the covering number and the dual variable. To address this, our algorithm incorporates two novel components: (i) periodic policy mixing and (ii) regularized dual updates. Since the covering number grows with the number of mixing steps, the purpose of periodic policy mixing is to regulate the frequency of mixing steps by applying it once in every specified mixing period, rather than in every episode. To incorporate periodic policy mixing, our dual update has to introduce an additional regularization term in order to obtain a compact bound on the dual variable. Together, these algorithmic components allow us to effectively control both the covering number and the dual variable, establishing a sublinear algorithm.

A more detailed review of related work is deferred to the appendix.

## 2   PROBLEM SETTING

**Finite-Horizon Adversarial CMDP**   A finite-horizon adversarial CMDP is defined by the tuple
$\mathcal{M} = (H, \mathcal{S}, \mathcal{A}, \{\mathbb{P}_h\}_{h=1}^H, \{f^k\}_{k=1}^K, \{g^k\}_{k=1}^K, s_1, b)$, where $H$ is the finite horizon, $\mathcal{S}$ is the finite
state space[1], and $\mathcal{A}$ is the finite action space. $\{\mathbb{P}_h\}_{h=1}^H$ is a collection of transition kernels for each
step $h \in [H]$, where $\mathbb{P}_h(s' \mid s, a)$ denotes the probability of transitioning from state $s$ to state $s'$
when action $a$ is taken at step $h$. $\{f^k\}_{k=1}^K$ and $\{g^k\}_{k=1}^K$ are the sequences of loss and cost functions
over episodes $k \in [K]$, where $f^k = \{f_h^k\}_{h=1}^H$ and $g^k = \{g_h^k\}_{h=1}^H$ satisfy $f_h^k, g_h^k : \mathcal{S} \times \mathcal{A} \to [0, 1]$.
$s_1 \in \mathcal{S}$ is the fixed initial state, and $b \in [0, H]$ is the cost budget.

We consider a setting where the loss functions are adversarial, while the cost functions are stochastic.
Specifically, at the beginning of each episode $k \in [K]$, an adversary chooses the loss function $f^k$,
which can be selected arbitrarily (i.e., not drawn from a distribution). In contrast, the cost function
$g^k$ is sampled i.i.d. from a fixed distribution $G$, satisfying $\mathbb{E}[g_h^k(s, a) \mid s, a] = g_h(s, a)$.

The interaction between the agent and the environment proceeds as follows. At the beginning of
each episode $k \in [K]$, $f^k$ is adversarially chosen, which is not revealed to the agent. Next, the agent
selects a collection of policies $\{\pi_h\}_{h=1}^H$, where $\pi_h(a \mid s)$ denotes the probability of taking action $a$
given state $s$ at step $h$. Once the episode begins, at each step $h \in [H]$, the agent samples an action
$a_h \sim \pi_h(\cdot \mid s_h)$. Upon taking $a_h$, the agent observes $f_h^k$ and $g_h^k(s_h, a_h)$, which are full-information
feedback for the adversarial loss and bandit feedback for the stochastic cost, respectively. Lastly, the
next state is sampled as $s_{h+1} \sim \mathbb{P}_h(\cdot \mid s_h, a_h)$.

We define the value function and the $Q$-function. Let $V_{\ell,h}^\pi(s)$ denote the value function at state $s$
and step $h$ with respect to a function $\ell = \{\ell_h\}_{h=1}^H$ and policy $\pi$, which is written as $V_{\ell,h}^\pi(s) =
\mathbb{E}_{\mathbb{P},\pi}[\sum_{j=h}^H \ell_j(s_j, a_j) \mid s_h = s]$. Similarly, the $Q$-function $Q_{\ell,h}^\pi(s, a)$ is defined as $Q_{\ell,h}^\pi(s, a) =
\mathbb{E}_{\mathbb{P},\pi}[\sum_{j=h}^H \ell_j(s_j, a_j) \mid s_h = s, a_h = a]$.

We define the performance metrics—regret and constraint violation—as follows. Given a se-
quence of policies $\pi^1, \ldots, \pi^K$ generated by the agent, the regret and constraint violation for
$K$ episodes are defined as $\text{Regret}(K) = \sum_{k=1}^K (V_{f^k,1}^{\pi^k}(s_1) - V_{f^k,1}^{\pi^*}(s_1))$ and $\text{Violation}(K) =
\left[\sum_{k=1}^K (V_{g,1}^{\pi^k}(s_1) - b)\right]_+$, where $[\cdot]_+$ denotes $\max\{\cdot, 0\}$. Here, $\pi^*$ is an optimal policy, de-
fined as a solution to the following optimization problem over the set of all policies $\Pi$: $\pi^* \in
\arg\min_{\pi \in \Pi} \sum_{k=1}^K V_{f^k,1}^\pi(s_1)$ s.t. $V_{g,1}^\pi(s_1) \le b$.

**Linear CMDP**   We consider a class of CMDP instances with an underlying linear structure, re-
ferred to as linear CMDP (Ghosh et al., 2022). Let $\phi : \mathcal{S} \times \mathcal{A} \to \mathbb{R}^d$ denote the known feature
mapping. With the feature $\phi$, the transition kernel is defined as $\mathbb{P}_h(s' \mid s, a) = \phi(s, a)^\top \psi_h(s')$
where $\psi_h(s') \in \mathbb{R}^d$ is an unknown signed measure. Similarly, the loss and cost functions are as-
sumed to be linear in $\phi$ and are defined as $f_h^k(s, a) = \phi(s, a)^\top \theta_{f,h}^k$ and $g_h(s, a) = \phi(s, a)^\top \theta_{g,h}$,
where $\theta_{f,h}^k, \theta_{g,h} \in \mathbb{R}^d$ are unknown parameters. Moreover, we further assume that the parame-
ters for linear CMDPs are all bounded as follows. For all $(s, a, h, k) \in \mathcal{S} \times \mathcal{A} \times [H] \times [K]$, we
have $\|\phi(s, a)\|_2 \le 1$ and $\max\{\|\sum_{s' \in \mathcal{S}} |\psi_h|(s')\|_2, \|\theta_{f,h}^k\|_2, \|\theta_{g,h}\|_2\} \le \sqrt{d}$, where $|\psi_h|(s')$ denotes
$(|(\psi_h(s'))_1|, |(\psi_h(s'))_2|, \ldots, |(\psi_h(s'))_d|)^\top \in \mathbb{R}^d$.

Next, we introduce the Slater condition, which is a mild assumption commonly made in the CMDP
literature (Efroni et al., 2020; Liu et al., 2021; Ding et al., 2021; Ghosh et al., 2022).

**Assumption 1** (Slater Condition). *We assume that there exists a Slater policy $\bar{\pi} \in \Pi$ such that*
$V_{g,1}^{\bar{\pi}}(s_1) + \gamma \le b$ *for some Slater constant $\gamma > 0$.*

---

[1]For simplicity, we assume that the state space is finite. However, the state space may be arbitrarily large,
as discussed in Cassel et al. (2024), since the computational complexity of our algorithm—as well as the regret
and constraint violation—does not scale with $|\mathcal{S}|$, which will be presented in the following section.

## 3 CHALLENGES AND NOVEL TECHNIQUES

**Novelty 1: Analysis for Weighted LogSumExp Softmax**  We construct a covering number argument with a new policy structure—weighted LogSumExp softmax policies—which arises from combining policy optimization with policy mixing. This policy is given by the weighted sum of exponentials of sums of $Q$-function estimates[2]: given a step size $\alpha$, weight parameters $\zeta_i$, and $Q$-function estimates $\widehat{Q}^j$,

$$\widehat{\pi}^k \propto \sum_{i=1}^{k} \zeta_i \exp\left(-\alpha \sum_{j=k-i}^{k-1} \widehat{Q}^j\right). \tag{1}$$

Let us explain how (1) arises in primal–dual policy optimization. Perturbing the primal variable before optimizing it is a simple yet effective technique for controlling the scale of the dual variable (Wei et al., 2020; Qiu et al., 2020). In the context of policy optimization, this technique translates into the following update: given a uniform policy $\pi_{\text{unif}}$ over $\mathcal{A}$ and a mixing parameter $\theta$,

$$\underbrace{\widehat{\pi}^{k-1} \leftarrow (1-\theta)\widehat{\pi}^{k-1} + \theta\pi_{\text{unif}}}_{\text{Policy Mixing}} \quad \text{and then} \quad \underbrace{\widehat{\pi}^k \propto \widehat{\pi}^{k-1}\exp(-\alpha\widehat{Q}^{k-1})}_{\text{Policy Optimization}}.$$

Here, the additive relation (Policy Mixing) breaks the recursion in the proportional relation (Policy Optimization). As a consequence, the resulting policy takes the form of the weighted LogSumExp softmax. In particular, (1) may assign different weights $\zeta_i$ to partial sums $\sum_{j=k-i}^{k-1}\widehat{Q}^j$ for $i \in [k]$. This yields a more expressive policy compared to the case without policy mixing, where the update simplifies to $\widehat{\pi}^k \propto \exp(-\alpha\sum_{j=0}^{k-1}\widehat{Q}^j)$.

Our first contribution is to provide a new covering number argument for the value function class, where the policy is given by (1). For comparison, Jin et al. (2020) studied the greedy policy, where the policy is defined as $\arg\max_{a\in\mathcal{A}}\widehat{Q}^k$. Then the covering number can be analyzed since the $\max$ operation is a contraction mapping. Moreover, the simple softmax policy has been studied in several works, e.g., Ghosh et al. (2022). In that case, leveraging well-established Lipschitz properties of the softmax function is sufficient to analyze the covering number.

We note that constructing a covering number argument for (1) is non-trivial. The main difficulty is that the weight parameters $\{\zeta_i\}_{i=1}^k$ depend not only on the mixing parameter $\theta$ but also on $Q$-function estimates. This means that different policies can have different weight parameters $\{\zeta_i\}_{i=1}^k$, and thus, well-known properties of LogSumExp cannot be applied. Despite these challenges, our analysis shows that the logarithm of the covering number under (1), denoted by $\log\mathcal{N}_\epsilon$, grows quadratically with $n$, where $n$ is the maximum number of mixing steps during the learning process:

$$\log\mathcal{N}_\epsilon = \widetilde{\mathcal{O}}(n^2 d^2). \tag{2}$$

**Novelty 2: Periodic Policy Mixing**  However, deriving an upper bound on the covering number alone is not sufficient to guarantee sublinear regret and violation. In particular, if mixing is applied in every episode, then $\log\mathcal{N}_\epsilon$ grows to the order of $\widetilde{\mathcal{O}}(K^2 d^2)$, which is too large to yield a sublinear guarantee. On the other hand, if mixing is performed insufficiently, then the dual variable cannot be bounded, which is critical for violation analysis. These observations highlight an inherent trade-off between the covering number and the size of dual variables, both of which heavily depend on the frequency of mixing. This necessitates a new algorithmic component to balance the two.

The aforementioned trade-off motivates our second contribution—periodic policy mixing—which applies the policy mixing every $K^B$ episodes[3], where $B$ is a period parameter between 0 and 1. The purpose of the periodic policy mixing is to balance the covering number and the size of dual variables. The covering number can be easily observed from (2), since the number of mixing steps is at most $K^{1-B}$ (i.e., the number of episodes divided by the mixing period). However, it remains unclear whether periodic policy mixing is effective in controlling the dual variable. To address this, in the next paragraph, we show that the dual variable can indeed be bounded when periodic policy mixing is combined with a new dual update rule.

---

[2]We call this formulation the weighted LogSumExp softmax, as it is equivalent to $\widehat{\pi}^k \propto \exp(\log(\sum_i \zeta_i \exp(-\alpha\sum_j \widehat{Q}^j)))$—a softmax of weighted LogSumExp with respect to $-\alpha\sum_j \widehat{Q}^j$.

[3]For simplicity, we assume that $K^B, K^{1-B}$ are integers to avoid additional notation such as $\lfloor K^B \rfloor$.

**Novelty 3: Regularized Dual Update** Our third contribution is another algorithmic component—a new dual update rule with additional regularization. When combined with periodic policy mixing, the dual variable $Y_k$ is bounded by $\widetilde{\mathcal{O}}(\eta K^B)$, where $\eta$ is the step size and $K^B$ is the mixing period. For clarity, this bound omits the dependence on $\gamma, H$ to highlight the impact of the mixing period.

To elaborate on our dual update method, it takes the following form: given regularization parameters $c_1, c_2 > 0$ and the cost value function estimate $\widehat{V}_{g,1}^k$,

$$Y_{k+1} \leftarrow [Y_k + \eta(\widehat{V}_{g,1}^k(s_1) - b) + \underbrace{(-c_1 Y_k - c_2)}_{\text{Regularization}}]_+.$$

For interpretation, $Y_k + \eta(\widehat{V}_{g,1}^k(s_1) - b)$ corresponds to the standard online gradient ascent step for dual updates in primal-dual algorithms, while the regularization term $(-c_1 Y_k - c_2)$ pulls the dual variable towards 0, keeping it compact. The regularization parameters $c_1, c_2$ will be specified in the following paragraph, along with the intuition for our design.

The intuition behind our regularization is that it serves as a crucial ingredient for a drift-based analysis, a well-known method for bounding dual variables (see, e.g., Yu et al. (2017); Wei et al. (2020) in constrained online convex optimization). To enable drift analysis, these works typically incorporate an inner product term in the dual update, determined by the decision variables and the gradient, i.e., $\langle x_{t+1} - x_t, \nabla_t \rangle$. In primal-dual policy optimization, we realize that this translates to a term involving the transition kernel, i.e., $\mathbb{E}_{\mathbb{P}}[\langle \widehat{\pi}^{k+1} - \widehat{\pi}^k, \widehat{Q}^k \rangle]$. However, since the transition kernel is unknown, this term cannot be directly incorporated into our algorithm. Instead, we take a lower bound on this term, which becomes our regularizing component with the choice of $c_1 = 4\alpha\eta H^3$ and $c_2 = 4\alpha\eta H^3 + 4\theta\eta H^2$. In this way, our dual update can be viewed as a key adaptation that enables drift analysis in primal-dual policy optimization for adversarial linear CMDPs.

## 4 ALGORITHM

We present Primal-Dual Policy Optimization for Linear CMDPs with Adversarial Losses(Algorithm 1). The algorithm consists of four main components: epoch initialization (lines 2-7), policy execution and estimation (lines 8-19), policy optimization with periodic policy mixing (lines 20-26), and updating the dual variable (line 27).

In lines 2-7, the algorithm initializes a new epoch when the determinant of the design matrix $\Lambda_{h'}^k$ increases by a multiplicative factor compared to that of $\Lambda_{h'}^{k_e}$ for some $h'$. Once the initialization procedure begins, the algorithm sets the policy to the uniform policy and initializes the dual variable to 0. Furthermore, it defines a contracted feature $\bar{\phi}_h^{k_e}$ by shrinking the original feature $\phi$. The multiplicative contraction factor is determined by $\sigma(-\beta_w \|\phi(\cdot, \cdot)\|_{(\Lambda_h^{k_e})^{-1}} + \log K)$, where $\sigma$ denotes the sigmoid function, and $\|\phi(\cdot, \cdot)\|_{(\Lambda_h^{k_e})^{-1}}$ quantifies the current uncertainty of least-squares estimators. This contracted feature is then used in the estimation of $Q$-functions.

**Remark 1.** The feature contraction—originally proposed by Cassel & Rosenberg (2024) for adversarial linear (unconstrained) MDPs—is necessary for the following reason. Specifically, it provides a simpler expression for the policy, which is useful in covering number arguments, via a low-dimensional representation of the sum of $Q$-function estimates. For this, they omitted a clipping operation in the definition of $Q$-function estimates, so the sum collapses into a simple inner product with an optimistic bonus. Instead of clipping, they properly contracted the feature to prevent $Q$-function estimates from expanding uncontrollably. This technique can be replaced with other approaches with the same purpose, such as Sherman et al. (2024), but it may lead to higher dependence on $d, H$ in regret and violation bounds.

In lines 8-10, the algorithm takes action $a_h^k \sim \widehat{\pi}_h^k(\cdot|s_h^k)$ for each step $h \in [H]$ and observes $\theta_{f,h}^k, g_h^k(s_h^k, a_h^k)$, and $s_{h+1}^k \sim \mathbb{P}_h(\cdot|s_h^k, a_h^k)$. In lines 11-14, the design matrix $\Lambda_h^{k+1}$ is updated, and the parameters for the loss and cost functions are estimated, denoted by $\widehat{\theta}_{f,h}^k$ and $\widehat{\theta}_{g,h}^k$, respectively. Based on these, in lines 15-19, for each $\ell = f, g$, it computes the $Q$-function estimates $\widehat{Q}_{\ell,h}^k(s, a)$ using the contracted feature and the value function estimates $\widehat{V}_{\ell,h}^k(s)$, which are defined by the inner product of $\widehat{\pi}_h^k(\cdot|s)$ and $\widehat{Q}_{\ell,h}^k(s, \cdot)$ for each $s \in \mathcal{S}$.

---

**Algorithm 1** Primal-Dual Policy Optimization for Linear CMDPs with Adversarial Losses

---

**Input:** $\delta \in (0,1), \beta_b = 2\sqrt{2d\log(6KH/\delta)} + 50(K^{1/4}+1)dH\sqrt{\log(5H^2K^2|\mathcal{A}|/\delta)}, \beta_w = 4\beta_b \log K, \alpha = H^{-1}K^{-3/4}, \eta = H^{-2}K^{-3/4}, \theta = K^{-1}$

**Initialization:** $Y_1 \leftarrow 0, \ e \leftarrow 0, \ \Lambda_h^1 \leftarrow I, \ \widehat{V}_{\ell,H+1}^k(s) \leftarrow 0 \quad \forall(h,k,s,\ell) \in [H] \times [K] \times \mathcal{S} \times \{f,g\}$

---

1: **for** $k = 1, \ldots, K$ **do**
2:     **if** $k = 1$ or $\exists h' \in [H]$ such that $\det(\Lambda_{h'}^k) \geq 2\det(\Lambda_{h'}^{k_e})$ **then**
3:         $e \leftarrow e + 1$ and $k_e \leftarrow k$
4:         $\widehat{\pi}_h^{k_e}(\cdot \mid s) \leftarrow \pi_{\text{unif}}(\cdot \mid s) \ \forall h \in [H]$
5:         $Y_{k_e} \leftarrow 0$
6:         $\bar{\phi}_h^{k_e}(\cdot,\cdot) = \phi(\cdot,\cdot) \cdot \sigma(-\beta_w\|\phi(\cdot,\cdot)\|_{(\Lambda_h^{k_e})^{-1}} + \log K) \ \forall h \in [H]$    $\triangleright$ Feature Contraction
7:     **end if**
8:     **for** $h = 1, \ldots, H$ **do**
9:         Take $a_h^k \sim \widehat{\pi}_h^k(\cdot \mid s_h^k)$, and observe $\theta_{f,h}^k, \ g_h^k(s_h^k,a_h^k), \ s_{h+1}^k \sim \mathbb{P}_h(\cdot \mid s_h^k,a_h^k)$
10:    **end for**
11:    **for** $h = H, \ldots, 1$ **do**
12:       $\Lambda_h^{k+1} \leftarrow I + \sum_{\tau \in [k]} \phi(s_h^\tau,a_h^\tau)\phi(s_h^\tau,a_h^\tau)^\top$
13:       $\widehat{\theta}_{f,h}^k \leftarrow \theta_{f,h}^k$
14:       $\widehat{\theta}_{g,h}^k \leftarrow (\Lambda_h^k)^{-1} \sum_{\tau \in [k-1]} \phi(s_h^\tau,a_h^\tau)g_h^\tau(s_h^\tau,a_h^\tau)$
15:       **for** $\ell \in \{f,g\}$ **do**
16:           $\widehat{\psi}_h^k \widehat{V}_{\ell,h+1}^k = (\Lambda_h^k)^{-1} \sum_{\tau \in [k-1]} \phi(s_h^\tau,a_h^\tau)\widehat{V}_{\ell,h+1}^k(s_{h+1}^\tau)$
17:           $\widehat{Q}_{\ell,h}^k(\cdot,\cdot) \leftarrow \bar{\phi}_h^{k_e}(\cdot,\cdot)^\top \left[\widehat{\theta}_{\ell,h}^k + \widehat{\psi}_h^k \widehat{V}_{\ell,h+1}^k\right] - \beta_b\|\bar{\phi}_h^{k_e}(\cdot,\cdot)\|_{(\Lambda_h^{k_e})^{-1}}$
18:           $\widehat{V}_{\ell,h}^k(\cdot) \leftarrow \sum_{a \in \mathcal{A}} \widehat{\pi}_h^k(a \mid \cdot)\widehat{Q}_{\ell,h}^k(\cdot,a)$
19:       **end for**
20:       **if** $k - k_e \equiv 0 \mod K^{3/4}$ **then**                $\triangleright$ Periodic Policy Mixing
21:         $\widetilde{\pi}_h^k(\cdot \mid s) \leftarrow (1-\theta)\widehat{\pi}_h^k(\cdot \mid s) + \theta\pi_{\text{unif}}(\cdot \mid s)$
22:       **else**
23:         $\widetilde{\pi}_h^k(\cdot \mid s) \leftarrow \widehat{\pi}_h^k(\cdot \mid s)$
24:       **end if**
25:       $\widehat{\pi}_h^{k+1}(\cdot \mid s) \propto \widetilde{\pi}_h^k(\cdot \mid s) \exp\left(-\alpha(\widehat{Q}_{f,h}^k(s,\cdot) + Y_k\widehat{Q}_{g,h}^k(s,\cdot))\right)$    $\triangleright$ Policy Optimization
26:    **end for**
27:    $Y_{k+1} \leftarrow \left[(1-4\alpha\eta H^3)Y_k + \eta\left(\widehat{V}_{g,1}^k(s_1) - b - 4\alpha H^3 - 4\theta H^2\right)\right]_+$    $\triangleright$ Dual Update
28: **end for**

---

In lines 20-24, the algorithm applies the policy mixing every $K^{3/4}$ episodes. Here, the mixed policy is obtained by taking a convex combination of $\widehat{\pi}_h^k$ and $\pi_{\text{unif}}$ with coefficients $1-\theta$ and $\theta$, respectively. After this, the algorithm performs policy optimization—equivalently, an online mirror descent step with Kullback-Leibler (KL) divergence over the policy space.

In line 27, the algorithm updates the dual variable, $Y_k$. First, it scales down the dual variable by a factor of $1 - 4\alpha\eta H^3$, and then adds $\eta(\widehat{V}_{g,1}^k(s_1) - b - 4\alpha H^3 - 4\theta H^2)$. Finally, it takes $[\cdot]_+$ to ensure that the dual variable remains nonnegative.

The computational complexity of Algorithm 1 is $\mathcal{O}(d^3HK + d^2|\mathcal{A}|HK^2)$, which is independent of $|\mathcal{S}|$. Specifically, in lines 2-7, computing determinants simply takes $\mathcal{O}(d^3HK)$ and contracting features takes $\mathcal{O}(d^2K \cdot |\mathcal{A}| \cdot HK)$, since the inverse of the design matrices can be computed in $\mathcal{O}(d^2K)$ by applying the Sherman-Morrison formula. In lines 8-29, the dominant step is function estimation: lines 16-18 take $\mathcal{O}(d^2|\mathcal{A}|HK^2)$.

## 4.1 COMPARISON OF DUAL UPDATES

Since our algorithm is designed for adversarial linear CMDPs, it is worth comparing our dual update with that of algorithms for (i) stochastic linear CMDPs (Ghosh et al., 2022) and (ii) tabular CMDPs with adversarial losses (Qiu et al., 2020).

First, we compare with Ghosh et al. (2022), which focused on stochastic linear CMDPs. The key difference comes from the way the dual variable is regularized. Specifically, while both approaches adopt the standard online gradient ascent procedure, their update rule truncates the dual variable at $2H/\gamma$ to ensure that it never exceeds this threshold. In contrast, our update incorporates an extra regularization term to keep the dual variable compact. Although their update is simple and effective in the stochastic setting, it cannot be extended to the adversarial setting, since their analysis relies on the fact that the loss and cost functions are fixed over episodes. This justifies the need for our design of dual update steps in handling adversarial losses.

Second, we compare with Qiu et al. (2020), which proposed an occupancy measure-based algorithm for adversarial tabular CMDPs. The main difference in the dual updates arises from the choice of primal variable: policy-based mirror descent versus occupancy measure-based mirror descent. Before elaborating on this, we recall a dual update from the constrained online convex optimization literature, proposed by Wei et al. (2020) with minor modifications: given a convex cost function $\ell : \mathbb{R}^d \to \mathbb{R}$ and primal variables $x^k, x^{k+1} \in \mathbb{R}^d$,

$$Y_{k+1} \leftarrow \left[ Y_k + \eta \left( \ell(x^k) - b + \langle \nabla \ell(x^k), x^{k+1} - x^k \rangle \right) \right]_+ .$$

Based on this update, let us show how the dual update for occupancy measure-based algorithms can be derived. Since the occupancy measure serves as the primal variable, we take $x^k \leftarrow q^k$, where $q^k$ denotes an occupancy measure in episode $k$. Furthermore, in CMDPs, note that the expected cost is given by $\langle g, q^k \rangle$[4], where $g \in \mathbb{R}^{|\mathcal{S}| \times |\mathcal{A}| \times H}$ denotes a vector representation of the cost function. Then we can take $\ell(x^k) \leftarrow \langle g, q^k \rangle$ and $\nabla \ell(x^k) \leftarrow g$. This leads to $Y_{k+1} \leftarrow [Y_k + \eta(\langle g, q^{k+1} \rangle - b)]_+$, which is the key intuition behind the dual update in Qiu et al. (2020).

However, this argument does not apply to policy-based mirror descent. This is because even if we take $x^k \leftarrow \pi^k$, the expected cost is not linear in $\pi^k$, unlike in the occupancy measure case. That said, the dual updates for occupancy measure-based algorithms can be extended from the online convex optimization literature, whereas extending this to policy-based algorithms is non-trivial. This highlights the significance of our proposed design.

## 4.2 Main Result

Finally, we present upper bounds on regret and constraint violation under Algorithm 1.

**Theorem 1.** *Let $H^2 \leq K$ and Assumption 1 hold. Suppose that we run Algorithm 1. Given $\delta > 0$, with probability at least $1 - \delta$, then we have*

$$\text{Regret}(K) = \widetilde{\mathcal{O}} \left( \sqrt{d^3 H^4} K^{3/4} + dH^3 K^{3/4} + d^3 H^4 K^{1/2} + \frac{H^6}{\gamma^2} K^{1/4} + \frac{dH^6}{\gamma^2} \right),$$

$$\text{Violation}(K) = \widetilde{\mathcal{O}} \left( \frac{dH^5}{\gamma} K^{3/4} + \sqrt{d^3 H^4} K^{3/4} + d^3 H^4 K^{1/2} \right)$$

*where $\widetilde{\mathcal{O}}(\cdot)$ hides polynomial factors in $\log(dHK|\mathcal{A}|/(\delta\gamma))$.*

## 5 Analysis

In this section, we present the proof outline of Theorem 1, where the details of the proofs can be found in the appendix. As a first step, we introduce two key ingredients: (i) a high-probability good event and (ii) bounding the dual variable. We note that our covering number argument plays a central role in showing that the good event holds with high probability. Furthermore, to bound the dual variable, the key part is to consider periodic policy mixing and the regularized dual update.

**Good Event**   We first introduce a high-probability event, denoted by $E_g$, whose formal definition is provided in the appendix. Basically, the event captures estimation errors for the loss, the cost, and the transition kernel. In addition, it guarantees the boundedness of $Q$-function estimates, reflecting the usefulness of the feature contraction. The following lemma shows that $E_g$ holds with high probability.

---

[4]For simplicity, we assume that $g$ is deterministic and known, and $q^k$ is induced by the true transition kernel.

**Lemma 1.** *Let $\beta_w \leq K$. Then for any $\delta \in (0, 1)$, $\Pr[E_g] \geq 1 - \delta$.*

In the proof of Lemma 1, the main distinction from previous works arises in our covering number argument. Specifically, our case incorporates weighted LogSumExp softmax policies, induced by mixing the policy. Then we have to derive a Lipschitz property of the policies that have been mixed $n$ times, as the number of mixing steps is a key factor in determining the policy structure. Note that the Lipschitzness of policies is fundamentally required in most covering number arguments.

To attain this property, we prove the following recursion in $n$, presented here in simplified form:

$$\|\widehat{\pi}_1^n - \widehat{\pi}_2^n\|_1 \leq c\|\widehat{\pi}_1^{n-1} - \widehat{\pi}_2^{n-1}\|_1 + \|\mathcal{P}_1^n - \mathcal{P}_2^n\|_2$$

where $\widehat{\pi}_1^n, \widehat{\pi}_2^n$ denote the policies that have been mixed $n$ times, $\mathcal{P}_1^n, \mathcal{P}_2^n$ denote the subsets of the corresponding parameters, and $c$ is a constant. By applying this recursion repeatedly, we can bound the difference between policies using the sum of differences in their parameters, establishing the Lipschitzness. Based on this, we can show that the covering number is bounded as $\widetilde{\mathcal{O}}(n^2 d^2)$.

**Dual Variable Bound** Under the good event $E_g$, we can establish another ingredient of our analysis—a drift analysis for bounding the dual variable.

**Lemma 2.** *Assume that the good event $E_g$ holds. Let $H^2 \leq K$. Let $Y_k$ be the dual variable generated by Algorithm 1 for each $k \in [K]$. Then, we have $Y_k = \widetilde{\mathcal{O}}(H^2/\gamma)$.*

Let us briefly explain our proof strategy for Lemma 2. Although the regularization in our dual update enables drift analysis, we cannot directly apply the previous proofs proposed by Wei et al. (2020); Qiu et al. (2020). This is because their analyses rely on applying policy mixing in every episode, whereas our algorithm applies it only sparsely. To exploit this sparse structure, we instead consider a subsequence of dual variables corresponding to the mixing episodes, denoted by $\{Z_n\}_{n \geq 1}$ where $Z_n = Y_{k_e + nK^B}$ for each epoch $e$. We first bound $Z_n$ for all $n$, and consequently extend the result to derive a bound on $Y_k$ for all $k$.

Next, we introduce decompositions of both $\mathrm{Regret}(K)$ and $\mathrm{Violation}(K)$. Let $E$ be the set of all epochs, and let $K_e$ be the set of episodes in epoch $e \in E$. We have

$$
\begin{aligned}
\mathrm{Regret}(K) \leq &\underbrace{\sum_{k=1}^{K} \left(V_{f^k,1}^{\pi^k}(s_1) - \widehat{V}_{f,1}^k(s_1)\right)}_{(I)} + \underbrace{\sum_{k=1}^{K} Y_k \left(b - \widehat{V}_{g,1}^k(s_1)\right)}_{(II)} \\
&+ \underbrace{\sum_{k=1}^{K} \left(\widehat{V}_{f,1}^k(s_1) + Y_k \widehat{V}_{g,1}^k(s_1) - V_{f^k,1}^{\pi^*} - Y_k V_{g,1}^{\pi^*}(s_1)\right)}_{(III)},
\end{aligned}
\tag{3}
$$

$$
\mathrm{Violation}(K) \leq \underbrace{\sum_{k=1}^{K} \left(V_{g,1}^{\pi^k}(s_1) - \widehat{V}_{g,1}^k(s_1)\right)}_{(IV)} + \underbrace{\sum_{k=1}^{K} \left(\widehat{V}_{g,1}^k(s_1) - b\right)}_{(V)}.
$$

Terms (I), (IV) arise from the difference between the true value function and its optimistic estimates, which are closely related to the optimistic bonus $-\beta_b \|\bar{\phi}_h^{k_e}(\cdot, \cdot)\|_{(\Lambda_h^{k_e})^{-1}}$. Since our parameter for optimistic bonus is set as $\beta_b = \widetilde{\mathcal{O}}(K^{1/4} dH)$, where $K^{1/4}$ comes from the covering number argument, these terms are bounded by $\widetilde{\mathcal{O}}(K^{3/4})$, as stated in the following lemma.

**Lemma 3.** *Let $H^2 \leq K$. Suppose that $E_g$ holds. For all $\ell \in \{f, g\}$,*

$$\sum_{k=1}^{K}(V_{\ell,1}^{\pi^k}(s_1) - \widehat{V}_{\ell,1}^k(s_1)) = \widetilde{\mathcal{O}}\left(\sqrt{d^3 H^4} K^{3/4} + d^3 H^4 K^{1/2}\right).$$

Term (II) arises from the dual update in the sense that if the dual variable is not updated (i.e., $Y_k = 0$ for all $k$), then this term vanishes. It can be bounded using the following lemma.

**Lemma 4.** *Let $H^2 \le K$. Suppose that $E_g$ holds. Then we have*

$$\sum_{k=1}^{K} Y_k(b - \widehat{V}_{g,1}^k(s_1)) = \widetilde{\mathcal{O}}\left(K^{1/4} + \frac{dH^6}{\gamma^2}\right).$$

To bound term (III), we further decompose it into two parts—optimism terms and an online mirror descent term—and then bound each individually. This leads to the following lemma.

**Lemma 5.** *Let $H^2 \le K$. Suppose that $E_g$ holds. Then we have*

$$\sum_{k=1}^{K} \left(\widehat{V}_{f,1}^k(s_1) + Y_k\widehat{V}_{g,1}^k(s_1) - (V_{f^k,1}^{\pi^*} + Y_k V_{g,1}^{\pi^*}(s_1))\right) = \widetilde{\mathcal{O}}\left(dH^3 K^{3/4} + \frac{H^6}{\gamma^2}K^{1/4} + \frac{dH^5}{\gamma}\right).$$

Finally, term (V) can be bounded via the dual variable bound. This is because $\eta(\widehat{V}_{g,1}^k(s_1) - b)$ accumulates in the dual variable, as it is repeatedly added in the dual update. Based on this idea, we show the following lemma, which bounds term (V).

**Lemma 6.** *Let $H^2 \le K$. Suppose that $E_g$ holds. Then we have*

$$\sum_{k=1}^{K}(\widehat{V}_{g,1}^k(s_1) - b) = \widetilde{\mathcal{O}}\left(\frac{dH^5}{\gamma}K^{3/4} + \frac{H^4}{\gamma}K^{1/4}\right).$$

## 6 NUMERICAL EXPERIMENT

We evaluate Algorithm 1 on a job-scheduling CMDP (Ghosh et al., 2022), modified to incorporate adversarial losses. We conduct 10 simulations with different random seeds, each running for $K = 10^5$ episodes. Additional details about the experimental setup are deferred to the appendix.

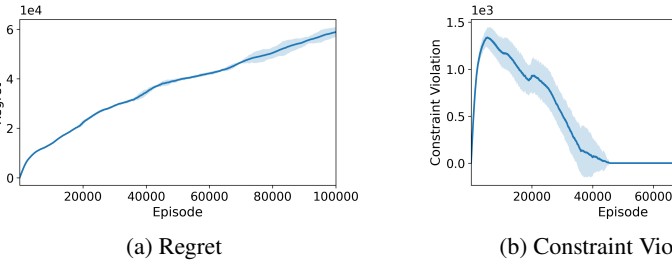

(a) Regret          (b) Constraint Violation

Figure 1: Plots of regret and constraint violation for $K = 100,000$ episodes. Each plot represents the average over 10 trials with random seeds, and shaded regions indicate $95\%$ confidence intervals.

Figure 1 summarizes the results. As shown in Figure 1a, the regret grows sublinearly in $K$, and Figure 1b shows that while the constraint violation grows rapidly in the early phase, it eventually converges to 0. These results support our theoretical claims.

## 7 CONCLUSION

This paper studies adversarial linear CMDPs, where the losses are adversarially chosen under full-information feedback and the costs are stochastic under bandit feedback. We propose a primal-dual policy optimization algorithm—the first provably efficient algorithm for safe RL with linear function approximation in adversarial settings. We establish a new covering number argument for weighted LogSumExp softmax policies, along with novel algorithmic components that jointly control the covering number and the dual variable. Building on these, we show that the proposed algorithm achieves $\widetilde{\mathcal{O}}(K^{3/4})$ regret and violation bounds. Moreover, our numerical experiments support this. As directions for future work, it remains open to investigate the following challenges: (i) whether $\widetilde{\mathcal{O}}(\sqrt{K})$ regret and violation bounds can be achieved in our setting, and (ii) whether a sample-efficient algorithm can be designed for linear CMDPs with adversarial losses under bandit feedback.

ACKNOWLEDGMENTS

We would like to thank the review team for their careful review and valuable feedback. This work was supported by the National Research Foundation of Korea (NRF) grant (No. RS-2024-00350703) and the Institute of Information & communications Technology Planning & evaluation (IITP) grants (No. IITP-2026-RS-2024-00437268) and (No. RS-2021-II211343, Artificial Intelligence Graduate School Program (Seoul National University)) funded by the Korea government (MSIT).

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

# A  NOTATION

Table 1: Summary of notation

| NOTATION | DEFINITION |
|---|---|
| $H, \mathcal{S}, \mathcal{A}, K$ | The finite-horizon, the state and action spaces, and the number of episodes |
| $d$ | The feature dimension |
| $\mathbb{P}$ | The transition kernel |
| $f, g$ | The loss and the cost functions |
| $b$ | The budget |
| $E$ | The set of epochs |
| $k_e$ | The first episode of epoch $e \in E$ |
| $\phi$ | The feature |
| $\bar{\phi}_h^{k_e}$ | The contracted feature at step $h$ in epoch $e$ |
| $\widehat{\theta}_{\ell,h}^k$ | The estimate of $\theta_{\ell,h}^k$ |
| $\Lambda_h^k$ | The design matrix at step $h$ in episode $k$ |
| $\psi_h V$ | $\sum_{s' \in \mathcal{S}} \psi_h(s') V(s')$ for $V : \mathcal{S} \to \mathbb{R}$ |
| $\widehat{\psi}_h^k V$ | The estimate of $\psi_h V$ at step $h$ in episode $k$ |
| $[n]$ | The set $\{1, 2, \dots, n\}$ for a positive integer $n$ |
| $\mathbb{Z}_+$ | The set $\{0, 1, 2, \dots\}$ |
| $\mathbb{R}_+$ | The set $\{z \in \mathbb{R} : z \geq 0\}$ |
| $\|\cdot\|_2$ | The $\ell_2$-norm for vectors and the operator norm for matrices |
| $\|\cdot\|_\infty$ | The $\ell_\infty$-norm |
| $\|\cdot\|_F$ | The Frobenius norm |
| $\|\cdot\|_\Lambda$ | $\|x\|_\Lambda = \sqrt{x^\top \Lambda x}$ for $\Lambda \succ 0$ |
| $\Delta(\mathcal{A})$ | The probability simplex over $\mathcal{A}$ |
| $I$ | The $d \times d$ identity matrix |
| $D(\cdot \| \cdot)$ | Kullback-Leibler divergence |
| $\sigma$ | The sigmoid function |
| $\gamma, \bar{\pi}$ | The Slater constant and the Slater policy |
| $Q_{\ell,h}^\pi(s, a)$ | The $Q$-function |
| $\widehat{Q}_{\ell,h}^k(s, a)$ | The $Q$-function estimate |
| $V_{\ell,h}^\pi(s)$ | The value function |
| $\widehat{V}_{\ell,h}^k(s)$ | the value function estimate |
| $\bar{V}_{\ell,h}^\pi(s; \rho)$ | The value function with respect to a $\rho$-contracted MDP |
| $\widehat{\pi}_h^k$ | The policy at step $h$ in episode $k$ |
| $\pi_{\mathrm{unif}}$ | $\pi_{\mathrm{unif}}(a \mid s) = 1/|\mathcal{A}|$ for all $(s, a) \in \mathcal{S} \times \mathcal{A}$ |
| $Y_k$ | The dual variable in episode $k$ |
| $K^B$ | The mixing period, $K^{3/4}$ |
| $\theta$ | The mixing parameter, $\theta = K^{-1}$ |
| $\alpha$ | The step size for the mirror descent, $\alpha = H^{-1} K^{-3/4}$ |
| $\eta$ | The step size for the dual update, $\eta = H^{-2} K^{-3/4}$ |
| $\beta_r$ | $2\sqrt{2d \log(6KH/\delta)}$ |
| $\beta_p$ | $50(K^{1/4} + 1)dH \sqrt{\log(5H^2 K^2 |\mathcal{A}|/\delta)}$ |
| $\beta_b$ | $\beta_r + \beta_p$ |
| $\beta_w$ | $4\beta_b \log K$ |
| $\beta_{Q,h}$ | $2(H - h + 1)$ |
| $\mathcal{N}_\epsilon(\widehat{\mathcal{V}})$ | The $\epsilon$-covering number of $\widehat{\mathcal{V}}$ with respect to the $\ell_\infty$-norm |

# B  LIMITATIONS OF PRIOR WORK AND NAÏVE EXTENSION

In this section, we clarify why previous works—and their naïve extensions—fail in our setting, where the losses are adversarially chosen in each episode. In particular, we address the following

two questions: (i) why the algorithm of Ghosh et al. (2022) fails in the adversarial setting; and (ii) why simply adapting a mirror-descent update is insufficient for adversarial linear CMDPs.

**Limitation of Prior Work** While Ghosh et al. (2022) proposed a primal-dual algorithm for linear CMDPs, their analysis is limited to the setting with stochastic losses and constraints. The fundamental reason is that their algorithm is value-based, whose policy is determined solely by the current $Q$-function estimates in each episode; that is $\widehat{\pi}^k(\cdot \mid s) \propto \exp(-\alpha \widehat{Q}^{k-1})$. Such a policy cannot adapt to time-varying environments, and in particular, cannot handle adversarially chosen losses.

Another limitation is that the dual update of Ghosh et al. (2022) fails to handle adversarial environments. Their key technique is a dual clipping technique—cutting off the dual variable when it exceeds $2H/\gamma$—which enables them to leverage the strong duality of CMDPs. Moreover, to leverage strong duality in their analysis, they reformulate the weighted sum of regret and violation into a simple Lagrangian form (e.g., Appendix D of Ghosh et al. (2022)), i.e., there exists a policy $\pi'$ such that

$$\frac{1}{K}\Big( \sum_{k=1}^{K}(V_{f,1}^{\widehat{\pi}^k} - V_{f,1}^{\pi^\star}) + Y\sum_{k=1}^{K}(V_{g,1}^{\widehat{\pi}^k} - b) \Big) = (V_{f,1}^{\pi'} - V_{f,1}^{\pi^\star}) + Y(V_{g,1}^{\pi'} - b).$$

In our case, when losses are adversarially chosen in each episode, reformulating the sum into a simple Lagrangian form is not allowed, i.e.,

$$\frac{1}{K}\Big( \sum_{k=1}^{K}(V_{f^k,1}^{\widehat{\pi}^k} - V_{f^k,1}^{\pi^\star}) + Y\sum_{k=1}^{K}(V_{g,1}^{\widehat{\pi}^k} - b) \Big) \neq (V_{f',1}^{\pi'} - V_{f',1}^{\pi^\star}) + Y(V_{g,1}^{\pi'} - b).$$

In turn, leveraging strong duality in our setting is non-trivial.

**Naïve Extension** To overcome the limitations of value-based algorithms in adversarial settings, a natural approach is to adopt a mirror-descent update, whose regularizer is given by a KL divergence. In our case, this corresponds to a policy optimization, i.e.,

$$\widehat{\pi}^k = \arg\min_{\pi \in \Pi} \langle \pi, \widehat{Q}^{k-1}\rangle + \frac{1}{\alpha}D(\pi||\widehat{\pi}^{k-1}) \quad \Rightarrow \quad \widehat{\pi}^k \propto \widehat{\pi}^{k-1}\exp\Big(-\alpha\widehat{Q}^{k-1}\Big).$$

Due to its recursive formulation, we can see that the resulting policy depends on the sum of all previous $Q$-function estimates, namely $\widehat{\pi}^k \propto \exp(-\alpha\sum_{j=1}^{k-1}\widehat{Q}^j)$. This is the key difference compared with value-based algorithms.

More technically, let us attempt to adapt the algorithm of Wei et al. (2020) to the linear CMDP setting. Their method is a mirror-descent type algorithm—in our case, policy optimization—designed for constrained online convex optimization with adversarial losses and stochastic constraints. Beyond policy optimization, there are two additional distinctions compared with Ghosh et al. (2022):

- (Drift Analysis) As previously mentioned, since strong duality is difficult to use in the adversarial setting, the dual clipping technique may not works in the adversarial setting. To address this, Wei et al. (2020) came up with a dual update that admits a *Lyapunov drift analysis*. In particular, Lyapunov drift analysis is a standard tool for bounding the dual variable. For this, we first derive an upper bound on the Lyapunov drift term defined as $\Delta(k) := (Y_{k+1}^2 - Y_k^2)/2$, and then utilize it to bound $Y_k$. The key point is to make $\Delta(k)$ small enough so that $Y_k$ stays stable, as $\Delta(k)$ captures the difference between successive dual variables.

- (Policy Mixing) Another key technique of Wei et al. (2020) is *policy mixing*—perturbing the policy before applying the policy optimization update. The motivation behind this technique is to make $\Delta(k)$ small. In particular, a typical bound on $\Delta(k)$ in policy optimization involves KL divergence terms as follows:

$$\Delta(k) \leq -c_1 Y_k + c_2 + D(\pi||\widetilde{\pi}_h^k) - D(\pi||\widehat{\pi}_h^{k+1})$$

  The key issue is that $D(\pi||\widetilde{\pi}_h^k)$ can become arbitrarily large when $\widetilde{\pi}_h^k(a) \approx 0$ for some $a \in \mathcal{A}$, because the KL divergence is unbounded near the simplex boundary. In contrast, when a mixing step is applied, we ensure that $\widetilde{\pi}_h^k(a) \geq \theta/|\mathcal{A}|$ for all $a$, where $\theta$ denotes the level of mixing. In this case, we can easily show that $D(\pi||\widetilde{\pi}_h^k) \leq \log(|\mathcal{A}|/\theta)$ (Lemma 25). Hence, the mixing step guarantees that the KL term remains bounded, which in turn prevents the dual variable from blowing up.

**Insufficiency of Naïve Extension**    While policy mixing is essential to keep the dual variable stable, it becomes problematic in the linear CMDP setting, where the main difficulty arises in the covering number argument. In particular, as described in Section 3, we have to derive an upper bound on the covering number of the weighted LogSumExp softmax policy. Establishing such a bound is one of our key challenges and is highly non-trivial. Thus, although adapting Wei et al. (2020) to the framework of Ghosh et al. (2022) is a natural step toward handling adversarial losses, obtaining sublinear regret and constraint violation bounds becomes unclear.

## C    ADDITIONAL DISCUSSIONS

**Intuition on Covering Number**    We provide an intuition on why the weighted LogSumExp softmax policy yields $\widetilde{\mathcal{O}}(n^2 d^2)$, where $n$ denotes the number of mixing steps, while the covering number under greedy or softmax policies is just $\widetilde{\mathcal{O}}(d^2)$. Before explaining this, we note that the covering number of a function class depends on (i) how many parameters are required and (ii) how close these parameters must be for the functions to be sufficiently similar. Based on this high-level idea, the number of parameters needed to determine a weighted LogSumExp softmax policy is $\mathcal{O}(nd^2)$, as each $\sum_j \widehat{Q}^j$ requires $\mathcal{O}(d^2)$ parameters once the feature contraction is applied. Furthermore, we observe that the impact of parameters decreases exponentially as mixing continues, meaning that the parameters must be chosen increasingly close (see the proof of Lemma 11). This leads to an additional multiplicative factor $n$, resulting in $\widetilde{\mathcal{O}}(n^2 d^2)$.

**Choice of Mixing Period**    To justify the choice of $K^{3/4}$, we first clarify how $\mathrm{Regret}(K)$ and $\mathrm{Violation}(K)$ depend on the mixing period $K^B$. Note that the covering number is closely related to terms (I) and (IV) of the decompositions in (3), and the dual variable directly affects term (V). This yields the following simplified regret and constraint violation bounds.

$$\mathrm{Regret}(K) = \widetilde{\mathcal{O}}(\sqrt{K \log \mathcal{N}_\epsilon}), \quad \mathrm{Violation}(K) = \widetilde{\mathcal{O}}(\sqrt{K \log \mathcal{N}_\epsilon} + Y_k/\eta).$$

We emphasize that $\mathrm{Violation}(K)$ can be bounded by $\widetilde{\mathcal{O}}(\sqrt{K^{3-2B}} + K^B)$. This is because the log covering number is bounded by $\widetilde{\mathcal{O}}(K^{2-2B} d^2)$, and the dual variable is bounded by $\widetilde{\mathcal{O}}(\eta K^B)$. Thus, to minimize the dependency on $K$, we set $B = 3/4$.

**Discussion on Lower Bound**    We note that the regret lower bound of $\Omega(\sqrt{H^3 d^2 K})$ also applies to our setting, which is for stochastic linear unconstrained MDPs (Zhou et al., 2021; He et al., 2022). This is because by taking the loss to be fixed across episodes and using a trivial constraint (i.e., taking $b = H$), our problem reduces to a stochastic linear unconstrained MDP. Therefore, we conjecture that there remains room for improving our regret bound by a factor of $\widetilde{\mathcal{O}}(K^{1/4})$.

Additionally, we outline a promising direction toward achieving $\widetilde{\mathcal{O}}(\sqrt{K})$ regret and violation in our setting. The main challenge is analyzing the constraint violation without relying on mixing steps. In our current analysis, the mixing step is inevitable to mitigate KL divergence terms in the drift upper bound (Lemma 17); these KL divergence terms arise from mirror-descent type updates to control adversarial losses. However, mixing becomes problematic for linear CMDPs because it enlarges the covering number. If one can design an approach that controls the dual variable without mixing, then achieving optimal bounds may become possible.

## D    RELATED WORK

**Online Tabular CMDP**    Starting from the seminal work of Efroni et al. (2020), minimizing regret and constraint violation in online tabular CMDPs has been studied under various settings. Several works (Liu et al., 2021; Bura et al., 2022; Yu et al., 2025) considered the case of zero constraint violation under the assumption of a known safe policy. Under the same assumption, Müller et al. (2023) studied hard constraint violation—the sum of only positive constraint violations. Without this assumption, the hard constraint violation was studied by Müller et al. (2024); Stradi et al. (2025b). Moreover, Wei et al. (2022b) proposed a model-free algorithm for finite-horizon CMDPs, and Wei et al. (2022a); Chen et al. (2022) proposed algorithms for infinite-horizon average-reward CMDPs.

However, these works assume stationary environments. To relax this assumption, Qiu et al. (2020) studied adversarial losses under full-information feedback. For both adversarial losses and costs, Stradi et al. (2024; 2025c) considered full-information feedback and bandit feedback, respectively. More recently, several papers proposed algorithms for adversarial CMDPs with hard constraint violation guarantees. Stradi et al. (2025a) proposed an algorithm for adversarial losses and stochastic costs under bandit feedback, and Zhu et al. (2025) studied stochastic losses and adversarial costs under full-information feedback.

**Online Linear CMDP**  For finite-horizon linear CMDPs, Ghosh et al. (2022; 2024) studied cumulative and hard constraint violations, respectively. Similarly, Ghosh et al. (2023) developed several algorithms for the infinite-horizon average-reward setting. Kitamura et al. (2025) studied achieving zero constraint violation under the assumption of a known safe policy, and Liu et al. (2025) studied sample complexity under the assumption of a generative model. Wei et al. (2023) studied non-stationary CMDPs, where components of the environment may change subject to bounded total variation. However, this setting differs from adversarial settings, in which the variation of functions is not assumed to be bounded by some factor. Amani et al. (2021); Wei et al. (2024); Roknilamouki et al. (2025) investigated hard instantaneous constraints, where unsafe actions must not be taken in each step. We also note additional works that are not linear CMDPs but incorporate linear function approximation. There are several works for linear mixture CMDPs with various settings (Ding et al., 2021; Ding & Lavaei, 2023; Shi et al., 2023). More generally, the $q_\pi$-realizable setting was studied by Tian et al. (2024), which only assumes that value functions can be represented as an inner product of a given feature. However, adversarial environments have not been considered in these settings.

**Online Adversarial Linear MDP**  Online adversarial linear MDPs have been studied under full-information feedback (Zhong & Zhang, 2023; Sherman et al., 2024; Cassel & Rosenberg, 2024) and bandit feedback (Neu & Olkhovskaya, 2021; Luo et al., 2021; Dai et al., 2023; Sherman et al., 2023; Kong et al., 2024; Liu et al., 2024). Specifically, in the full-information feedback setting, Zhong & Zhang (2023) proposed a multi-batched policy optimization algorithm, achieving a $\widetilde{\mathcal{O}}(K^{3/4})$ regret bound. Sherman et al. (2024) achieved a $\widetilde{\mathcal{O}}(\sqrt{K})$ regret bound, adopting a warm-up phase to obtain a simple expression for policies. In addition, Cassel & Rosenberg (2024) proposed a warm-up free policy optimization algorithm with an improved regret bound. In the bandit feedback setting, Luo et al. (2021) introduced the notion of dilated bonus, and Liu et al. (2024) proposed two algorithms: one achieved a $\widetilde{\mathcal{O}}(\sqrt{K})$ regret bound but was computationally inefficient, and the other achieved $\widetilde{\mathcal{O}}(K^{3/4})$ and was computationally efficient.

## E  Contracted MDP

In this section, we explain the notion of a contracted MDP (Cassel & Rosenberg, 2024), which is essential for deriving our main results. A contracted MDP is defined by the tuple $\bar{\mathcal{M}} = (H, \mathcal{S}, \mathcal{A}, \{\bar{\mathbb{P}}_h\}_{h=1}^H, \{\bar{\ell}_h\}_{h=1}^H, s_1, \rho)$. Here, $\rho : \mathcal{S} \times \mathcal{A} \times [H] \to [0,1]$ specifies the level of contraction. In particular, the loss function and transition kernel are defined as

$$\bar{\ell}_h(s,a) = \rho(s,a,h)\ell_h(s,a),$$
$$\bar{\mathbb{P}}_h(s' \mid s,a) = \rho(s,a,h)\mathbb{P}_h(s' \mid s,a).$$

Since $\rho(s,a,h) \in [0,1]$, it follows that $\bar{\ell}_h(s,a) \in [0,1]$, meaning that the contraction preserves the boundedness of the original loss function. On the other hand, $\sum_{s' \in \mathcal{S}} \bar{\mathbb{P}}_h(s' \mid s,a) \le 1$, which implies that $\bar{\mathbb{P}}$ defines a sub-probability measure. Although this does not satisfy the definition of a probability measure, it is sufficient for our purposes, as the contracted transition kernel is only used in the analysis. Furthermore, it is often called a sub-MDP because its transition kernel is a sub-probability measure.

Accordingly, $\bar{V}_{\ell,h}^\pi(s; \rho)$ denotes the $\rho$-contracted value function with respect to a policy $\pi$ and a contracted MDP $\bar{\mathcal{M}}$. Given $\bar{V}_{\ell,H+1}^\pi(s; \rho) = 0$ for all $s \in \mathcal{S}$, $\bar{V}_{\ell,h}^\pi(s; \rho)$ is defined recursively as

$$\bar{V}_{\ell,h}^\pi(s; \rho) = \mathbb{E}_{\bar{\mathbb{P}},\pi}\left[\sum_{j=h}^H \bar{\ell}_j(s_j, a_j)|s_h = s\right] = \mathbb{E}_{a\sim\pi(\cdot|s)}\left[\bar{\ell}_h(s,a) + \sum_{s'\in\mathcal{S}}\bar{\mathbb{P}}_h(s' \mid s,a)\bar{V}_{\ell,h+1}^\pi(s'; \rho)\right].$$
$$(4)$$

We next introduce a lemma that highlights a key property of contracted MDPs, which states a $\rho$-contracted value function is less than or equal to its original value function.

**Lemma 7** (Lemma 2 of Cassel & Rosenberg (2024)). *For any* $\rho : \mathcal{S} \times \mathcal{A} \times [H] \to [0, 1]$, $\pi \in \Pi$, $h \in [H]$, $s \in \mathcal{S}$, *and* $\ell : \mathcal{S} \times \mathcal{A} \times [H] \to [0, 1]$, *we have* $\bar{V}_{\ell,h}^{\pi}(s; \rho) \le V_{\ell,h}^{\pi}(s)$.

Since $\bar{\mathbb{P}}$ is a sub-probability, we note that $\mathbb{E}_{\bar{\mathbb{P}},\pi}$ applied to a constant $c \ge 0$ could be less than $c$ itself. Although this is trivial, we state it formally below for completeness, as this relation is used frequently in our analysis.

$$\mathbb{E}_{\bar{\mathbb{P}},\pi}[c \mid s_1 = s] \le \mathbb{E}_{\mathbb{P},\pi}[c \mid s_1 = s] = c \quad \forall s \in \mathcal{S} \tag{5}$$

*Proof of* (5). First, we prove $\mathbb{E}_{\bar{\mathbb{P}},\pi}[\sum_{h \in [H]} \ell_h(s, a) | s_1 = s] \le \mathbb{E}_{\mathbb{P},\pi}[\sum_{h \in [H]} \ell_h(s, a) | s_1 = s]$ for any $s \in \mathcal{S}$ and $\{\ell_h\}_{h=1}^H$ where $\ell_h : \mathcal{S} \times \mathcal{A} \to \mathbb{R}_+$, using induction on $h$. For the base case, $h = H$, $\mathbb{E}_{\bar{\mathbb{P}},\pi}[\ell_H(s_H, a_H)|s_H] = \mathbb{E}_{\mathbb{P},\pi}[\ell_H(s_H, a_H)|s_H] = \mathbb{E}_{a_H \sim \pi(\cdot|s_H)}[\ell_H(s_H, a_H)]$. Assuming the statement is true for $h + 1$, we have

$$\mathbb{E}_{\bar{\mathbb{P}},\pi}\left[\sum_{j=h}^H \ell_j(s_j, a_j)|s_h = s\right] = \mathbb{E}_{a \sim \pi(\cdot|s)}\left[\ell_h(s_h, a_h) + \sum_{s' \in \mathcal{S}} \bar{\mathbb{P}}_h(s' \mid s_h, a_h)\mathbb{E}_{\bar{\mathbb{P}},\pi}\left[\sum_{j=h+1}^H \ell_j(s_j, a_j)|s'\right]\right]$$

$$\le \mathbb{E}_{a \sim \pi(\cdot|s)}\left[\ell_h(s_h, a_h) + \sum_{s' \in \mathcal{S}} \mathbb{P}_h(s' \mid s_h, a_h)\mathbb{E}_{\mathbb{P},\pi}\left[\sum_{j=h+1}^H \ell_j(s_j, a_j)|s'\right]\right]$$

$$= \mathbb{E}_{\mathbb{P},\pi}\left[\sum_{j=h}^H \ell_j(s_j, a_j)|s_h = s\right].$$

This completes the induction. Furthermore, by taking $\ell_h(s, a) = c/H$ for all $(s, a, h) \in \mathcal{S} \times \mathcal{A} \times [H]$, we have $\sum_{h \in [H]} \ell_h(s, a) = c$, and thus $\mathbb{E}_{\bar{\mathbb{P}},\pi}[c|s] \le \mathbb{E}_{\mathbb{P},\pi}[c|s]$ for any $s \in \mathcal{S}$. Since $\mathbb{P}$ is not contracted, we know that $\mathbb{E}_{\mathbb{P},\pi}[c|s] = c$. This completes the proof. $\square$

The following lemma is an extension of a well-known value difference lemma to incorporate contracted MDPs.

**Lemma 8** (Lemma 1 of Shani et al. (2020) and Lemma 14 of Cassel & Rosenberg (2024)). *Let* $\pi, \widehat{\pi}$ *be two policies, and let* $\mathcal{M} = (H, \mathcal{S}, \mathcal{A}, \{\mathbb{P}_h\}_{h=1}^H, \{\ell_h\}_{h=1}^H, s_1)$ *be a (possibly sub) MDP. For all* $h \in [H]$, *let* $\widehat{Q}_{\ell,h} : \mathcal{S} \times \mathcal{A} \to \mathbb{R}$ *be an arbitrary function, and let* $\widehat{V}_{\ell,h}(s) = \left\langle \widehat{Q}_{\ell,h}(s, \cdot), \widehat{\pi}_h(\cdot \mid s) \right\rangle$ *for all* $s \in \mathcal{S}$. *Then,*

$$V_{\ell,1}^{\pi}(s_1) - \widehat{V}_{\ell,1}(s_1) = \mathbb{E}_{\mathbb{P},\pi}\left[\sum_{h=1}^H \left\langle \widehat{Q}_{\ell,h}(s_h, \cdot), \pi_h(\cdot \mid s_h) - \widehat{\pi}_h(\cdot \mid s_h) \right\rangle \bigg| s_1 \right]$$

$$+ \mathbb{E}_{\mathbb{P},\pi}\left[\sum_{h=1}^H \ell_h(s_h, a_h) + \mathbb{P}_h\widehat{V}_{\ell,h+1}(s_h, a_h) - \widehat{Q}_{\ell,h}(s_h, a_h) \bigg| s_1 \right],$$

*where* $V_{\ell,1}^{\pi}$ *is the value function of* $\pi$, *and* $\mathbb{P}_h\widehat{V}_{\ell,h+1}(s, a) = \sum_{s' \in \mathcal{S}} \mathbb{P}_h(s' \mid s, a)\widehat{V}_{\ell,h+1}(s')$.

Since we assume that the initial state $s_1$ is fixed, we omit it when clear from the context for simplicity.

## F   PARAMETERIZATIONS AND FUNCTION CLASSES

In this section, we introduce the parameterizations of $\widehat{Q}$ and $\widehat{\pi}$. Following this, we define the function classes to which the value function estimates and policies belong.

**Parameterization**   For the parameterization of $\widehat{Q}$, given $\beta \in \mathbb{R}, w \in \mathbb{R}^d, \Lambda \in \mathbb{R}^{d \times d}$, we define $\widehat{Q}(\cdot, \cdot; \beta, w, \Lambda)$ as

$$\widehat{Q}(\cdot, \cdot; \beta, w, \Lambda) = \left(\phi(\cdot, \cdot)^\top w - \beta \|\phi(\cdot, \cdot)\|_\Lambda\right) \cdot \sigma(-\beta_w \|\phi(\cdot, \cdot)\|_\Lambda + \log K). \tag{6}$$

Next, we consider the parameterization of $\widehat{\pi}$. Let $n \in \mathbb{Z}_+$ denote the number of mixing steps.

For the parameterization of $\widehat{\pi}$, given $n, \{\beta_i, w_i\}_{i=0}^n, \Lambda$ and a mixing parameter $\theta \in (0, 1)$, we first generate policies $\{\widehat{\pi}_i\}_{i=0}^{n+1}$ recursively, and define the final policy as $\widehat{\pi}(\cdot|\cdot; \{\beta_i, w_i\}_{i=0}^n, \Lambda)$:

$$
\begin{aligned}
\text{Generate } \widehat{\pi}_i: \quad & \widehat{\pi}_0(\cdot \mid s) = \pi_{\text{unif}}(\cdot \mid s), \\
& \widetilde{\pi}_i(\cdot \mid s) = (1 - \theta)\widehat{\pi}_i(\cdot \mid s) + \theta \pi_{\text{unif}}(\cdot \mid s) \quad i = 0, \dots, n, \\
& \widehat{\pi}_{i+1}(\cdot \mid s) \propto \widetilde{\pi}_i(\cdot \mid s) \exp\left(\widehat{Q}(s, \cdot; \beta_i, w_i, \Lambda)\right) \quad i = 0, \dots, n, \\
\text{Define}: \quad & \widehat{\pi}(\cdot \mid s; \{\beta_i, w_i\}_{i=0}^n, \Lambda) = \widehat{\pi}_{n+1}(\cdot \mid s).
\end{aligned}
\tag{7}
$$

We keep the policy parameterization in its recursive form for the following reason. Although we can easily show that $\widehat{\pi}(\cdot|\cdot; \{\beta_i, w_i\}_{i=0}^n, \Lambda)$ follows the weighted LSE softmax, i.e., $\sum_i \zeta_i \exp(\sum_j \widehat{Q}_j)$, the weight parameters $\zeta_i$ depend on $\{\beta_i, w_i\}_{i=0}^n$ and $\Lambda$, which makes analyzing this form difficult. Thus, obtaining the closed form of $\widehat{\pi}(\cdot|\cdot; \{\beta_i, w_i\}_{i=0}^n, \Lambda)$ is intractable, as specifying exact $\zeta_i$ is difficult.

Since $\widehat{\pi}_{n+1}(\cdot \mid s)$ induces a probability distribution over $\mathcal{A}$ for each $s \in \mathcal{S}$, (7) indeed defines a valid policy. Furthermore, the following lemma shows that $Q$-function estimates and policies generated by Algorithm 1 can be parameterized using (6), (7).

**Lemma 9.** *For any $e \in E$, consider $k \in K_e$. For some $n \geq 0$, let $k_e + nK^B$ be the last index that the mixing is applied before episode $k$, i.e., $n = \max\{0, \lfloor (k - 1 - k_e)/K^B \rfloor\}$. Let $\widehat{\pi}_h^k, \{\widehat{Q}_{f,h}^j, \widehat{Q}_{g,h}^j\}_{j=k_e}^{k-1}$ be the policy and $Q$-function estimates generated by Algorithm 1, respectively. Let $S_i$ be the index set defined as*

$$
S_i = \begin{cases} \{k_e + iK^B, \dots, k_e + (i+1)K^B - 1\} & \text{for } i = 0, \dots, n-1, \\ \{k_e + nK^B, \dots, k - 1\} & \text{for } i = n. \end{cases} \tag{8}
$$

*Then there exists $\{w_{f,h}^j, w_{g,h}^j\}_{j=k_e}^{k-1}, \Lambda$ such that for $j = k_e, \dots, k - 1$,*

$$\widehat{Q}_{f,h}^j(\cdot, \cdot) = \widehat{Q}(\cdot, \cdot; \beta_b, w_{f,h}^j, \Lambda), \quad \widehat{Q}_{g,h}^j(\cdot, \cdot) = \widehat{Q}(\cdot, \cdot; \beta_b, w_{g,h}^j, \Lambda).$$

*Furthermore, we have*

$$\widehat{\pi}_h^k(\cdot|\cdot) = \widehat{\pi}(\cdot \mid \cdot; \{\beta_i, w_i\}_{i=0}^n, \Lambda)$$

*where $\beta_i = -\alpha\beta_b \sum_{j \in S_i}(1 + Y_j), w_i = -\alpha \sum_{j \in S_i}(w_{f,h}^j + Y_j w_{g,h}^j)$ for $i = 0, \dots, n$, and $Y_j$ is the dual variable in episode $j$.*

*Proof.* Note that for any $j$, by algorithm

$$\widehat{Q}_{f,h}^j(\cdot, \cdot) = \left(\phi(\cdot, \cdot)^\top \left[\widehat{\theta}_{f,h}^j + \widehat{\psi}_h^j \widehat{V}_{f,h+1}^j\right] - \beta_b \|\phi(\cdot, \cdot)\|_{(\Lambda_h^{k_e})^{-1}}\right) \sigma(-\beta_w \|\phi(\cdot, \cdot)\|_{(\Lambda_h^{k_e})^{-1}} + \log K).$$

Then we can take $w_{f,h}^j = \widehat{\theta}_{f,h}^j + \widehat{\psi}_h^j \widehat{V}_{f,h+1}^j$ and $\Lambda = (\Lambda_h^{k_e})^{-1}$. Here, it is clear that $(2K)^{-1} I \preceq (\Lambda_h^{k_e})^{-1} \preceq I$. Also, we can apply the same argument to $\widehat{Q}_{g,h}^j$. Then the first statement is proved.

Let us prove the second statement. By the definition of $n$, the mixing is not applied from episode $k_e + nK^B + 1$ to $k - 1$. Then we have

$$\widehat{\pi}_h^k(\cdot \mid s) \propto \widetilde{\pi}_h^{k_e + nK^B}(\cdot \mid s) \exp\left(-\alpha \sum_{j \in S_n} (\widehat{Q}_{f,h}^j(s, \cdot) + Y_j \widehat{Q}_{g,h}^j(s, \cdot))\right).$$

Furthermore, since $(k_e + nK^B) - k_e \equiv 0 \mod K^B$, we know that $\widetilde{\pi}_h^{k_e + nK^B}$ is mixed. Then we have

$$\widetilde{\pi}_h^{k_e + nK^B}(\cdot \mid s) = (1 - \theta)\widehat{\pi}_h^{k_e + nK^B}(\cdot \mid s) + \theta \pi_{\text{unif}}(\cdot \mid s).$$

Similarly, we can deduce $\widehat{\pi}_h^{k_e+nK^B}$ using the fact that the mixing is not applied from episode $k_e + (n-1)K^B$ to $k_e + nK^B - 1$.

$$\widehat{\pi}_h^{k_e+nK^B}(\cdot \mid s) \propto \widetilde{\pi}_h^{k_e+(n-1)K^B}(\cdot \mid s)\exp\left(-\alpha\sum_{j\in S_{n-1}}(\widehat{Q}_{f,h}^j(s,\cdot)+Y_j\widehat{Q}_{g,h}^j(s,\cdot))\right).$$

We repeatedly apply these steps until episode $k_e$. Then we have for $i=0,\ldots,n-1$,

$$\widetilde{\pi}_h^{k_e+iK^B}(\cdot \mid s) = (1-\theta)\widehat{\pi}_h^{k_e+iK^B}(\cdot \mid s) + \theta\pi_{\mathrm{unif}}(\cdot \mid s),$$

$$\widehat{\pi}_h^{k_e+(i+1)K^B}(\cdot \mid s) \propto \widetilde{\pi}_h^{k_e+iK^B}(\cdot \mid s)\exp\left(-\alpha\sum_{j\in S_i}(\widehat{Q}_{f,h}^j(s,\cdot)+Y_j\widehat{Q}_{g,h}^j(s,\cdot))\right).$$

Note that $\widehat{\pi}_h^{k_e} = \pi_{\mathrm{unif}}$. Then we have $\widehat{\pi}_h^k = \widehat{\pi}_{n+1}$, where $\widehat{\pi}_{n+1}$ is recursively defined as

$$\widehat{\pi}_0(\cdot \mid s) = \pi_{\mathrm{unif}}(\cdot \mid s)$$
$$\widetilde{\pi}_i(\cdot \mid s) = (1-\theta)\widehat{\pi}_i(\cdot \mid s) + \theta\pi_{\mathrm{unif}}(\cdot \mid s) \quad \forall i = 0,\ldots,n$$

$$\widehat{\pi}_{i+1}(\cdot \mid s) \propto \widetilde{\pi}_i(\cdot \mid s)\exp\left(-\alpha\sum_{j\in S_i}(\widehat{Q}_{f,h}^j(s,\cdot)+Y_j\widehat{Q}_{g,h}^j(s,\cdot))\right) \quad \forall i = 0,\ldots,n.$$

Note that

$$-\alpha\sum_{j\in S_i}(\widehat{Q}_{f,h}^j(s,a)+Y_j\widehat{Q}_{g,h}^j(s,a))$$
$$= -\alpha\sum_{j\in S_i}\left(\phi(s,a)^\top(w_{f,h}^j + Y_jw_{g,h}^j) - \beta_b(1+Y_j)\|\phi(s,a)\|_\Lambda\right)\sigma(-\beta_w\|\phi(s,a)\|_\Lambda + \log K)$$
$$= (\phi(s,a)^\top w_i - \beta_i\|\phi(s,a)\|_\Lambda)\sigma(-\beta_w\|\phi(s,a)\|_\Lambda + \log K)$$
$$= \widehat{Q}(s,a;\beta_i,w_i,\Lambda)$$

where $\beta_i = -\alpha\beta_b\sum_{j\in S_i}(1+Y_j), w_i = -\alpha\sum_{j\in S_i}(w_{f,h}^j + Y_jw_{g,h}^j)$. This completes the proof for the second statement. $\square$

**Function Class**  Now, we define the function classes as follows. Given some boundedness constants $C_\beta, C_w, C_Q \geq 0$,

$$\widehat{\mathcal{Q}}(C_\beta,C_w,C_Q) = \left\{\widehat{Q}(\cdot,\cdot;\beta,w,\Lambda) : |\beta| \leq C_\beta, \|w\|_2 \leq C_w, (2K)^{-1}I \preceq \Lambda \preceq I, \|\widehat{Q}(\cdot,\cdot;\beta,w,\Lambda)\|_\infty \leq C_Q\right\}.$$

Unlike $\widehat{\mathcal{Q}}$, the class of policies has to be defined based on the number of mixing steps, since the formulation is determined by this number. Let $\widehat{\Pi}_n$ denote the set of policies that involve exactly $n$ mixing operations. Furthermore, we consider a boundedness constant $C_Y$ to incorporate the scale of the dual variable. Given $n \in \mathbb{Z}_+$ and some boundedness constants $C_\beta, C_w, C_Q, C_Y \geq 0$,

$$\widehat{\Pi}_n(C_\beta,C_w,C_Q,C_Y)$$
$$= \left\{\widehat{\pi}(\cdot \mid s; \{\beta_i,w_i\}_{i=0}^n, \Lambda) : \begin{array}{l} |\beta_i| \leq (1+C_Y)C_\beta, \|w_i\|_2 \leq (1+C_Y)C_w, \\ (2K)^{-1}I \preceq \Lambda \preceq I, \|\widehat{Q}(\cdot,\cdot;\beta_i,w_i,\Lambda)\|_\infty \leq (1+C_Y)C_Q, \end{array} i=0,\ldots,n\right\}.$$

Similar to $\widehat{\Pi}_n$, since $\widehat{V}$ is defined by $\widehat{Q}$ and $\widehat{\pi}$, we define the function class of $\widehat{V}$ for each $n$.

$$\widehat{\mathcal{V}}_n(C_\beta,C_w,C_Q,C_Y) = \left\{\widehat{V}(\cdot) : \widehat{V}(\cdot) = \sum_{a\in\mathcal{A}}\widehat{\pi}(a \mid \cdot)\widehat{Q}(\cdot,a), \begin{array}{l}\widehat{Q} \in \widehat{\mathcal{Q}}(C_\beta,C_w,C_Q), \\ \widehat{\pi} \in \widehat{\Pi}_n(KC_\beta,KC_w,KC_Q,C_Y)\end{array}\right\}.$$

Note that if we apply the policy mixing every $K^B$ episodes, then the number of mixing steps is at most $K^{1-B} = K^L$, where $L = 1 - B$. Thus, we define $\widehat{\mathcal{V}}(C_\beta,C_w,C_Q,C_Y), \widehat{\Pi}(C_\beta,C_w,C_Q,C_Y)$

as the unions over $n = 0, \ldots, K^L$ as follows.

$$\widehat{\mathcal{V}}(C_\beta, C_w, C_Q, C_Y) = \bigcup_{n=0}^{K^L} \widehat{\mathcal{V}}_n(C_\beta, C_w, C_Q, C_Y),$$

$$\widehat{\Pi}(C_\beta, C_w, C_Q, C_Y) = \bigcup_{n=0}^{K^L} \widehat{\Pi}_n(C_\beta, C_w, C_Q, C_Y).$$

## G  COVERING NUMBER

In this section, we show an upper bound on the covering number of $\widehat{\mathcal{V}}(C_\beta, C_w, C_Q, C_Y)$, which is crucial to analyze linear CMDPs. As a first step, we show that any $\widehat{Q} \in \widehat{\mathcal{Q}}(C_\beta, C_w, C_Q)$ is Lipschitz, i.e., the $\ell_\infty$-norm between $Q$-function estimates is bounded by the $\ell_2$-norm between their parameters. We closely follow the proof of Lemma 10 of Cassel & Rosenberg (2024).

**Lemma 10** (Lipschitz $\widehat{Q}$). *Let $1 \leq \beta_w, C_w, C_\beta$. For any $\widehat{Q}(\cdot, \cdot; \beta^1, w^1, \Lambda^1), \widehat{Q}(\cdot, \cdot; \beta^2, w^2, \Lambda^2) \in \widehat{\mathcal{Q}}(C_\beta, C_w, C_Q)$, we have*

$$\left\| \widehat{Q}(\cdot, \cdot; \beta^1, w^1, \Lambda^1) - \widehat{Q}(\cdot, \cdot; \beta^2, w^2, \Lambda^2) \right\|_\infty \leq 4\sqrt{K} \beta_w \max\{C_w, C_\beta\} \left\| (\beta^1, w^1, \Lambda^1) - (\beta^2, w^2, \Lambda^2) \right\|_2$$

*where $\left\| (\beta^1, w^1, \Lambda^1) - (\beta^2, w^2, \Lambda^2) \right\|_2$ is defined in* (9).

*Proof.* Consider

$$|\widehat{Q}(s, a; \beta^1, w^1, \Lambda^1) - \widehat{Q}(s, a; \beta^2, w^2, \Lambda^2)|$$
$$\leq \underbrace{|\widehat{Q}(s, a; \beta^1, w^1, \Lambda^1) - \widehat{Q}(s, a; \beta^2, w^1, \Lambda^1)|}_{(I)} + \underbrace{|\widehat{Q}(s, a; \beta^2, w^1, \Lambda^1) - \widehat{Q}(s, a; \beta^2, w^2, \Lambda^1)|}_{(II)}$$
$$+ \underbrace{|\widehat{Q}(s, a; \beta^2, w^2, \Lambda^1) - \widehat{Q}(s, a; \beta^2, w^2, \Lambda^2)|}_{(III)}.$$

We bound each term individually. Note that $(2K)^{-1}I \preceq \Lambda^1 \preceq I$. For (I), since $\|\phi(s, a)\|_{\Lambda^1} \leq \|(\Lambda^1)^{1/2}\|_2 \|\phi(s, a)\|_2 \leq 1$ and $|\sigma(z)| \leq 1$ for any $z \in \mathbb{R}$,

$$(I) = |\beta^1 - \beta^2| \cdot \|\phi(s, a)\|_{\Lambda^1} \sigma(-\beta_w \|\phi(s, a)\|_{\Lambda^1} + \log K)$$
$$\leq |\beta^1 - \beta^2|.$$

For (II), by the Cauchy-Schwarz inequality,

$$(II) = |\phi(s, a)^\top (w^1 - w^2)| \sigma(-\beta_w \|\phi(s, a)\|_{\Lambda^1} + \log K)$$
$$\leq \|w^1 - w^2\|_2$$

For (III), by the triangle inequality,

$$(III) = |\widehat{Q}(s, a; \beta^2, w^2, \Lambda^1) - \widehat{Q}(s, a; \beta^2, w^2, \Lambda^2)|$$
$$\leq |\phi(s, a)^\top w^2| \cdot |\sigma(-\beta_w \|\phi(s, a)\|_{\Lambda^1} + \log K) - \sigma(-\beta_w \|\phi(s, a)\|_{\Lambda^2} + \log K)|$$
$$+ \beta^2 |\|\phi(s, a)\|_{\Lambda^1} - \|\phi(s, a)\|_{\Lambda^2}| \cdot \sigma(-\beta_w \|\phi(s, a)\|_{\Lambda^1} + \log K)$$
$$+ \beta^2 \|\phi(s, a)\|_{\Lambda^2} \cdot |\sigma(-\beta_w \|\phi(s, a)\|_{\Lambda^1} + \log K) - \sigma(-\beta_w \|\phi(s, a)\|_{\Lambda^2} + \log K)|.$$

Note that the sigmoid function is 1-Lipschitz on $\mathbb{R}$, and the triangle inequality implies that $|\|\phi(s, a)\|_{\Lambda^1} - \|\phi(s, a)\|_{\Lambda^2}| \leq \|(\Lambda^1)^{1/2} - (\Lambda^2)^{1/2}\|_2$. Then we can deduce that

$$|\widehat{Q}(s, a; \beta^2, w^2, \Lambda^1) - \widehat{Q}(s, a; \beta^2, w^2, \Lambda^2)|$$
$$\leq C_w \beta_w \|(\Lambda^1)^{1/2} - (\Lambda^2)^{1/2}\|_2 + C_\beta \|(\Lambda^1)^{1/2} - (\Lambda^2)^{1/2}\|_2 + C_\beta \beta_w \|(\Lambda^1)^{1/2} - (\Lambda^2)^{1/2}\|_2$$
$$\leq 3 \max\{C_w \beta_w, C_\beta, C_\beta \beta_w\} \|(\Lambda^1)^{1/2} - (\Lambda^2)^{1/2}\|_2$$
$$\leq 3 \max\{C_w \beta_w, C_\beta, C_\beta \beta_w\} \cdot \frac{1}{2\sqrt{1/(2K)}} \|\Lambda^1 - \Lambda^2\|_2$$
$$\leq (3/\sqrt{2})\sqrt{K} \beta_w \max\{C_w, C_\beta\} \|\Lambda^1 - \Lambda^2\|_2$$

where the third inequality is due to Lemma 31, and the last inequality is because we assumed that $\beta_w \geq 1$. Note that we know $\|\Lambda^1 - \Lambda^2\|_2 \leq \|\Lambda^1 - \Lambda^2\|_F$ where $\|\cdot\|_F$ denotes the Frobenius norm. Finally, we show that for any $(s, a) \in \mathcal{S} \times \mathcal{A}$,

$$
\begin{aligned}
&|\widehat{Q}(s, a; \beta^1, w^1, \Lambda^1) - \widehat{Q}(s, a; \beta^2, w^2, \Lambda^2)| \\
&\leq \|w^1 - w^2\|_2 + |\beta^1 - \beta^2| + (3/\sqrt{2})\sqrt{K}\beta_w \max\{C_w, C_\beta\}\|\Lambda^1 - \Lambda^2\|_F \\
&\leq \sqrt{3\left(\|w^1 - w^2\|_2^2 + |\beta^1 - \beta^2|^2 + ((3/\sqrt{2})\sqrt{K}\beta_w \max\{C_w, C_\beta\})^2\|\Lambda^1 - \Lambda^2\|_F^2\right)} \\
&\leq 4\sqrt{K}\beta_w \max\{C_w, C_\beta\}\left\|(\beta^1, w^1, \Lambda^1) - (\beta^2, w^2, \Lambda^2)\right\|_2
\end{aligned}
$$

where the second inequality follows from the Cauchy-Schwarz inequality, and the last inequality is due to $1 \leq (3/\sqrt{2})\sqrt{K}\beta_w \max\{C_w, C_\beta\}$, as we assumed that $1 \leq \beta_w, C_w, C_\beta$. Here, $\left\|(\beta^1, w^1, \Lambda^1) - (\beta^2, w^2, \Lambda^2)\right\|_2$ denotes

$$
\left\|(\beta^1, w^1, \Lambda^1) - (\beta^2, w^2, \Lambda^2)\right\|_2 = \sqrt{|\beta^1 - \beta^2|^2 + \|w^1 - w^2\|_2^2 + \|\Lambda^1 - \Lambda^2\|_F^2}. \tag{9}
$$

$\square$

Now, we have to show that the policy parameterization given in (7) satisfies a Lipschitz property, i.e., $\ell_1$-norm between any two policies is bounded by the $\ell_2$-norm between their parameters.

**Lemma 11** (Lipschitz $\widehat{\pi}$). *Let* $1 \leq \beta_w, C_w, C_\beta$ *and* $0 \leq C_Y$. *Suppose that two policies* $\widehat{\pi}^1, \widehat{\pi}^2 \in \widehat{\Pi}_n(C_\beta, C_w, C_Q, C_Y)$ *are parameterized by* $\{\beta_i^1, w_i^1\}_{i=0}^n, \Lambda^1$ *and* $\{\beta_i^2, w_i^2\}_{i=0}^n, \Lambda^2$, *respectively. Then the following holds for any* $s \in \mathcal{S}$.

$$
\|\widehat{\pi}^1(\cdot \mid s) - \widehat{\pi}^2(\cdot \mid s)\|_1
$$

$$
\leq 32\sqrt{(n+1)K}\beta_w(1 + C_Y)\max\{C_w, C_\beta\}\left(\frac{8|\mathcal{A}|}{\theta}\right)^n \sqrt{\sum_{i=0}^n \|(\beta_i^1, w_i^1, \Lambda^1) - (\beta_i^2, w_i^2, \Lambda^2)\|_2^2}.
$$

*Proof.* Fix $s \in \mathcal{S}$. Let $\{\widehat{\pi}_i^1, \widetilde{\pi}_i^1\}_{i=0}^{n+1}, \{\widehat{\pi}_i^2, \widetilde{\pi}_i^2\}_{i=0}^{n+1}$ be the sequences of policies recursively generated by (7) to define $\widehat{\pi}^1, \widehat{\pi}^2$, respectively. Then it follows that

$$
\begin{aligned}
\widehat{\pi}^1(\cdot \mid s) &= \widehat{\pi}_{n+1}^1(\cdot \mid s) \propto \widetilde{\pi}_n^1(\cdot \mid s)\exp(\widehat{Q}(s, \cdot; \beta_n^1, w_n^1, \Lambda^1)), \\
\widehat{\pi}^2(\cdot \mid s) &= \widehat{\pi}_{n+1}^2(\cdot \mid s) \propto \widetilde{\pi}_n^2(\cdot \mid s)\exp(\widehat{Q}(s, \cdot; \beta_n^2, w_n^2, \Lambda^2)).
\end{aligned}
$$

Note that $\widetilde{\pi}_n^1(a \mid s), \widetilde{\pi}_n^2(a \mid s) > 0$ for all $a \in \mathcal{A}$, since they are perturbed. Then we can define $\log \widetilde{\pi}_n^1(a \mid s), \log \widetilde{\pi}_n^2(a \mid s)$, and it leads to

$$
\begin{aligned}
\widehat{\pi}_{n+1}^1(\cdot \mid s) &\propto \exp\left(\log \widetilde{\pi}_n^1(\cdot \mid s) + \widehat{Q}(s, \cdot; \beta_n^1, w_n^1, \Lambda^1)\right), \\
\widehat{\pi}_{n+1}^2(\cdot \mid s) &\propto \exp\left(\log \widetilde{\pi}_n^2(\cdot \mid s) + \widehat{Q}(s, \cdot; \beta_n^2, w_n^2, \Lambda^2)\right).
\end{aligned} \tag{10}
$$

By Lemma 24,

$$
\begin{aligned}
&\|\widehat{\pi}_{n+1}^1(\cdot \mid s) - \widehat{\pi}_{n+1}^2(\cdot \mid s)\|_1 \\
&\leq 8\left\|\log \widetilde{\pi}_n^1(\cdot \mid s) + \widehat{Q}(s, \cdot; \beta_n^1, w_n^1, \Lambda^1) - \log \widetilde{\pi}_n^2(\cdot \mid s) - \widehat{Q}(s, \cdot; \beta_n^2, w_n^2, \Lambda^2)\right\|_\infty \\
&\leq 8\left\|\log \widetilde{\pi}_n^1(\cdot \mid s) - \log \widetilde{\pi}_n^2(\cdot \mid s)\right\|_\infty + 8\left\|\widehat{Q}(s, \cdot; \beta_n^1, w_n^1, \Lambda^1) - \widehat{Q}(s, \cdot; \beta_n^2, w_n^2, \Lambda^2)\right\|_\infty.
\end{aligned}
$$

Note that $\widetilde{\pi}_n^1(a \mid s), \widetilde{\pi}_n^2(a \mid s) \geq \theta/|\mathcal{A}|$ for all $a \in \mathcal{A}$ due to the definition. Then we can utilize the Lipschitzness of log function in $[\theta/|\mathcal{A}|, \infty)^{|\mathcal{A}|}$. Thus, by Lemma 21,

$$
\begin{aligned}
\left\|\log \widetilde{\pi}_n^1(\cdot \mid s) - \log \widetilde{\pi}_n^2(\cdot \mid s)\right\|_\infty &\leq \frac{|\mathcal{A}|}{\theta}\|\widetilde{\pi}_n^1(\cdot \mid s) - \widetilde{\pi}_n^2(\cdot \mid s)\|_1 \\
&= \frac{|\mathcal{A}|}{\theta}(1 - \theta)\|\widehat{\pi}_n^1(\cdot \mid s) - \widehat{\pi}_n^2(\cdot \mid s)\|_1 \\
&\leq \frac{|\mathcal{A}|}{\theta}\|\widehat{\pi}_n^1(\cdot \mid s) - \widehat{\pi}_n^2(\cdot \mid s)\|_1
\end{aligned} \tag{11}
$$

where the equality is due to $\widetilde{\pi}_n^1(\cdot \mid s) = (1-\theta)\widehat{\pi}_n^1(\cdot \mid s) + \theta\pi_{\text{unif}}(\cdot \mid s)$ and $\widetilde{\pi}_n^2(\cdot \mid s) = (1-\theta)\widehat{\pi}_n^2(\cdot \mid s) + \theta\pi_{\text{unif}}(\cdot \mid s)$. Plugging (11) into (10), we have a recursive relation, and it leads to

$$\|\widehat{\pi}_{n+1}^1(\cdot \mid s) - \widehat{\pi}_{n+1}^2(\cdot \mid s)\|_1$$

$$\leq \frac{8|\mathcal{A}|}{\theta} \left\|\widehat{\pi}_n^1(\cdot \mid s) - \widehat{\pi}_n^2(\cdot \mid s)\right\|_1 + 8 \left\|\widehat{Q}(s, \cdot; \beta_n^1, w_n^1, \Lambda^1) - \widehat{Q}(s, \cdot; \beta_n^2, w_n^2, \Lambda^2)\right\|_\infty$$

$$\leq \left(\frac{8|\mathcal{A}|}{\theta}\right)^2 \left\|\widehat{\pi}_{n-1}^1(\cdot \mid s) - \widehat{\pi}_{n-1}^2(\cdot \mid s)\right\|_1$$

$$+ 8\sum_{i=0}^{1} \left(\frac{8|\mathcal{A}|}{\theta}\right)^i \left\|\widehat{Q}(s, \cdot; \beta_{n-i}^1, w_{n-i}^1, \Lambda^1) - \widehat{Q}(s, \cdot; \beta_{n-i}^2, w_{n-i}^2, \Lambda^2)\right\|_\infty$$

$$\vdots$$

$$\leq \left(\frac{8|\mathcal{A}|}{\theta}\right)^{n+1} \left\|\widehat{\pi}_0^1(\cdot \mid s) - \widehat{\pi}_0^2(\cdot \mid s)\right\|_1$$

$$+ 8\sum_{i=0}^{n} \left(\frac{8|\mathcal{A}|}{\theta}\right)^i \left\|\widehat{Q}(s, \cdot; \beta_{n-i}^1, w_{n-i}^1, \Lambda^1) - \widehat{Q}(s, \cdot; \beta_{n-i}^2, w_{n-i}^2, \Lambda^2)\right\|_\infty$$

$$= 8\sum_{i=0}^{n} \left(\frac{8|\mathcal{A}|}{\theta}\right)^i \left\|\widehat{Q}(s, \cdot; \beta_{n-i}^1, w_{n-i}^1, \Lambda^1) - \widehat{Q}(s, \cdot; \beta_{n-i}^2, w_{n-i}^2, \Lambda^2)\right\|_\infty$$

where the equality is due to $\widehat{\pi}_0^1(\cdot \mid s) = \widehat{\pi}_0^2(\cdot \mid s) = \pi_{\text{unif}}(\cdot \mid s)$. Furthermore, by the Cauchy-Schwarz inequality,

$$\|\widehat{\pi}_{n+1}^1(\cdot \mid s) - \widehat{\pi}_{n+1}^2(\cdot \mid s)\|_1$$

$$\leq 8\sqrt{\sum_{i=0}^{n} \left(\frac{8|\mathcal{A}|}{\theta}\right)^{2i}} \sqrt{\sum_{i=0}^{n} \left\|\widehat{Q}(s, \cdot; \beta_i^1, w_i^1, \Lambda^1) - \widehat{Q}(s, \cdot; \beta_i^2, w_i^2, \Lambda^2)\right\|_\infty^2}$$

$$\leq 8\sqrt{(n+1)\left(\frac{8|\mathcal{A}|}{\theta}\right)^{2n}} \sqrt{\sum_{i=0}^{n} \left\|\widehat{Q}(s, \cdot; \beta_i^1, w_i^1, \Lambda^1) - \widehat{Q}(s, \cdot; \beta_i^2, w_i^2, \Lambda^2)\right\|_\infty^2}.$$

Note that $\widehat{Q}(s, \cdot; \beta_i^1, w_i^1, \Lambda^1), \widehat{Q}(s, \cdot; \beta_i^2, w_i^2, \Lambda^2) \in \widehat{\mathcal{Q}}((1+C_Y)C_\beta, (1+C_Y)C_w, (1+C_Y)C_Q)$. By the Lipschitzness of $\widehat{Q}$ (Lemma 10),

$$\|\widehat{\pi}_{n+1}^1(\cdot \mid s) - \widehat{\pi}_{n+1}^2(\cdot \mid s)\|_1$$

$$\leq 8\sqrt{(n+1)\left(\frac{8|\mathcal{A}|}{\theta}\right)^{2n}} \sqrt{\sum_{i=0}^{n} \left(4\sqrt{K}\beta_w(1+C_Y)\max\{C_w, C_\beta\}\right)^2 \|(\beta_i^1, w_i^1, \Lambda^1) - (\beta_i^2, w_i^2, \Lambda^2)\|_2^2}$$

$$= 32\sqrt{(n+1)K}\beta_w(1+C_Y)\max\{C_w, C_\beta\}\left(\frac{8|\mathcal{A}|}{\theta}\right)^n \sqrt{\sum_{i=0}^{n} \|(\beta_i^1, w_i^1, \Lambda^1) - (\beta_i^2, w_i^2, \Lambda^2)\|_2^2}$$

as desired. $\qquad\square$

Based on the Lipschitz properties that we have shown, we show a Lipschitz property of $\widehat{V}$, and it leads to an upper bound on the covering number of $\widehat{\mathcal{V}}_n(C_\beta, C_w, C_Q, C_Y)$.

**Lemma 12.** *Let $1 \leq \beta_w, C_\beta, C_w, C_Q$ and $0 \leq C_Y$. Given $\epsilon > 0$, let $\mathcal{N}_\epsilon(\widehat{\mathcal{V}}_n(C_\beta, C_w, C_Q, C_Y))$ denote the $\epsilon$-covering number of $\widehat{\mathcal{V}}_n(C_\beta, C_w, C_Q, C_Y)$ with respect to the $\ell_\infty$-norm. Then we have*

$$\log \mathcal{N}_\epsilon(\widehat{\mathcal{V}}_n(C_\beta, C_w, C_Q, C_Y)) \leq 3(n+2)^2 d^2 \log((8|\mathcal{A}|/\theta)(1 + 2C_1C_2/\epsilon))$$

*where*

$$\begin{aligned}
C_1 &= 33\sqrt{(n+1)K^3}\beta_w \max\{C_w, C_\beta\}C_Q(1+C_Y), \\
C_2 &= (n+2)(1+C_Y)K(C_\beta + C_w) + (n+2)\sqrt{d}.
\end{aligned} \tag{12}$$

*Proof.* Recall that $\widehat{V} \in \widehat{\mathcal{V}}_n(C_\beta, C_w, C_Q, C_Y)$ can be expressed as $\widehat{V}(\cdot) = \sum_{a \in \mathcal{A}} \widehat{\pi}(a \mid \cdot)\widehat{Q}(\cdot, a)$, where $\widehat{Q} \in \widehat{\mathcal{Q}}(C_\beta, C_w, C_Q)$, $\widehat{\pi} \in \widehat{\Pi}_n(KC_\beta, KC_w, KC_Q, C_Y)$. Then consider $\widehat{V}_1, \widehat{V}_2 \in \widehat{\mathcal{V}}_n(C_\beta, C_w, C_Q, C_Y)$ such that $\widehat{V}^1(\cdot) = \sum_{a \in \mathcal{A}} \widehat{\pi}^1(a \mid \cdot)\widehat{Q}^1(\cdot, a)$ and $\widehat{V}^2(\cdot) = \sum_{a \in \mathcal{A}} \widehat{\pi}^2(a \mid \cdot)\widehat{Q}^2(\cdot, a)$. Suppose that each of those are parameterized as follows.

$$\widehat{\pi}^1 = \widehat{\pi}(\cdot | \cdot; \{\beta_i^{\pi,1}, w_i^{\pi,1}\}_{i=0}^n, \Lambda^{\pi,1}) \in \widehat{\Pi}_n(KC_\beta, KC_w, KC_Q, C_Y),$$
$$\widehat{\pi}^2 = \widehat{\pi}(\cdot | \cdot; \{\beta_i^{\pi,2}, w_i^{\pi,2}\}_{i=0}^n, \Lambda^{\pi,2}) \in \widehat{\Pi}_n(KC_\beta, KC_w, KC_Q, C_Y),$$
$$\widehat{Q}^1 = \widehat{Q}(\cdot, \cdot; \beta^{Q,1}, w^{Q,1}, \Lambda^{Q,1}) \in \widehat{\mathcal{Q}}(C_\beta, C_w, C_Q),$$
$$\widehat{Q}^2 = \widehat{Q}(\cdot, \cdot; \beta^{Q,2}, w^{Q,2}, \Lambda^{Q,2}) \in \widehat{\mathcal{Q}}(C_\beta, C_w, C_Q).$$

Then for any $s \in \mathcal{S}$,

$$\left|\widehat{V}^1(s) - \widehat{V}^2(s)\right|$$

$$= \left|\sum_{a \in \mathcal{A}} \widehat{\pi}^1(a \mid s)\widehat{Q}^1(s, a) - \sum_{a \in \mathcal{A}} \widehat{\pi}^2(a \mid s)\widehat{Q}^2(s, a)\right|$$

$$\leq \left|\sum_{a \in \mathcal{A}} \widehat{\pi}^1(a \mid s)\widehat{Q}^1(s, a) - \sum_{a \in \mathcal{A}} \widehat{\pi}^1(a \mid s)\widehat{Q}^2(s, a)\right| + \left|\sum_{a \in \mathcal{A}} \widehat{\pi}^1(a \mid s)\widehat{Q}^2(s, a) - \sum_{a \in \mathcal{A}} \widehat{\pi}^2(a \mid s)\widehat{Q}^2(s, a)\right|$$

$$\leq \|\widehat{\pi}^1(\cdot \mid s)\|_1 \|\widehat{Q}^1(s, \cdot) - \widehat{Q}^2(s, \cdot)\|_\infty + \|\widehat{\pi}^1(\cdot \mid s) - \widehat{\pi}^2(\cdot \mid s)\|_1 \|\widehat{Q}^2(s, \cdot)\|_\infty$$

$$= \|\widehat{Q}^1(s, \cdot) - \widehat{Q}^2(s, \cdot)\|_\infty + \|\widehat{\pi}^1(\cdot \mid s) - \widehat{\pi}^2(\cdot \mid s)\|_1 \|\widehat{Q}^2(s, \cdot)\|_\infty$$

where the first inequality is due to the triangle inequality, and the second inequality is due to Hölder's inequality. By Lemma 10,

$$\|\widehat{Q}^1(s, \cdot) - \widehat{Q}^2(s, \cdot)\|_\infty \leq 4\sqrt{K}\beta_w \max\{C_w, C_\beta\} \left\|(\beta^{Q,1}, w^{Q,1}, \Lambda^{Q,1}) - (\beta^{Q,2}, w^{Q,2}, \Lambda^{Q,2})\right\|_2.$$

Furthermore, for the second term,

$$\|\widehat{\pi}^1(\cdot \mid s) - \widehat{\pi}^2(\cdot \mid s)\|_1 \|\widehat{Q}^2(s, \cdot)\|_\infty$$

$$\leq C_Q \|\widehat{\pi}^1(\cdot \mid s) - \widehat{\pi}^2(\cdot \mid s)\|_1$$

$$\leq C_Q \cdot 32\sqrt{(n+1)K}\beta_w(1+C_Y)\max\{KC_w, KC_\beta\}\left(\frac{8|\mathcal{A}|}{\theta}\right)^n$$

$$\times \sqrt{\sum_{i=0}^n \|(\beta_i^{\pi,1}, w_i^{\pi,1}, \Lambda^{\pi,1}) - (\beta_i^{\pi,2}, w_i^{\pi,2}, \Lambda^{\pi,2})\|_2^2}$$

where the first inequality is due to $\|\widehat{Q}^2\|_\infty \leq C_Q$ for any $\widehat{Q}^2 \in \widehat{\mathcal{Q}}(C_\beta, C_w, C_Q)$, and the second inequality is due to Lemma 11. Then we deduce that

$$\max_{s \in \mathcal{S}} \left|\widehat{V}^1(s) - \widehat{V}^2(s)\right|$$

$$\leq 4\sqrt{K}\beta_w \max\{C_w, C_\beta\} \left\|(\beta^{Q,1}, w^{Q,1}, \Lambda^{Q,1}) - (\beta^{Q,2}, w^{Q,2}, \Lambda^{Q,2})\right\|_2$$

$$+ 32C_Q\sqrt{(n+1)K^3}\beta_w(1+C_Y)\max\{C_w, C_\beta\}\left(\frac{8|\mathcal{A}|}{\theta}\right)^n \sqrt{\sum_{i=0}^n \|(\beta_i^{\pi,1}, w_i^{\pi,1}, \Lambda^{\pi,1}) - (\beta_i^{\pi,2}, w_i^{\pi,2}, \Lambda^{\pi,2})\|_2^2}$$

$$\leq \sqrt{(4\sqrt{K}\beta_w \max\{C_w, C_\beta\})^2 + (32C_Q\sqrt{(n+1)K^3}\beta_w(1+C_Y)\max\{C_w, C_\beta\}(8|\mathcal{A}|/\theta)^n)^2}$$

$$\times \sqrt{\|(\beta^{Q,1}, w^{Q,1}, \Lambda^{Q,1}) - (\beta^{Q,2}, w^{Q,2}, \Lambda^{Q,2})\|_2^2 + \sum_{i=0}^n \|(\beta_i^{\pi,1}, w_i^{\pi,1}, \Lambda^{\pi,1}) - (\beta_i^{\pi,2}, w_i^{\pi,2}, \Lambda^{\pi,2})\|_2^2}$$

$$\leq 33\sqrt{(n+1)K^3}\beta_w(1+C_Y)\max\{C_w, C_\beta\}C_Q\left(\frac{8|\mathcal{A}|}{\theta}\right)^n$$

$$\times \sqrt{\|(\beta^{Q,1}, w^{Q,1}, \Lambda^{Q,1}) - (\beta^{Q,2}, w^{Q,2}, \Lambda^{Q,2})\|_2^2 + \sum_{i=0}^n \|(\beta_i^{\pi,1}, w_i^{\pi,1}, \Lambda^{\pi,1}) - (\beta_i^{\pi,2}, w_i^{\pi,2}, \Lambda^{\pi,2})\|_2^2}$$

where the second inequality is due to the Cauchy-Schwarz inequality, and the last inequality is due to the assumption that $1 \leq C_Q$ and $\theta \in (0, 1)$.

Note that

$$\|\beta^{Q,1}, w^{Q,1}, \Lambda^{Q,1}, \{\beta_i^{\pi,1}, w_i^{\pi,1}, \Lambda^{\pi,1}\}_{i=0}^n\|_2 \leq |\beta^{Q,1}| + \|w^{Q,1}\|_2 + \|\Lambda^{Q,1}\|_F + \sum_{i=0}^n (|\beta_i^{\pi,1}| + \|w_i^{\pi,1}\|_2 + \|\Lambda^{\pi,1}\|_F)$$

$$\leq C_\beta + C_w + \sqrt{d} + (n+1)(1 + C_Y)K(C_\beta + C_w) + (n+1)\sqrt{d}$$

$$\leq (n+2)(1 + C_Y)K(C_\beta + C_w) + (n+2)\sqrt{d}.$$

Note that $(\beta^{Q,1}, w^{Q,1}, \Lambda^{Q,1}, \{\beta_i^{\pi,1}, w_i^{\pi,1}, \Lambda^{\pi,1}\}_{i=0}^n)$ can be viewed as a $(n+2)(1 + d + d^2)$-dimensional vector. Since $1 + d + d^2 \leq 3d^2$, by Lemma 26,

$$\log \mathcal{N}_\epsilon(\widehat{\mathcal{V}}_n(C_\beta, C_w, C_Q, C_Y)) \leq 3(n+2)d^2 \log\left(1 + 2(8|\mathcal{A}|/\theta)^n C_1 C_2 / \epsilon\right)$$

where

$$C_1 = 33\sqrt{(n+1)K^3}\beta_w C_Q (1 + C_Y)\max\{C_w, C_\beta\},$$

$$C_2 = (n+2)(1 + C_Y)K(C_\beta + C_w) + (n+2)\sqrt{d}.$$

Furthermore, the log term contains an exponential term in $n$, we further deduce as follows.

$$\log\left(1 + 2(8|\mathcal{A}|/\theta)^n C_1 C_2 / \epsilon\right) \leq n\log(8|\mathcal{A}|/\theta) + \log(1 + 2C_1 C_2 / \epsilon)$$

$$\leq (n+1)\log((8|\mathcal{A}|/\theta)(1 + 2C_1 C_2 / \epsilon)).$$

Finally, we have

$$\log \mathcal{N}_\epsilon(\widehat{\mathcal{V}}_n(C_\beta, C_w, C_Q, C_Y)) \leq 3(n+2)^2 d^2 \log((8|\mathcal{A}|/\theta)(1 + 2C_1 C_2 / \epsilon)).$$

$\square$

Finally, we show an upper bound on the covering number of $\widehat{\mathcal{V}}(C_\beta, C_w, C_Q, C_Y)$.

**Lemma 13.** *Let* $1 \leq \beta_w, C_\beta, C_w, C_Q$ *and* $0 \leq C_Y$. *Given* $\epsilon > 0$, *let* $\mathcal{N}_\epsilon(\widehat{\mathcal{V}}(C_\beta, C_w, C_Q, C_Y))$ *denote the* $\epsilon$-*covering number of* $\widehat{\mathcal{V}}(C_\beta, C_w, C_Q, C_Y)$ *with respect to the* $\ell_\infty$-*norm, where* $\widehat{\mathcal{V}}(C_\beta, C_w, C_Q, C_Y) = \bigcup_{n=0}^{K^L} \widehat{\mathcal{V}}_n(C_\beta, C_w, C_Q, C_Y)$. *Then we have*

$$\log \mathcal{N}_\epsilon(\widehat{\mathcal{V}}(C_\beta, C_w, C_Q, C_Y)) \leq 3(K^L + 2)^2 d^2 \log\left((K^L + 1)\frac{8|\mathcal{A}|}{\theta}(1 + \frac{2C_1 C_2}{\epsilon})\right)$$

*where* $C_1, C_2$ *are defined in* (12) *with* $n = K^L$.

*Proof.* For each $n = 0, \ldots, K^L$, let $\mathcal{C}_n \subseteq \widehat{\mathcal{V}}_n(C_\beta, C_w, C_Q, C_Y)$ be an $\epsilon$-cover of $\widehat{\mathcal{V}}_n(C_\beta, C_w, C_Q, C_Y)$ with respect to the $\ell_\infty$-norm. By Lemma 12, suppose that the covers satisfy

$$\log|\mathcal{C}_n| \leq 3(n+2)^2 d^2 \log((8|\mathcal{A}|/\theta)(1 + 2C_1 C_2 / \epsilon)) \quad \forall n = 0, \ldots, K^L$$

where $C_1, C_2$ are defined in (12) with $n = K^L$. Furthermore, let

$$\mathcal{C} = \bigcup_{n=0}^{K^L} \mathcal{C}_n.$$

Then we claim that $\mathcal{C}$ is an $\epsilon$-cover of $\widehat{\mathcal{V}}(C_\beta, C_w, C_Q, C_Y)$. For any $\widehat{V} \in \widehat{\mathcal{V}}(C_\beta, C_w, C_Q, C_Y)$, since $\widehat{\mathcal{V}}(C_\beta, C_w, C_Q, C_Y)$ is defined as the union, there exists $m \in \{0, \ldots, K^L\}$ such that

$$\widehat{V} \in \widehat{\mathcal{V}}_m(C_\beta, C_w, C_Q, C_Y).$$

Since $\mathcal{C}_m$ is an $\epsilon$-cover of $\widehat{\mathcal{V}}_m(C_\beta, C_w, C_Q, C_Y)$, there exists $\widehat{V}_m \in \mathcal{C}_m \subseteq \mathcal{C}$ such that

$$\|\widehat{V} - \widehat{V}_m\|_\infty \leq \epsilon.$$

This implies that $\mathcal{C}$ is an $\epsilon$-cover of $\widehat{\mathcal{V}}(C_\beta, C_w, C_Q, C_Y)$ with respect to the $\ell_\infty$-norm. Furthermore, we have

$$\mathcal{N}_\epsilon(\widehat{\mathcal{V}}(C_\beta, C_w, C_Q, C_Y)) \leq |\mathcal{C}| \leq \sum_{n=0}^{K^L} |\mathcal{C}_n| \leq (K^L + 1)\left(\frac{8|\mathcal{A}|}{\theta}(1 + \frac{2C_1 C_2}{\epsilon})\right)^{3(K^L + 2)^2 d^2}$$

$$\leq \left((K^L + 1)\frac{8|\mathcal{A}|}{\theta}(1 + \frac{2C_1 C_2}{\epsilon})\right)^{3(K^L + 2)^2 d^2}$$

where the second inequality is true because $z \leq z^y$ for any $z, y \geq 1$. Finally, by taking $\log$ on both sides, we have

$$\log \mathcal{N}_\epsilon(\widehat{\mathcal{V}}(C_\beta, C_w, C_Q, C_Y)) \leq 3(K^L + 2)^2 d^2 \log\left((K^L + 1)\frac{8|\mathcal{A}|}{\theta}(1 + \frac{2C_1 C_2}{\epsilon})\right)$$

as desired. $\qquad\qquad\qquad\qquad\qquad\qquad\qquad\qquad\qquad\qquad\qquad\qquad\qquad\qquad\square$

## H GOOD EVENT

In this section, we introduce a high probability good event, denoted by $E_g$, which simplifies our analysis. We begin by presenting the formal definition of $E_g$. We define $E_g$ as

$$E_g = E_1 \cap E_2 \cap E_3. \tag{13}$$

$E_1, E_2, E_3$ are defined as

$$E_1 = \left\{\forall(k, h) \in [K] \times [H] : \|\theta_{f,h}^k - \widehat{\theta}_{f,h}^k\|_{\Lambda_h^k} \leq \beta_r, \|\theta_{g,h} - \widehat{\theta}_{g,h}^k\|_{\Lambda_h^k} \leq \beta_r\right\}, \tag{14}$$

$$E_2 = \left\{\forall(k, h, \ell) \in [K] \times [H] \times \{f, g\} : \|(\psi_h - \widehat{\psi}_h^k)\widehat{V}_{\ell,h+1}^k\|_{\Lambda_h^k} \leq \beta_p, \|\widehat{Q}_{\ell,h}^k\|_\infty \leq \beta_Q, Y_k \leq 11\eta H^3 K\right\}, \tag{15}$$

$$E_3 = \left\{\sum_{k \in [K]} \mathbb{E}_{\mathbb{P}, \widehat{\pi}^k}[W_k] \leq 2 \sum_{k \in [K]} W_k + 4H(3\beta_b + 8\beta_Q \beta_w^2) \log \frac{6K}{\delta}\right\}, \tag{16}$$

where

$$\beta_r = 2\sqrt{2d \log(6KH/\delta)},$$
$$\beta_p = 50(K^{1/4} + 1)dH\sqrt{\log(5H^2 K^2 |\mathcal{A}|/\delta)},$$
$$\beta_Q = 2H,$$
$$\beta_b = \beta_r + \beta_p,$$
$$\beta_w = 4\beta_b \log K,$$
$$W_k = \sum_{h \in [H]} \left(3\beta_b \|\phi(s_h^k, a_h^k)\|_{(\Lambda_h^k)^{-1}} + 8\beta_Q \beta_w^2 \|\phi(s_h^k, a_h^k)\|_{(\Lambda_h^k)^{-1}}^2\right).$$

We note that one of the key differences from Cassel & Rosenberg (2024) is that $E_2$ involves an upper bound of $Y_k$. This is because a (possibly polynomial in $d, H, K$) upper bound of $Y_k$ is required to prove that $E_g$ holds with high probability. In contrast, since we do not truncate $\widehat{Q}_{g,h}^k$, its trivial upper bound cannot be obtained. Thus, to avoid circular logic, we include it in $E_2$, and use induction to show (Step 3-2 of Lemma 16).

However, since directly proving $E_g$ holds with high probability is difficult, we instead consider a proxy good event and then show that it implies $E_g$. Here, we define the proxy good event $\bar{E}_g$ as

$$\bar{E}_g = E_1 \cap \bar{E}_2 \cap E_3, \tag{17}$$

where $\bar{E}_2$ is defined as

$$\bar{E}_2 = \left\{\forall(k, h, \widehat{V}) \in [K] \times [H] \times \widehat{\mathcal{V}}(\beta_b, 2K\beta_Q, \beta_Q, 11\eta H^3 K) : \|(\psi_h - \widehat{\psi}_h^k)\widehat{V}\|_{\Lambda_h^k} \leq \beta_p\right\}.$$

Based on the upper bound of the covering number of $\widehat{\mathcal{V}}(C_\beta, C_w, C_Q, C_Y)$, we prove that $\bar{E}_g$ holds with high probability.

**Lemma 14** (Proxy Good Event, $\bar{E}_g$). *Let $1 \leq \beta_w \leq K$, and let $\eta, \alpha \leq 1$. Then $\Pr[\bar{E}_g] \geq 1 - \delta$ for any $\delta \in (0, 1)$.*

*Proof.* We prove the statement by showing $\Pr[E_1] \geq 1 - \delta/3$, $\Pr[\bar{E}_2] \geq 1 - \delta/3$, and $\Pr[E_3] \geq 1 - \delta/3$. For $E_1$, by Lemma 29, we have for all $h \in [H], k \geq 1$ with probability at least $1 - \delta/3$,

$$\|\theta_{g,h} - \widehat{\theta}_{g,h}^k\|_{\Lambda_h^k} \leq 2\sqrt{2d\log(6KH/\delta)} := \beta_r.$$

Furthermore, it is clear that $\|\theta_{f,h}^k - \widehat{\theta}_{f,h}^k\|_{\Lambda_h^k} = 0 \leq \beta_r$ because we take $\widehat{\theta}_{f,h}^k = \theta_{f,h}^k$. Then for any $h, k, \ell \in \{f, g\}$, $E_1$ holds with probability at least $1 - \delta/3$.

Now we consider $\bar{E}_2$. By Lemma 30, for all $\widehat{V} \in \widehat{\mathcal{V}}(\beta_b, 2K\beta_Q, \beta_Q, 11\eta H^3 K)$, with probability at least $1 - \delta/3$,

$$\|(\psi_h - \widehat{\psi}_h^k)\widehat{V}\|_{\Lambda_h^k} \leq 4\beta_{Q,h}\sqrt{d\log(K+1) + 2\log(3H^2/\delta) + 2\log\mathcal{N}_\epsilon(\widehat{\mathcal{V}}(\beta_b, 2K\beta_Q, \beta_Q, 11\eta H^3 K))}. \tag{18}$$

The parameters in Algorithm 1 satisfy $1 \leq \beta_b, 2K\beta_Q, \beta_Q$. Then Lemma 13 can be applied to deduce the covering number. It follows that

$$\log\mathcal{N}_\epsilon(\widehat{\mathcal{V}}(\beta_b, 2K\beta_Q, \beta_Q, 11\eta H^3 K)) \leq 3(K^L + 2)^2 d^2 \log\left((K^L + 1)\frac{8|\mathcal{A}|}{\theta}(1 + \frac{2C_1 C_2}{\epsilon})\right).$$

Since we assume $1 \leq \beta_w \leq K$, $\beta_Q \leq 2H$, $\eta, \alpha \leq 1$, and $K^L \leq K$, we have the following bounds on $C_1$, $C_2$ with $n = K^L$.

$$\begin{aligned}
C_1 &= 33\sqrt{(K^L + 1)K^3}\beta_w \max\{2K\beta_Q, \beta_b\}\beta_Q(1 + 11\eta H^3 K) \\
&\leq 4481 H^5 K^6, \\
C_2 &= (K^L + 2)(1 + 11\eta H^3 K)K(\beta_b + 2K\beta_Q) + (K^L + 2)\sqrt{d} \\
&\leq 242\sqrt{d}H^4 K^4.
\end{aligned}$$

By Lemma 30, we can take $\epsilon = \sqrt{d}/(2K)$, and thus the covering number is bounded as

$$\log\mathcal{N}_\epsilon(\widehat{\mathcal{V}}(\beta_b, 2K\beta_Q, \beta_Q, 11\eta H^3 K)) \leq 36(K^L + 2)^2 d^2 \log(5HK|\mathcal{A}|/\theta).$$

Applying this to (18), since $\theta = K^{-1}$,

$$\begin{aligned}
\|(\psi_h - \widehat{\psi}_h^k)\widehat{V}\|_{\Lambda_h^k} &\leq 8H\sqrt{d\log(K+1) + 2\log(3H^2/\delta) + 36(K^L + 2)^2 d^2 \log(5HK|\mathcal{A}|/\theta)} \\
&\leq 50(K^L + 1)dH\sqrt{\log(5H^2 K^2 |\mathcal{A}|/\delta)} \\
&:= \beta_p.
\end{aligned}$$

Thus, we showed that $\Pr[\bar{E}_2] \geq 1 - \delta/3$ holds. For $E_3$, note that for any $(s, a) \in \mathcal{S} \times \mathcal{A}$,

$$\sum_{h \in [H]} 3\beta_b \|\phi(s, a)\|_{(\Lambda_h^k)^{-1}} + 8\beta_Q \beta_w^2 \|\phi(s, a)\|_{(\Lambda_h^k)^{-1}}^2 \leq H\left(3\beta_b + 8\beta_Q \beta_w^2\right).$$

Furthermore, $s_h^k, a_h^k$ are generated under $\mathbb{P}, \widehat{\pi}^k$. Then, by Lemma 28 with probability at least $1 - \delta/3$,

$$\sum_{k \in [K]} \mathbb{E}_{\mathbb{P},\widehat{\pi}^k}[W_k] \leq 2\sum_{k \in [K]} W_k + 4H(3\beta_b + 8\beta_Q \beta_w^2)\log\frac{6K}{\delta}.$$

Consequently, by union bound, we have $\Pr[\bar{E}_g] \geq 1 - \delta$. $\qquad\square$

Before proving that $E_g$ holds with high probability, we show the following lemma, which is a modification of Lemma 12 of Cassel & Rosenberg (2024) to our CMDP setting. This lemma plays a crucial role in establishing the connection between $\bar{E}_g$ and $E_g$.

**Lemma 15.** *Suppose that $\bar{E}_g$ holds. Given $k \in [K]$, if $\widehat{\pi}_h^k \in \widehat{\Pi}(K\beta_b, 2K^2\beta_Q, K\beta_{Q,h}, 11\eta H^3 K)$ for all $h \in [H]$, then $\widehat{Q}_{f,h}^k, \widehat{Q}_{g,h}^k \in \widehat{\mathcal{Q}}(\beta_b, 2K\beta_Q, \beta_{Q,h})$, and $\widehat{V}_{f,h}^k, \widehat{V}_{g,h}^k \in \widehat{\mathcal{V}}(\beta_b, 2K\beta_Q, \beta_{Q,h}, 11\eta H^3 K)$ for all $h \in [H+1]$.*

*Proof.* To show the statement, we apply induction on $h$ for fixed $k$. For the base case, consider $h = H + 1$. As we initialize as $\widehat{Q}_{f,H+1}^k(s,a) = \widehat{Q}_{g,H+1}^k(s,a) = \widehat{V}_{f,H+1}^k(s) = \widehat{V}_{g,H+1}^k(s) = 0$ for all $(s,a)$, it is clear that $\widehat{Q}_{f,H+1}^k, \widehat{Q}_{g,H+1}^k \in \widehat{\mathcal{Q}}(\beta_b, 2K\beta_Q, \beta_{Q,H+1})$ and $\widehat{V}_{f,H+1}^k, \widehat{V}_{g,H+1}^k \in \widehat{\mathcal{V}}(\beta_b, 2K\beta_Q, \beta_{Q,H+1}, 11\eta H^3 K)$. Next, we assume that the statement is true for $h+1$, i.e., $\widehat{Q}_{f,h+1}^k, \widehat{Q}_{g,h+1}^k \in \widehat{\mathcal{Q}}(\beta_b, 2K\beta_Q, \beta_{Q,h+1})$, $\widehat{V}_{f,h+1}^k, \widehat{V}_{g,h+1}^k \in \widehat{\mathcal{V}}(\beta_b, 2K\beta_Q, \beta_{Q,h+1}, 11\eta H^3 K)$. It follows that for any $\ell \in \{f, g\}$,

$$
\begin{aligned}
|\widehat{Q}_{\ell,h}^k(s,a)| &= \left| \bar{\phi}_h^{k_e}(s,a)^\top \left( \widehat{\theta}_{\ell,h}^k + \widehat{\psi}_h^k \widehat{V}_{\ell,h+1}^k \right) - \beta_b \left\| \bar{\phi}_h^{k_e}(s,a) \right\|_{(\Lambda_h^{k_e})^{-1}} \right| \\
&\leq \left| \bar{\phi}_h^{k_e}(s,a)^\top \left( \theta_{\ell,h}^k + \psi_h \widehat{V}_{\ell,h+1}^k \right) \right| \\
&\quad + \left( \beta_b + \left\| \widehat{\theta}_{\ell,h}^k - \theta_{\ell,h}^k \right\|_{\Lambda_h^{k_e}} + \left\| (\widehat{\psi}_h^k - \psi_h) \widehat{V}_{\ell,h+1}^k \right\|_{\Lambda_h^{k_e}} \right) \left\| \bar{\phi}_h^{k_e}(s,a) \right\|_{(\Lambda_h^{k_e})^{-1}} \\
&\leq \left| \bar{\phi}_h^{k_e}(s,a)^\top \left( \theta_{\ell,h}^k + \psi_h \widehat{V}_{\ell,h+1}^k \right) \right| \\
&\quad + \left( \beta_b + \left\| \widehat{\theta}_{\ell,h}^k - \theta_{\ell,h}^k \right\|_{\Lambda_h^k} + \left\| (\widehat{\psi}_h^k - \psi_h) \widehat{V}_{\ell,h+1}^k \right\|_{\Lambda_h^k} \right) \left\| \bar{\phi}_h^{k_e}(s,a) \right\|_{(\Lambda_h^{k_e})^{-1}}
\end{aligned}
$$

where the first inequality is due to the triangle inequality and the Cauchy-Schwarz inequality, and the second inequality is due to the fact that $\Lambda_h^{k_e} \preceq \Lambda_h^k$.

We bound each term individually. For the first term, for all $\ell \in \{f, g\}$,

$$
\begin{aligned}
\left| \bar{\phi}_h^{k_e}(s,a)^\top \left( \theta_{\ell,h}^k + \psi_h \widehat{V}_{\ell,h+1}^k \right) \right| &= \sigma \left( -\beta_w \|\phi(s,a)\|_{(\Lambda_h^{k_e})^{-1}} + \log K \right) \left| \phi(s,a)^\top (\theta_{\ell,h}^k + \psi_h \widehat{V}_{\ell,h+1}^k) \right| \\
&\leq \left| \phi(s,a)^\top (\theta_{\ell,h}^k + \psi_h \widehat{V}_{\ell,h+1}^k) \right| \\
&= \left| \ell_h^k(s,a) + \sum_{s' \in \mathcal{S}} \mathbb{P}_h(s' \mid s, a) \widehat{V}_{\ell,h+1}^k(s') \right| \\
&\leq 1 + \|\widehat{V}_{\ell,h+1}^k\|_\infty
\end{aligned}
$$

where the first and second equality are due to the definition of $\bar{\phi}_h^{k_e}$ and linear MDPs, respectively. Next, we can bound the second term, since the proxy good event $\bar{E}_g$ is assumed.

$$
\left( \beta_b + \left\| \widehat{\theta}_{\ell,h}^k - \theta_{\ell,h}^k \right\|_{\Lambda_h^k} + \left\| (\widehat{\psi}_h^k - \psi_h) \widehat{V}_{\ell,h+1}^k \right\|_{\Lambda_h^k} \right) \left\| \bar{\phi}_h^{k_e}(s,a) \right\|_{(\Lambda_h^{k_e})^{-1}} \leq (\beta_b + \beta_r + \beta_p) \left\| \bar{\phi}_h^{k_e}(s,a) \right\|_{(\Lambda_h^{k_e})^{-1}}.
$$

Recall that $\|\bar{\phi}_h^{k_e}(s,a)\|_{(\Lambda_h^{k_e})^{-1}} = \|\phi(s,a)\|_{(\Lambda_h^{k_e})^{-1}} \sigma(-\beta_w \|\phi(s,a)\|_{(\Lambda_h^{k_e})^{-1}} + \log K) \leq \max_{y \geq 0} y \cdot \sigma(-\beta_w y + \log K)$. It follows that

$$
\begin{aligned}
|\widehat{Q}_{\ell,h}^k(s,a)| &\leq 1 + \|\widehat{V}_{\ell,h+1}^k\|_\infty + (\beta_b + \beta_r + \beta_p) \left\| \bar{\phi}_h^{k_e}(s,a) \right\|_{(\Lambda_h^{k_e})^{-1}} \\
&\leq 1 + \|\widehat{V}_{\ell,h+1}^k\|_\infty + (\beta_b + \beta_r + \beta_p) \max_{y \geq 0} \left[ y \cdot \sigma(-\beta_w y + \log K) \right] \\
&\leq 1 + \|\widehat{V}_{\ell,h+1}^k\|_\infty + \frac{2 \log K}{\beta_w} (\beta_r + \beta_p + \beta_b) \\
&= 2 + \|\widehat{V}_{\ell,h+1}^k\|_\infty \\
&\leq 2 + \beta_{Q,h+1} \\
&= \beta_{Q,h}
\end{aligned}
$$

where the third inequality follows from Lemma 27, the equality is due to $\beta_w = 2(\beta_r + \beta_p + \beta_b) \log K$, and the last inequality holds because of the induction hypothesis.

So far, we have shown that $\|\widehat{Q}_{\ell,h}^k\|_\infty \leq \beta_{Q,h}$. To show $\widehat{Q}_{\ell,h}^k \in \widehat{\mathcal{Q}}(\beta_b, 2K\beta_Q, \beta_{Q,h})$, it remains to show that the corresponding parameters are upper bounded. Recall that $\widehat{Q}_{\ell,h}^k$ is defined as

$$\widehat{Q}_{\ell,h}^k(s,a) = \bar{\phi}_h^{k_e}(s,a)^\top w_{\ell,h}^k - \beta_b \left\| \bar{\phi}_h^{k_e}(s,a) \right\|_{(\Lambda_h^{k_e})^{-1}}$$

where $w_{\ell,h}^k = \widehat{\theta}_{\ell,h}^k + \widehat{\psi}_h^k \widehat{V}_{\ell,h+1}^k$. Note that

$$\|\widehat{\theta}_{\ell,h}^k\|_2 \leq \left\| (\Lambda_h^k)^{-1} \sum_{\tau \in [k-1]} \phi(s_h^\tau, a_h^\tau) \ell_h^\tau(s_h^\tau, a_h^\tau) \right\|_2 \leq \left\| (\Lambda_h^k)^{-1} \right\|_2 \left\| \sum_{\tau \in [k-1]} \phi(s_h^\tau, a_h^\tau) \ell_h^\tau(s_h^\tau, a_h^\tau) \right\|_2 \leq K.$$

Furthermore, the induction hypothesis implies that

$$\left\| \widehat{\psi}_h^k \widehat{V}_{\ell,h+1}^k \right\|_2 = \left\| (\Lambda_h^k)^{-1} \sum_{\tau \in [k-1]} \phi(s_h^\tau, a_h^\tau) \widehat{V}_{\ell,h+1}^k(s_{h+1}^\tau) \right\|_2 \leq \beta_{Q,h+1} K.$$

It follows that

$$\|w_{\ell,h}^k\|_2 \leq \|\widehat{\theta}_{\ell,h}^k\|_2 + \left\| \widehat{\psi}_h^k \widehat{V}_{\ell,h+1}^k \right\|_2 \leq 2\beta_Q K.$$

Furthermore, we have $(2K)^{-1} I \preceq (\Lambda_h^{k_e})^{-1} \preceq I$. Thus, we have

$$\widehat{Q}_{\ell,h}^k \in \widehat{\mathcal{Q}}(\beta_b, 2K\beta_Q, \beta_{Q,h}).$$

By definition, since we have $\widehat{V}_{\ell,h}^k(s) = \sum_{a \in \mathcal{A}} \widehat{\pi}_h^k(a \mid s) \widehat{Q}_{\ell,h}^k(s,a)$ and the assumption $\widehat{\pi}_h^k \in \widehat{\Pi}(K\beta_b, 2K^2\beta_Q, K\beta_{Q,h}, 11\eta H^3 K)$, it follows that

$$\widehat{V}_{\ell,h}^k \in \widehat{\mathcal{V}}(\beta_b, 2K\beta_Q, \beta_{Q,h}, 11\eta H^3 K).$$

This completes the proof. $\qquad\square$

Finally, we prove that $E_g$ holds with high probability. The proof closely follows Lemma 6 of Cassel & Rosenberg (2024), with modifications for the CMDP setting.

**Lemma 16** (Restatement of Lemma 1). *Let* $1 \leq \beta_w \leq K$, *let* $\eta, \alpha \leq 1$ *and* $4\alpha\eta H^3 \leq 1$. *Then* $\Pr[E_g] \geq 1 - \delta$ *for any* $\delta \in (0,1)$.

*Proof.* We assume $\bar{E}_g$, which holds with probability at least $1 - \delta$ by Lemma 14. Next, under $\bar{E}_g$, we focus on showing $E_2$. As a first step, we show that $\widehat{\pi}_h^k \in \widehat{\Pi}(K\beta_b, 2K^2\beta_Q, K\beta_{Q,h}, 11\eta H^3 K)$ and $Y_k \in [0, 11\eta H^3 k]$ for all $k, h$ using induction on $k \in K_e$ for each epoch $e \in E$. Finally, based on this induction, we prove that $E_2$ holds.

**Step 1: Base Case** First, let us fix $e \in E$. For the base case, consider $k = k_e$. Since $\widehat{\pi}_h^{k_e} = \pi_{\text{unif}}$ for all $h$, it follows that $\widehat{\pi}_h^{k_e} \in \widehat{\Pi}(K\beta_b, 2K^2\beta_Q, K\beta_{Q,h}, 11\eta H^3 K)$, as $\pi_{\text{unif}}$ can be viewed as $\pi(a \mid s; 0, 0, I)$ with $n = 0$. Furthermore, we initialize $Y_{k_e} = 0$. Thus, the base case holds.

**Step 2: Induction Hypothesis** For $k \in K_e$, we assume that $\widehat{\pi}_h^{k'} \in \widehat{\Pi}(K\beta_b, 2K^2\beta_Q, K\beta_{Q,h}, 11\eta H^3 K)$ and $Y_{k'} \in [0, 11\eta H^3 k']$ for all $h$ and $k_e \leq k' < k$. Then, by Lemma 15, it follows that for all $(h, k', \ell) \in [H] \times \{k_e, \ldots, k-1\} \times \{f, g\}$,

$$\widehat{Q}_{\ell,h}^{k'} \in \widehat{\mathcal{Q}}(\beta_b, 2K\beta_Q, \beta_{Q,h}), \quad \widehat{V}_{\ell,h}^{k'} \in \widehat{\mathcal{V}}(\beta_b, 2K\beta_Q, \beta_{Q,h}, 11\eta H^3 K). \tag{19}$$

Furthermore, let $\beta_b, w_{\ell,h}^{k'}, \Lambda$ denote the parameters that specify $\widehat{Q}_{\ell,h}^{k'}$, i.e., for all $h, k' < k, \ell \in \{f, g\}, \widehat{Q}_{\ell,h}^{k'}(\cdot, \cdot) = \widehat{Q}(\cdot, \cdot; \beta_b, w_{\ell,h}^{k'}, \Lambda)$.

**Step 3-1: Induction Step** $(\widehat{\pi}_h^k)$   Next, we show that $\widehat{\pi}_h^k \in \widehat{\Pi}(K\beta_b, 2K^2\beta_Q, K\beta_{Q,h}, 11\eta H^3 K)$ for all $h$. By Lemma 9,

$$\widehat{\pi}_h^k(\cdot \mid \cdot) = \widehat{\pi}(\cdot \mid \cdot; \{\beta_i, w_i\}_{i=0}^n, \Lambda)$$

where $w_i = -\alpha \sum_{j \in S_i} (w_{f,h}^j + Y_j w_{g,h}^j)$, $\beta_i = -\alpha\beta_b \sum_{j \in S_i}(1 + Y_j)$. Note that $j \in S_i$ satisfies $j < k$ for each $i$, thus we can use (19) to bound the parameters. For $w_i$,

$$
\begin{aligned}
\|w_i\|_2 &\le \sum_{j \in S_i} \|w_{f,h}^j + Y_j w_{g,h}^j\|_2 \\
&\le \sum_{j \in S_i} \|w_{f,h}^j\|_2 + Y_j\|w_{g,h}^j\|_2 \\
&\le \sum_{j \in S_i}(1 + 11\eta H^3 K)2K\beta_Q \\
&\le (1 + 11\eta H^3 K)2K^2\beta_Q
\end{aligned}
\tag{20}
$$

where the first inequality is due to $|\alpha| \le 1$, and the third inequality is due to the induction hypothesis ($Y_{k'} < 11\eta H^3 k'$ for all $k' < k$) and that (19) implies $\|w_{\ell,h}^j\|_2 \le 2K\beta_Q$. Similarly,

$$|\beta_i| \le (1 + 11\eta H^3 K)K\beta_b. \tag{21}$$

Again, by (19), we have $\|\widehat{Q}_{f,h}^j\|_\infty, \|\widehat{Q}_{g,h}^j\|_\infty \le \beta_Q$. Then, for any $(s,a) \in \mathcal{S} \times \mathcal{A}$ and $i = 0, \ldots, n$,

$$\left| -\alpha \sum_{j \in S_i} (\widehat{Q}_{f,h}^j(s,a) + Y_j \widehat{Q}_{g,h}^j(s,a)) \right| \le (1 + 11\eta H^3 K)K\beta_{Q,h}. \tag{22}$$

Note that $n = \max\{0, \lfloor (k-1-k_e)/K^B \rfloor\} \le \lfloor K/K^B \rfloor \le K^{1-B} = K^L$. Furthermore, its parameters are bounded by (20), (21), and (22), and the same argument can be applied for all $h \in [H]$. Thus, for all $h \in [H]$,

$$\widehat{\pi}_h^k \in \widehat{\Pi}(K\beta_b, 2K^2\beta_Q, K\beta_{Q,h}, 11\eta H^3 K).$$

**Step 3-2: Induction Step** $(Y_k)$   To bound $Y_k$,

$$
\begin{aligned}
Y_k &= \left[ (1 - 4\alpha\eta H^3)Y_{k-1} + \eta\left(\widehat{V}_{g,1}^{k-1}(s_1) - b - 4\alpha H^3 - 4\theta H^2\right)\right]_+ \\
&\le \left| (1 - 4\alpha\eta H^3)Y_{k-1} + \eta\left(\widehat{V}_{g,1}^{k-1}(s_1) - b - 4\alpha H^3 - 4\theta H^2\right)\right| \\
&\le (1 - 4\alpha\eta H^3)|Y_{k-1}| + \eta|\widehat{V}_{g,1}^{k-1}(s_1) - b - 4\alpha H^3 - 4\theta H^2| \\
&\le |Y_{k-1}| + \eta|\widehat{V}_{g,1}^{k-1}(s_1) - b - 4\alpha H^3 - 4\theta H^2| \\
&\le 11\eta H^3(k-1) + 11\eta H^3 \\
&\le 11\eta H^3 k
\end{aligned}
$$

where the first inequality is due to the fact that $\max\{0, z\} \le |z|$ for all $z \in \mathbb{R}$, the second and third inequality follows from the triangle inequality and $0 \le 1 - 4\alpha\eta H^3 \le 1$, and the fourth inequality is due the induction hypothesis, i.e., $Y_{k'} \le 11\eta H^3 k'$ and $\|\widehat{V}_{g,1}^{k-1}\|_\infty \le 2H$ for all $k' < k$.

These complete the induction, i.e., $\widehat{\pi}_h^k \in \widehat{\Pi}(K\beta_b, 2K^2\beta_Q, K\beta_{Q,h}, 11\eta H^3 K)$ and $Y_k \in [0, 11\eta H^3 k]$ for all $(h,k) \in [H] \times K_e$. Furthermore, we can apply the same argument for all $e \in E$. Thus, it holds for all $(h,k) \in [H] \times [K]$.

**Step 4: Showing $E_g$**   By Lemma 15, we have for all $h, k$,

$$\widehat{Q}_{f,h}^k, \widehat{Q}_{g,h}^k \in \widehat{\mathcal{Q}}(\beta_b, 2K\beta_Q, \beta_{Q,h}), \quad \widehat{V}_{f,h}^k, \widehat{V}_{g,h}^k \in \widehat{\mathcal{V}}(\beta_b, 2K\beta_Q, \beta_{Q,h}, 11\eta H^3 K).$$

As a result, since $\bar{E}_2$ is assumed, we have $\|(\psi_h - \widehat{\psi}_h^k)\widehat{V}_{\ell,h+1}^k\|_{\Lambda_h^k} \le \beta_p$. Thus, $E_2$ holds. Furthermore, $E_1, E_3$ hold by $\bar{E}_g$. This completes the proof. $\qquad\square$

## I  LYAPUNOV DRIFT ANALYSIS

In this section, we upper bound the dual variable based on a Lyapunov drift analysis. As a first step, we bound the Lyapunov drift $(Y_{k+1}^2 - Y_k^2)/2$.

**Lemma 17.** *Assume that the good event $E_g$ holds. For all $e \in E$ and $k, k+1 \in K_e$, the Lyapunov drift is bounded as*

$$
\frac{Y_{k+1}^2 - Y_k^2}{2} \leq -\eta\gamma Y_k + \frac{\eta}{\alpha}\mathbb{E}_{\mathbb{P}^{k_e},\bar{\pi}}\left[\sum_{h \in [H]} D(\bar{\pi}_h(\cdot|s_h)||\widetilde{\pi}_h^k(\cdot|s_h)) - D(\bar{\pi}_h(\cdot|s_h)||\widehat{\pi}_h^{k+1}(\cdot|s_h))\right]
$$
$$
+ \eta(2\alpha H^3 + 4H^2\theta + 4H^2) + 2\eta^2(9H^2 + 16\alpha^2 H^6 + 1936\alpha^2\eta^2 H^{12}K^2 + 16\theta^2 H^4).
$$

*Proof.* Recall that the dual variable follows $Y_{k+1} = [(1 - 4\alpha\eta H^3)Y_k + \eta(\widehat{V}_{g,1}^k(s_1) - b - 4\alpha H^3 - 4\theta H^2)]_+$. It can be rewritten as

$$
Y_{k+1} = \left[Y_k + \eta\left(\widehat{V}_{g,1}^k(s_1) - b - 4\alpha H^3(1 + Y_k) - 4\theta H^2\right)\right]_+.
$$

Note that $\max\{0, z\}^2 \leq z^2$ for any $z \in \mathbb{R}$. Then, if we square both sides, we have

$$
Y_{k+1}^2 \leq Y_k^2 + 2Y_k\eta\left(\widehat{V}_{g,1}^k(s_1) - b - 4\alpha H^3(1 + Y_k) - 4\theta H^2\right) + \eta^2\left(\widehat{V}_{g,1}^k(s_1) - b - 4\alpha H^3(1 + Y_k) - 4\theta H^2\right)^2.
$$

It can be rewritten as

$$
\frac{Y_{k+1}^2 - Y_k^2}{2}
$$
$$
\leq \underbrace{Y_k\eta\left(\widehat{V}_{g,1}^k(s_1) - b - 4\alpha H^3(1 + Y_k) - 4\theta H^2\right)}_{\text{(I)}} + \underbrace{\frac{\eta^2}{2}\left(\widehat{V}_{g,1}^k(s_1) - b - 4\alpha H^3(1 + Y_k) - 4\theta H^2\right)^2}_{\text{(II)}}.
$$
$$
\tag{23}
$$

(II) can be bounded as follows.

$$
\text{(II)} \leq \frac{\eta^2}{2} \cdot 4((\widehat{V}_{g,1}^k(s_1) - b)^2 + (4\alpha H^3)^2 + (4\alpha H^3 Y_k)^2 + (4\theta H^2)^2)
$$
$$
\leq \frac{\eta^2}{2} \cdot 4(9H^2 + 16\alpha^2 H^6 + 16\alpha^2 H^6 Y_k^2 + 16\theta^2 H^4)
$$
$$
\leq \frac{\eta^2}{2} \cdot 4(9H^2 + 16\alpha^2 H^6 + 1936\alpha^2\eta^2 H^{12}K^2 + 16\theta^2 H^4)
$$

where the first inequality follows from the Cauchy-Schwarz inequality, the second and third inequalities follow from that $E_g$ implies $|\widehat{V}_{g,1}^k(s_1)| \leq 2H$ and $0 \leq Y_k \leq 11\eta H^3 k$ for all $k$.

Next, we bound (I). To obtain a negative drift, we first deduce a bound on $Y_k\langle\widehat{\pi}_h^{k+1}(\cdot|s_h) - \widehat{\pi}_h^k(\cdot|s_h), \widehat{Q}_{g,h}^k(s_h, \cdot)\rangle$. Since we assumed $k, k+1 \in K_e$, $\widehat{\pi}_h^{k+1} \neq \pi_{\text{unif}}$. Then $\widehat{\pi}_h^{k+1}$ satisfies

$$
\widehat{\pi}_h^{k+1}(\cdot \mid s) = \underset{\pi(\cdot|s) \in \Delta(\mathcal{A})}{\arg\min} \ \langle\pi, \widehat{Q}_{f,h}^k + Y_k\widehat{Q}_{g,h}^k\rangle + \frac{1}{\alpha}D(\pi||\widetilde{\pi}_h^k).
$$

Applying Lemma 23 and letting $z = \bar{\pi}_h$, we have for any $s_h \in \mathcal{S}$,

$$
\left\langle\widehat{\pi}_h^{k+1}(\cdot|s_h), \widehat{Q}_{f,h}^k(s_h, \cdot) + Y_k\widehat{Q}_{g,h}^k(s_h, \cdot)\right\rangle + \frac{1}{\alpha}D(\widehat{\pi}_h^{k+1}(\cdot|s_h)||\widetilde{\pi}_h^k(\cdot|s_h))
$$
$$
\leq \left\langle\bar{\pi}_h(\cdot|s_h), \widehat{Q}_{f,h}^k(s_h, \cdot) + Y_k\widehat{Q}_{g,h}^k(s_h, \cdot)\right\rangle + \frac{1}{\alpha}D(\bar{\pi}_h(\cdot|s_h)||\widetilde{\pi}_h^k(\cdot|s_h)) - \frac{1}{\alpha}D(\bar{\pi}_h(\cdot|s_h)||\widehat{\pi}_h^{k+1}(\cdot|s_h))
$$

where $\bar{\pi}$ is the Slater policy satisfying $V_{g,1}^{\bar{\pi}}(s_1) \leq b - \gamma$ for some $\gamma > 0$. Next, summing over $h$ and rearranging terms yield

$$
\begin{aligned}
& Y_k \sum_{h \in [H]} \langle \widehat{\pi}_h^{k+1}(\cdot|s_h) - \widehat{\pi}_h^k(\cdot|s_h), \widehat{Q}_{g,h}^k(s_h, \cdot) \rangle \\
& \leq \frac{1}{\alpha} \sum_{h \in [H]} D(\bar{\pi}_h(\cdot|s_h) || \widetilde{\pi}_h^k(\cdot|s_h)) - \frac{1}{\alpha} \sum_{h \in [H]} D(\bar{\pi}_h(\cdot|s_h) || \widehat{\pi}_h^{k+1}(\cdot|s_h)) \\
& \quad + \sum_{h \in [H]} \langle \widehat{\pi}_h^k(\cdot|s_h) - \widehat{\pi}_h^{k+1}(\cdot|s_h), \widehat{Q}_{f,h}^k(s_h, \cdot) \rangle - \frac{1}{\alpha} \sum_{h \in [H]} D(\widehat{\pi}_h^{k+1}(\cdot|s_h) || \widetilde{\pi}_h^k(\cdot|s_h)) \\
& \quad + \sum_{h \in [H]} \langle \bar{\pi}_h(\cdot|s_h) - \widehat{\pi}_h^k(\cdot|s_h), \widehat{Q}_{f,h}^k(s_h, \cdot) \rangle + Y_k \sum_{h \in [H]} \langle \bar{\pi}_h(\cdot|s_h) - \widehat{\pi}_h^k(\cdot|s_h), \widehat{Q}_{g,h}^k(s_h, \cdot) \rangle.
\end{aligned}
\tag{24}
$$

Now, we take $\mathbb{E}_{\bar{\mathbb{P}}^{k_e}, \bar{\pi}}$, which is taken over $\{s_h\}_{h=1}^H$ under $\bar{\mathbb{P}}^{k_e}, \bar{\pi}$ for a fixed $s_1$. Note that since $\bar{\mathbb{P}}^{k_e}$ is a transition kernel of a contracted MDP, it could be $\sum_{s'} \bar{\mathbb{P}}^{k_e}(s'|s, a) \leq 1$. However, taking $\mathbb{E}_{\bar{\mathbb{P}}^{k_e}, \bar{\pi}}$ can be viewed as a linear combination, where its coefficients is in the form of a sub-probability measure defined as $\Pr[s_1 = s, \ldots, s_H = s' \mid s_1, \bar{\mathbb{P}}^{k_e}, \bar{\pi}] \in [0, 1]$. This implies that taking $\mathbb{E}_{\bar{\mathbb{P}}^{k_e}, \bar{\pi}}$ guarantees monotonicity, i.e.,

$$
\begin{aligned}
& Y_k \mathbb{E}_{\bar{\mathbb{P}}^{k_e}, \bar{\pi}} \left[ \sum_{h \in [H]} \langle \widehat{\pi}_h^{k+1}(\cdot|s_h) - \widehat{\pi}_h^k(\cdot|s_h), \widehat{Q}_{g,h}^k(s_h, \cdot) \rangle \right] \\
& \leq \frac{1}{\alpha} \mathbb{E}_{\bar{\mathbb{P}}^{k_e}, \bar{\pi}} \left[ \sum_{h \in [H]} D(\bar{\pi}_h(\cdot|s_h) || \widetilde{\pi}_h^k(\cdot|s_h)) - D(\bar{\pi}_h(\cdot|s_h) || \widehat{\pi}_h^{k+1}(\cdot|s_h)) \right] \\
& \quad + \underbrace{\mathbb{E}_{\bar{\mathbb{P}}^{k_e}, \bar{\pi}} \left[ \sum_{h \in [H]} \langle \widehat{\pi}_h^k(\cdot|s_h) - \widehat{\pi}_h^{k+1}(\cdot|s_h), \widehat{Q}_{f,h}^k(s_h, \cdot) \rangle - \frac{1}{\alpha} D(\widehat{\pi}_h^{k+1}(\cdot|s_h) || \widetilde{\pi}_h^k(\cdot|s_h)) \right]}_{\text{(III)}} \\
& \quad + \underbrace{\mathbb{E}_{\bar{\mathbb{P}}^{k_e}, \bar{\pi}} \left[ \sum_{h \in [H]} \langle \bar{\pi}_h(\cdot|s_h) - \widehat{\pi}_h^k(\cdot|s_h), \widehat{Q}_{f,h}^k(s_h, \cdot) \rangle \right]}_{\text{(IV)}} \\
& \quad + \underbrace{Y_k \mathbb{E}_{\bar{\mathbb{P}}^{k_e}, \bar{\pi}} \left[ \sum_{h \in [H]} \langle \bar{\pi}_h(\cdot|s_h) - \widehat{\pi}_h^k(\cdot|s_h), \widehat{Q}_{g,h}^k(s_h, \cdot) \rangle \right]}_{\text{(V)}}.
\end{aligned}
\tag{25}
$$

We bound (III). By the third statement of Lemma 22, for any $s_h \in \mathcal{S}$, we have

$$
\langle \widehat{\pi}_h^k(\cdot|s_h) - \widehat{\pi}_h^{k+1}(\cdot|s_h), \widehat{Q}_{f,h}^k(s_h, \cdot) \rangle - \frac{1}{\alpha} D(\widehat{\pi}_h^{k+1}(\cdot|s_h) || \widetilde{\pi}_h^k(\cdot|s_h)) \leq 2\alpha H^2 + 4H\theta.
$$

Thus, summing over $h \in [H]$ and taking $\mathbb{E}_{\bar{\mathbb{P}}^{k_e}, \bar{\pi}}$, we have

$$
\text{(III)} \leq \mathbb{E}_{\bar{\mathbb{P}}^{k_e}, \bar{\pi}} \left[ 2\alpha H^3 + 4H^2\theta \right] \leq 2\alpha H^3 + 4H^2\theta
$$

where the second inequality is due to (5). To bound (IV),

$$
\begin{aligned}
\text{(IV)} & \leq \mathbb{E}_{\bar{\mathbb{P}}^{k_e}, \bar{\pi}} \left[ \sum_{h \in [H]} \|\bar{\pi}_h(\cdot|s_h) - \widehat{\pi}_h^k(\cdot|s_h)\|_1 \|\widehat{Q}_{f,h}^k(s_h, \cdot)\|_\infty \right] \\
& \leq \mathbb{E}_{\bar{\mathbb{P}}^{k_e}, \bar{\pi}} \left[ 4H^2 \right] \\
& \leq 4H^2
\end{aligned}
$$

where the first inequality is due to Hölder's inequality, and the last inequality is due to (5).

To bound (V), we observe the following. Let $\bar{V}_{g,1}^{\bar{\pi}}(s)$ denote the $\rho$-contracted value function, where $\rho(s,a,h) = \sigma(-\beta_w \|\phi(s,a)\|_{(\Lambda_h^{k_e})^{-1}} + \log K)$. By Lemmas 7 and 8, we have

$$V_{g,1}^{\bar{\pi}}(s_1) - \widehat{V}_{g,1}^k(s_1) \geq \bar{V}_{g,1}^{\bar{\pi}}(s_1) - \widehat{V}_{g,1}^k(s_1)$$

$$= \mathbb{E}_{\bar{\mathbb{P}}^{k_e}, \bar{\pi}}\left[\sum_{h \in [H]} \langle \bar{\pi}_h(\cdot \mid s_h) - \widehat{\pi}_h^k(\cdot \mid s_h), \widehat{Q}_{g,h}^k(\cdot \mid s_h)\rangle\right]$$

$$+ \mathbb{E}_{\bar{\mathbb{P}}^{k_e}, \bar{\pi}}\left[\sum_{h \in [H]} \bar{g}_h(s_h, a_h) + \sum_{s' \in \mathcal{S}} \bar{\mathbb{P}}_h^{k_e}(s' \mid s_h, a_h)\widehat{V}_{g,h+1}^k(s') - \widehat{Q}_{g,h}^k(s_h, a_h)\right].$$

To bound the latter term, we have for any $(s_h, a_h) \in \mathcal{S} \times \mathcal{A}$,

$$\bar{g}_h(s_h, a_h) + \sum_{s' \in \mathcal{S}} \bar{\mathbb{P}}_h(s' \mid s_h, a_h)\widehat{V}_{g,h+1}^k(s') - \widehat{Q}_{g,h}^k(s_h, a_h)$$

$$= \bar{\phi}_h^{k_e}(s_h, a_h)^\top \theta_{g,h} + \bar{\phi}_h^{k_e}(s_h, a_h)^\top \psi_h \widehat{V}_{g,h+1}^k - \bar{\phi}_h^{k_e}(s_h, a_h)^\top \left[\widehat{\theta}_{g,h}^k + \widehat{\psi}_h^k \widehat{V}_{g,h+1}^k\right] + \beta_b\|\bar{\phi}_h^{k_e}(s_h, a_h)\|_{(\Lambda_h^{k_e})^{-1}}$$

$$\geq -\|\theta_{g,h} - \widehat{\theta}_{g,h}^k\|_{\Lambda_h^{k_e}}\|\bar{\phi}_h^{k_e}(s_h, a_h)\|_{(\Lambda_h^{k_e})^{-1}} - \|(\psi_h - \widehat{\psi}_h^k)\widehat{V}_{g,h+1}^k\|_{\Lambda_h^{k_e}}\|\bar{\phi}_h^{k_e}(s_h, a_h)\|_{(\Lambda_h^{k_e})^{-1}} + \beta_b\|\bar{\phi}_h^{k_e}(s_h, a_h)\|_{(\Lambda_h^{k_e})^{-1}}$$

$$\geq -\|\theta_{g,h} - \widehat{\theta}_{g,h}^k\|_{\Lambda_h^k}\|\bar{\phi}_h^{k_e}(s_h, a_h)\|_{(\Lambda_h^{k_e})^{-1}} - \|(\psi_h - \widehat{\psi}_h^k)\widehat{V}_{g,h+1}^k\|_{\Lambda_h^k}\|\bar{\phi}_h^{k_e}(s_h, a_h)\|_{(\Lambda_h^{k_e})^{-1}} + \beta_b\|\bar{\phi}_h^{k_e}(s_h, a_h)\|_{(\Lambda_h^{k_e})^{-1}}$$

$$\geq -\beta_r\|\bar{\phi}_h^{k_e}(s_h, a_h)\|_{(\Lambda_h^{k_e})^{-1}} - \beta_p\|\bar{\phi}_h^{k_e}(s_h, a_h)\|_{(\Lambda_h^{k_e})^{-1}} + \beta_b\|\bar{\phi}_h^{k_e}(s_h, a_h)\|_{(\Lambda_h^{k_e})^{-1}}$$

$$= 0$$

where the first equality is due to the definition of contracted MDP, the first inequality is due to the Cauchy-Schwarz inequality, the second inequality is due to $\Lambda_h^{k_e} \preceq \Lambda_h^k$, the third inequality is due to $E_g$, and the last equality is due to $\beta_b = \beta_r + \beta_p$. This implies that the latter term is nonnegative, as $\mathbb{E}_{\bar{\mathbb{P}}^{k_e}, \bar{\pi}}[0] = 0$. Thus, it follows that

$$\text{(V)} = Y_k \mathbb{E}_{\bar{\mathbb{P}}^{k_e}, \bar{\pi}}\left[\sum_{h \in [H]} \langle \bar{\pi}_h(\cdot \mid s_h) - \widehat{\pi}_h^k(\cdot \mid s_h), \widehat{Q}_{g,h}^k(\cdot \mid s_h)\rangle\right]$$

$$\leq Y_k(V_{g,1}^{\bar{\pi}}(s_1) - \widehat{V}_{g,1}^k(s_1))$$

$$\leq Y_k(b - \gamma - \widehat{V}_{g,1}^k(s_1))$$

where the last inequality is due to the Slater condition and $Y_k \geq 0$. Finally, plugging the bounds on (III),(IV), and (V) into (25), we have

$$Y_k \mathbb{E}_{\bar{\mathbb{P}}^{k_e}, \bar{\pi}}\left[\sum_{h \in [H]} \langle \widehat{\pi}_h^{k+1}(\cdot|s_h) - \widehat{\pi}_h^k(\cdot|s_h), \widehat{Q}_{g,h}^k(s_h, \cdot)\rangle\right]$$

$$\leq \frac{1}{\alpha}\mathbb{E}_{\bar{\mathbb{P}}^{k_e}, \bar{\pi}}\left[\sum_{h \in [H]} D(\bar{\pi}_h(\cdot|s_h)\|\widetilde{\pi}_h^k(\cdot|s_h)) - D(\bar{\pi}_h(\cdot|s_h)\|\widehat{\pi}_h^{k+1}(\cdot|s_h))\right]$$

$$+ 2\alpha H^3 + 4H^2\theta + 4H^2 + Y_k(b - \gamma - \widehat{V}_{g,1}^k(s_1)).$$

Then we can bound (I) as follows.

$$\text{(I)} = Y_k\eta\left(\widehat{V}_{g,1}^k(s_1) - b - 4\alpha H^3(1+Y_k) - 4\theta H^2\right)$$

$$\leq Y_k\eta\left(\widehat{V}_{g,1}^k(s_1) - b + \mathbb{E}_{\bar{\mathbb{P}}^{k_e},\bar{\pi}}\left[\sum_{h\in[H]}\langle\widehat{\pi}_h^{k+1}(\cdot|s_h) - \widehat{\pi}_h^k(\cdot|s_h), \widehat{Q}_{g,h}^k(s_h,\cdot)\rangle\right]\right)$$

$$\leq Y_k\eta(\widehat{V}_{g,1}^k(s_1) - b) + \frac{\eta}{\alpha}\mathbb{E}_{\bar{\mathbb{P}}^{k_e},\bar{\pi}}\left[\sum_{h\in[H]}D(\bar{\pi}_h(\cdot|s_h)||\widetilde{\pi}_h^k(\cdot|s_h)) - D(\bar{\pi}_h(\cdot|s_h)||\widehat{\pi}_h^{k+1}(\cdot|s_h))\right]$$

$$+ \eta(2\alpha H^3 + 4H^2\theta + 4H^2) + \eta Y_k(b - \gamma - \widehat{V}_{g,1}^k(s_1))$$

$$= -\eta\gamma Y_k + \frac{\eta}{\alpha}\mathbb{E}_{\bar{\mathbb{P}}^{k_e},\bar{\pi}}\left[\sum_{h\in[H]}D(\bar{\pi}_h(\cdot|s_h)||\widetilde{\pi}_h^k(\cdot|s_h)) - D(\bar{\pi}_h(\cdot|s_h)||\widehat{\pi}_h^{k+1}(\cdot|s_h))\right]$$

$$+ \eta(2\alpha H^3 + 4H^2\theta + 4H^2).$$

Here, the first inequality is true as follows. For any $s_1,\ldots,s_H \in \mathcal{S}$,

$$\left|\sum_{h\in[H]}\langle\widehat{\pi}_h^{k+1}(\cdot|s_h) - \widehat{\pi}_h^k(\cdot|s_h), \widehat{Q}_{g,h}^k(s_h,\cdot)\rangle\right| \leq \sum_{h\in[H]}\left|\langle\widehat{\pi}_h^{k+1}(\cdot|s_h) - \widehat{\pi}_h^k(\cdot|s_h), \widehat{Q}_{g,h}^k(s_h,\cdot)\rangle\right|$$

$$\leq \sum_{h\in[H]}(4\alpha H^2(1+Y_k) + 4\theta H)$$

$$= 4\alpha H^3(1+Y_k) + 4\theta H^2$$

where the first inequality is due to the triangle inequality, and the second inequality is due to the second statement of Lemma 22. Note that the second inequality holds regardless of whether $\widetilde{\pi}_h^k$ is perturbed, because when $\widetilde{\pi}_h^k$ is not perturbed, it can be viewed as $\theta = 0$. It follows that

$$\mathbb{E}_{\bar{\mathbb{P}}^{k_e},\bar{\pi}}\left[\sum_{h\in[H]}\langle\widehat{\pi}_h^{k+1}(\cdot|s_h) - \widehat{\pi}_h^k(\cdot|s_h), \widehat{Q}_{g,h}^k(s_h,\cdot)\rangle\right] \geq -\mathbb{E}_{\bar{\mathbb{P}}^{k_e},\bar{\pi}}[4\alpha H^3(1+Y_k) + 4\theta H^2]$$

$$\geq -(4\alpha H^3(1+Y_k) + 4\theta H^2)$$

where the second inequality is due to (5).

Consequently, plugging the bounds on (I) and (II) into (23), the Lyapunov drift is bounded as

$$\frac{Y_{k+1}^2 - Y_k^2}{2} \leq -\eta\gamma Y_k + \frac{\eta}{\alpha}\mathbb{E}_{\bar{\mathbb{P}}^{k_e},\bar{\pi}}\left[\sum_{h\in[H]}D(\bar{\pi}_h(\cdot|s_h)||\widetilde{\pi}_h^k(\cdot|s_h)) - D(\bar{\pi}_h(\cdot|s_h)||\widehat{\pi}_h^{k+1}(\cdot|s_h))\right]$$

$$+ \eta(2\alpha H^3 + 4H^2\theta + 4H^2) + 2\eta^2(9H^2 + 16\alpha^2H^6 + 1936\alpha^2\eta^2H^{12}K^2 + 16\theta^2H^4).$$

$$\square$$

**Lemma 18** (Restatement of Lemma 2). *Assume that the good event $E_g$ holds. Let $H^2 \leq K$. For $k \in [K]$, we have*

$$Y_k \leq \frac{2C_4}{K^B\eta\gamma} + 2K^B\delta_{\max}.$$

*where $\delta_{\max}$ and $C_4$ are given in (26) and (31), respectively. Furthermore, under the parameter choice of Algorithm 1, we have*

$$Y_k = \widetilde{\mathcal{O}}(H^2/\gamma).$$

*Proof.* For ease of notation, let $N_e = \max\{n \in \mathbb{Z}_+ : k_e + nK^B \in K_e\}$, and let

$$Z_n = Y_{k_e+nK^B}.$$

Fix $e \in E$. We first upper bound $Z_n$ for $0 \le n \le N_e$. Note that $|\max\{z_1, 0\} - z_2| \le |z_1 - z_2|$ for any $z_1 \in \mathbb{R}$ and $z_2 \in \mathbb{R}_+$. Then it follows that

$$
\begin{aligned}
|Y_{k+1} - Y_k| &\le \left| -4\alpha\eta H^3 Y_k + \eta(\widehat{V}_{g,1}^k(s_1) - b - 4\alpha H^3 - 4\theta H^2) \right| \\
&\le 4\eta\alpha H^3(11\eta H^3 K) + 3\eta H + 4\eta\alpha H^3 + 4\eta\theta H^2 \\
&:= \delta_{\max}.
\end{aligned}
\tag{26}
$$

where the second inequality is due to the triangle inequality and the fact that $Y_k \le 11\eta H^3 K$ and $\|\widehat{V}_{g,1}^k\|_\infty \le 2H$ under $E_g$. Thus, by the triangle inequality,

$$
|Z_{n+1} - Z_n| = \left| \sum_{\tau = k_e + nK^B}^{k_e + (n+1)K^B - 1} (Y_{\tau+1} - Y_\tau) \right| \le K^B \delta_{\max}.
\tag{27}
$$

By Lemma 17, we deduce the Lyapunov drift of $Z_n$ as

$$
\begin{aligned}
\frac{Z_{n+1}^2 - Z_n^2}{2} &= \sum_{\tau = k_e + nK^B}^{k_e + (n+1)K^B - 1} \frac{Y_{\tau+1}^2 - Y_\tau^2}{2} \\
&\le \sum_{\tau = k_e + nK^B}^{k_e + (n+1)K^B - 1} -\eta\gamma Y_\tau \\
&\quad + \underbrace{\sum_{\tau = k_e + nK^B}^{k_e + (n+1)K^B - 1} \frac{\eta}{\alpha} \mathbb{E}_{\bar{\mathbb{P}}^{k_e}, \bar{\pi}} \left[ \sum_{h \in [H]} D(\bar{\pi}_h(\cdot|s_h)\|\widetilde{\pi}_h^\tau(\cdot|s_h)) - D(\bar{\pi}_h(\cdot|s_h)\|\widehat{\pi}_h^{\tau+1}(\cdot|s_h)) \right]}_{(I)} \\
&\quad + K^B C_3
\end{aligned}
\tag{28}
$$

where

$$
C_3 = \eta(2\alpha H^3 + 4H^2\theta + 4H^2) + 2\eta^2(9H^2 + 16\alpha^2 H^6 + 1936\alpha^2\eta^2 H^{12}K^2 + 16\theta^2 H^4)
\tag{29}
$$

To bound (I),

$$
\begin{aligned}
(I) &= \frac{\eta}{\alpha} \sum_{h \in [H]} \mathbb{E}_{\bar{\mathbb{P}}^{k_e}, \bar{\pi}} \left[ \sum_{\tau = k_e + nK^B}^{k_e + (n+1)K^B - 1} D(\bar{\pi}_h(\cdot|s_h)\|\widetilde{\pi}_h^\tau(\cdot|s_h)) - D(\bar{\pi}_h(\cdot|s_h)\|\widehat{\pi}_h^{\tau+1}(\cdot|s_h)) \right] \\
&= \frac{\eta}{\alpha} \sum_{h \in [H]} \mathbb{E}_{\bar{\mathbb{P}}^{k_e}, \bar{\pi}} \left[ D(\bar{\pi}(\cdot|s_h)\|\widetilde{\pi}_h^{k_e + nK^B}(\cdot|s_h)) - D(\bar{\pi}(\cdot|s_h)\|\widehat{\pi}_h^{k_e + (n+1)K^B}(\cdot|s_h)) \right] \\
&\quad + \frac{\eta}{\alpha} \sum_{h \in [H]} \mathbb{E}_{\bar{\mathbb{P}}^{k_e}, \bar{\pi}} \left[ \sum_{\tau = k_e + nK^B + 1}^{k_e + (n+1)K^B - 1} D(\bar{\pi}_h(\cdot|s_h)\|\widetilde{\pi}_h^\tau(\cdot|s_h)) - D(\bar{\pi}_h(\cdot|s_h)\|\widehat{\pi}_h^\tau(\cdot|s_h)) \right] \\
&= \frac{\eta}{\alpha} \sum_{h \in [H]} \mathbb{E}_{\bar{\mathbb{P}}^{k_e}, \bar{\pi}} \left[ D(\bar{\pi}(\cdot|s_h)\|\widetilde{\pi}_h^{k_e + nK^B}(\cdot|s_h)) - D(\bar{\pi}(\cdot|s_h)\|\widehat{\pi}_h^{k_e + (n+1)K^B}(\cdot|s_h)) \right] \\
&\le \frac{\eta}{\alpha} \sum_{h \in [H]} \mathbb{E}_{\bar{\mathbb{P}}^{k_e}, \bar{\pi}} \left[ \log(|\mathcal{A}|/\theta) \right] \\
&\le \frac{\eta}{\alpha} H \log(|\mathcal{A}|/\theta)
\end{aligned}
$$

where the last equality is because $\widetilde{\pi}_h^\tau = \widehat{\pi}_h^\tau$ for all $\tau$ such that $\tau - k_e \not\equiv 0 \mod K^B$ by algorithm. The first inequality is because the KL divergence is nonnegative, and we apply Lemma 25, as we know $\widetilde{\pi}_h^{k_e + nK^B} = (1 - \theta)\widehat{\pi}_h^{k_e + nK^B} + \theta\pi_{\text{unif}}$ by algorithm. The last inequality is due to (5). Then we have

$$
\frac{Z_{n+1}^2 - Z_n^2}{2} \le -\eta\gamma \sum_{\tau = k_e + nK^B}^{k_e + (n+1)K^B - 1} Y_\tau + C_4
\tag{30}
$$

where

$$C_4 = \frac{\eta}{\alpha} H \log(|\mathcal{A}|/\theta) + K^B C_3 \tag{31}$$

Suppose that there exists $n \in \{0, \ldots, N_e\}$ such that $Z_n > \frac{2C_4}{K^B \eta \gamma} + K^B \delta_{\max}$. Then we can define the first $n$ that exceeds the threshold $\frac{2C_4}{K^B \eta \gamma} + K^B \delta_{\max}$ as

$$n_{\text{hit}} = \min\left\{n \in \{0, \ldots, N_e\} : Z_n > \frac{2C_4}{K^B \eta \gamma} + K^B \delta_{\max}\right\}.$$

Note that $n_{\text{hit}} \neq 0$, as we set $Y_{k_e} = 0$. Since $n_{\text{hit}}$ is the first time, we have $Z_{n_{\text{hit}}-1} \leq \frac{2C_4}{K^B \eta \gamma} + K^B \delta_{\max}$. Moreover, we have $Y_{k_e+(n_{\text{hit}}-1)K^B}, \ldots, Y_{k_e+n_{\text{hit}}K^B-1} > \frac{2C_4}{K^B \eta \gamma}$. If not, $Z_{n_{\text{hit}}} = Y_{k_e+n_{\text{hit}}K^B}$ cannot be larger than $\frac{2C_4}{K^B \eta \gamma} + K^B \delta_{\max}$. By (30), this implies that

$$\frac{Z_{n_{\text{hit}}}^2 - Z_{n_{\text{hit}}-1}^2}{2} \leq -\eta\gamma \sum_{\tau=k_e+(n_{\text{hit}}-1)K^B}^{k_e+n_{\text{hit}}K^B-1} Y_\tau + C_4 \leq -\eta\gamma K^B \frac{2C_4}{K^B \eta \gamma} + C_4 = -C_4 < 0.$$

This implies that $Z_{n_{\text{hit}}} < Z_{n_{\text{hit}}-1} < \frac{2C_4}{K^B \eta \gamma} + K^B \delta_{\max}$. This contradicts the definition of $n_{\text{hit}}$. Therefore, we have for all $e \in [E]$ and $n \in \{0, \ldots, N_e\}$,

$$Y_{k_e+nK^B} \leq \frac{2C_4}{K^B \eta \gamma} + K^B \delta_{\max}.$$

Moreover, by (26), we have for all $k \in [K]$,

$$Y_k \leq \frac{2C_4}{K^B \eta \gamma} + 2K^B \delta_{\max}.$$

This completes the first statement. Next, we carefully plug our parameter choice into the upper bound on $Y_k$. Recall that the definitions of $C_3, C_4, \delta_{\max}$, and our parameter choice such that

$$B = \frac{3}{4}, \eta = H^{-2}K^{-B}, \alpha = H^{-1}K^{-B}, \theta = K^{-1}.$$

$\delta_{\max}$ is bounded as

$$\begin{aligned}
\delta_{\max} &= 4\eta\alpha H^3(11\eta H^3 K) + 3\eta H + 4\eta\alpha H^3 + 4\eta\theta H^2 \\
&= 44\eta^2 \alpha H^6 K + 3\eta H + 4\eta\alpha H^3 + 4\eta\theta H^2 \\
&= 44HK^{-3B+1} + 3H^{-1}K^{-B} + 4K^{-2B} + 4K^{-1-B} \\
&= \widetilde{\mathcal{O}}\left(HK^{-5/4} + H^{-1}K^{-3/4}\right)
\end{aligned}$$

Since we assumed $H^2 \leq K$, it follows that $HK^{-1/2} \leq 1$. Then we have

$$\delta_{\max} = \widetilde{\mathcal{O}}\left(K^{-3/4}\right)$$

$C_3$ is bounded as

$$\begin{aligned}
C_3 &= \eta(2\alpha H^3 + 4H^2\theta + 4H^2) + 2\eta^2(9H^2 + 16\alpha^2 H^6 + 1936\alpha^2\eta^2 H^{12}K^2 + 16\theta^2 H^4) \\
&= 2K^{-2B} + 4K^{-1-B} + 4K^{-B} + 18H^{-2}K^{-2B} + 32K^{-4B} + 3872H^2K^{2-6B} + 32K^{-2-2B}.
\end{aligned}$$

$2C_4/(\eta K^B \gamma)$ is bounded as

$$\begin{aligned}
\frac{2C_4}{\eta K^B \gamma} &\leq \frac{2H \log(|\mathcal{A}|/\theta)}{K^B \gamma \alpha} + \frac{2C_3}{\eta\gamma} \\
&= \frac{2}{\gamma} H^2 \log(|\mathcal{A}|K) \\
&\quad + \frac{2}{\gamma}(2H^2 K^{-B} + 4H^2 K^{-1} + 4H^2 + 18K^{-B} + 32H^2 K^{-3B} + 3872H^4 K^{2-5B} + 32H^2 K^{-2-B}) \\
&= \widetilde{\mathcal{O}}\left(H^2/\gamma + H^4 K^{-7/4}/\gamma\right).
\end{aligned}$$

Since we assumed $H^2 \le K$, we have $H^4 K^{-7/4} \le H^2$. Then we can drop $\widetilde{\mathcal{O}}(H^4 K^{-7/4}/\gamma)$. $2K^B \delta_{\max}$ is bounded as

$$2K^B \delta_{\max} = 88HK^{-2B+1} + 6H^{-1} + 8K^{-2B} + 8K^{-1}$$
$$= \widetilde{\mathcal{O}}\left(HK^{-1/2}\right)$$

Finally, $Y_k$ is bounded as

$$Y_k = \widetilde{\mathcal{O}}\left(H^2/\gamma\right).$$

$\square$

## J    DETAILED PROOFS FOR THE ANALYSIS

In this section, we first introduce lemmas, which bound an online mirror descent term and optimism terms, and these are useful to prove Lemma 5. Then we present the proofs of Lemmas 3, 4, 5, 6. Then we conclude the section by providing the proof of Theorem 1.

The following lemma is to bound the regret due to online mirror descent. Here, the main difference with the standard online mirror descent lemma (e.g., Hazan et al. (2016); Lattimore & Szepesvári (2020)) comes from the periodic policy mixing, which requires a modified analysis.

**Lemma 19.** *Let $H^2 \le K$. Suppose that $E_g$ holds. Then we have*

$$\sum_{e \in E} \sum_{k \in K_e} \mathbb{E}_{\bar{\mathbb{P}}^{k_e}, \pi^*} \left[ \sum_{h \in [H]} \langle \widehat{Q}^k_{f,h}(s_h, \cdot) + Y_k \widehat{Q}^k_{g,h}(s_h, \cdot), \widehat{\pi}^k_h(\cdot \mid s_h) - \pi^*_h(\cdot \mid s_h) \rangle \right]$$
$$= \widetilde{\mathcal{O}}\left( dH^3 K^{3/4} + \frac{H^6}{\gamma^2} K^{1/4} + \frac{dH^5}{\gamma} \right).$$

*Proof.* Consider $e \in E$. For any $s \in \mathcal{S}$ and $k \in K_e$ such that $\widehat{\pi}^{k+1}_h \ne \pi_{\text{unif}}$ (i.e., $k = k_e, \ldots, k_{e+1} - 2$), we have

$$\widehat{\pi}^{k+1}_h(\cdot \mid s) = \underset{\pi(\cdot \mid s) \in \Delta(\mathcal{A})}{\arg\min} \langle \widehat{Q}^k_{f,h}(s, \cdot) + Y_k \widehat{Q}^k_{g,h}(s, \cdot), \pi(\cdot \mid s) \rangle + \frac{1}{\alpha} D(\pi(\cdot \mid s) || \widetilde{\pi}^k_h(\cdot \mid s)).$$

For ease of notation, we omit $(s, \cdot)$ and $(\cdot \mid s)$. By Lemma 23, for any policy $\pi$,

$$\langle \widehat{Q}^k_{f,h} + Y_k \widehat{Q}^k_{g,h}, \widehat{\pi}^{k+1}_h - \pi \rangle \le \frac{1}{\alpha} D(\pi || \widetilde{\pi}^k_h) - \frac{1}{\alpha} D(\pi || \widehat{\pi}^{k+1}_h) - \frac{1}{\alpha} D(\widehat{\pi}^{k+1}_h || \widetilde{\pi}^k_h).$$

By adding $\langle \widehat{Q}^k_{f,h} + Y_k \widehat{Q}^k_{g,h}, \widehat{\pi}^k_h - \widehat{\pi}^{k+1}_h \rangle$ on both sides, we have for $k = k_e, \ldots, k_{e+1} - 2$,

$$\langle \widehat{Q}^k_{f,h} + Y_k \widehat{Q}^k_{g,h}, \widehat{\pi}^k_h - \pi \rangle$$
$$\le \frac{1}{\alpha} D(\pi || \widetilde{\pi}^k_h) - \frac{1}{\alpha} D(\pi || \widehat{\pi}^{k+1}_h) - \frac{1}{\alpha} D(\widehat{\pi}^{k+1}_h || \widetilde{\pi}^k_h) + \langle \widehat{Q}^k_{f,h} + Y_k \widehat{Q}^k_{g,h}, \widehat{\pi}^k_h - \widehat{\pi}^{k+1}_h \rangle$$
$$\le \frac{1}{\alpha} D(\pi || \widetilde{\pi}^k_h) - \frac{1}{\alpha} D(\pi || \widehat{\pi}^{k+1}_h) - \frac{1}{2\alpha} \| \widehat{\pi}^{k+1}_h - \widetilde{\pi}^k_h \|^2_1 + \| \widehat{Q}^k_{f,h} + Y_k \widehat{Q}^k_{g,h} \|_\infty \| \widehat{\pi}^k_h - \widehat{\pi}^{k+1}_h \|_1$$
$$\le \frac{1}{\alpha} D(\pi || \widetilde{\pi}^k_h) - \frac{1}{\alpha} D(\pi || \widehat{\pi}^{k+1}_h) - \frac{1}{2\alpha} \| \widehat{\pi}^{k+1}_h - \widetilde{\pi}^k_h \|^2_1 + \| \widehat{Q}^k_{f,h} + Y_k \widehat{Q}^k_{g,h} \|_\infty \| \widehat{\pi}^{k+1}_h - \widetilde{\pi}^k_h \|_1$$
$$\quad + \| \widehat{Q}^k_{f,h} + Y_k \widehat{Q}^k_{g,h} \|_\infty \| \widetilde{\pi}^k_h - \widehat{\pi}^k_h \|_1$$
$$= \frac{1}{\alpha} D(\pi || \widetilde{\pi}^k_h) - \frac{1}{\alpha} D(\pi || \widehat{\pi}^k_h) + \frac{1}{\alpha} D(\pi || \widehat{\pi}^k_h) - \frac{1}{\alpha} D(\pi || \widehat{\pi}^{k+1}_h)$$
$$\quad - \frac{1}{2\alpha} \| \widehat{\pi}^{k+1}_h - \widetilde{\pi}^k_h \|^2_1 + \| \widehat{Q}^k_{f,h} + Y_k \widehat{Q}^k_{g,h} \|_\infty \| \widehat{\pi}^{k+1}_h - \widetilde{\pi}^k_h \|_1 + \| \widehat{Q}^k_{f,h} + Y_k \widehat{Q}^k_{g,h} \|_\infty \| \widetilde{\pi}^k_h - \widehat{\pi}^k_h \|_1$$

$$(32)$$

where the second inequality follows from Pinsker's inequality and Hölder's inequality, and the last inequality is due to the triangle inequality. By taking $\sum_{k \in K_e}$ on both sides, we have

$$\sum_{k \in K_e} \langle \widehat{Q}_{f,h}^k + Y_k \widehat{Q}_{g,h}^k, \widehat{\pi}_h^k - \pi \rangle$$

$$\leq \underbrace{\frac{1}{\alpha} \sum_{k=k_e}^{k_{e+1}-2} \left( D(\pi||\widetilde{\pi}_h^k) - D(\pi||\widehat{\pi}_h^k) \right)}_{\text{(I)}} + \underbrace{\frac{1}{\alpha} \sum_{k=k_e}^{k_{e+1}-2} \left( D(\pi||\widehat{\pi}_h^k) - D(\pi||\widehat{\pi}_h^{k+1}) \right)}_{\text{(II)}}$$

$$+ \underbrace{\sum_{k=k_e}^{k_{e+1}-2} \left( -\frac{1}{2\alpha} \|\widehat{\pi}_h^{k+1} - \widetilde{\pi}_h^k\|_1^2 + \|\widehat{Q}_{f,h}^k + Y_k \widehat{Q}_{g,h}^k\|_\infty \|\widehat{\pi}_h^{k+1} - \widetilde{\pi}_h^k\|_1 \right)}_{\text{(III)}} + \underbrace{\sum_{k=k_e}^{k_{e+1}-2} \|\widehat{Q}_{f,h}^k + Y_k \widehat{Q}_{g,h}^k\|_\infty \|\widetilde{\pi}_h^k - \widehat{\pi}_h^k\|_1}_{\text{(IV)}}$$

$$+ \langle \widehat{Q}_{f,h}^{k_{e+1}-1} + Y_{k_{e+1}-1} \widehat{Q}_{g,h}^{k_{e+1}-1}, \widehat{\pi}_h^{k_{e+1}-1} - \pi \rangle.$$

Note that $\langle \widehat{Q}_{f,h}^{k_{e+1}-1} + Y_{k_{e+1}-1} \widehat{Q}_{g,h}^{k_{e+1}-1}, \widehat{\pi}_h^{k_{e+1}-1} - \pi \rangle$ is added, as (32) does not holds for $k_{e+1} - 1$, i.e., the last episode in epoch $e$. Furthermore, this term can be bounded by $2\|\widehat{Q}_{f,h}^{k_{e+1}-1} + Y_{k_{e+1}-1} \widehat{Q}_{g,h}^{k_{e+1}-1}\|_\infty$ using Hölder's inequality.

To bound (I), we observe the following. If $k - k_e \not\equiv 0 \mod K^B$, then $\widetilde{\pi}_h^k = \widehat{\pi}_h^k$. Thus, $D(\pi||\widetilde{\pi}_h^k) - D(\pi||\widehat{\pi}_h^k) = 0$. Otherwise, since $\widetilde{\pi}_h^k = (1 - \theta)\widehat{\pi}_h^k + \theta \pi_{\text{unif}}$, we can apply Lemma 25, and thus $D(\pi||\widetilde{\pi}_h^k) - D(\pi||\widehat{\pi}_h^k) \leq \theta \log|\mathcal{A}|$, i.e.,

$$D(\pi||\widetilde{\pi}_h^k) - D(\pi||\widehat{\pi}_h^k) \leq \begin{cases} \theta \log|\mathcal{A}| & \text{if } k - k_e \equiv 0 \mod K^B, \\ 0 & \text{otherwise.} \end{cases}$$

It follows that

$$\text{(I)} \leq \frac{1}{\alpha} \sum_{k \in K_e: k-k_e \equiv 0 \mod K^B} \theta \log|\mathcal{A}| \leq \frac{\theta |K_e| \log|\mathcal{A}|}{\alpha K^B} + \frac{\theta \log|\mathcal{A}|}{\alpha}$$

where the second inequality is due to $|\{k \in K_e : k - k_e \equiv 0 \mod K^B\}| \leq \lceil |K_e|/K^B \rceil \leq |K_e|/K^B + 1$. Furthermore, since (II) is in the form of a telescoping sum and $\widehat{\pi}_h^{k_e} = \pi_{\text{unif}}$, we have

$$\text{(II)} \leq \frac{1}{\alpha} D(\pi||\widehat{\pi}_h^{k_e}) \leq \frac{\log|\mathcal{A}|}{\alpha}.$$

To bound (III), since $-ax^2 + bx \leq b^2/(4a)$ for $a, b, x \geq 0$, we have

$$\text{(III)} \leq \frac{\alpha |K_e| C_5^2}{2}.$$

where $C_5$ is a constant such that $\|\widehat{Q}_{f,h}^k + Y_k \widehat{Q}_{g,h}^k\|_\infty \leq C_5$ for all $h, k$. Again, by definition of $\widetilde{\pi}_h^k$, we have

$$\text{(IV)} = \sum_{k \in K_e: k-k_e \equiv 0 \mod K^B} \|\widehat{Q}_{f,h}^k + Y_k \widehat{Q}_{g,h}^k\|_\infty \|\widetilde{\pi}_h^k - \widehat{\pi}_h^k\|_1$$

$$= \sum_{k \in K_e: k-k_e \equiv 0 \mod K^B} \|\widehat{Q}_{f,h}^k + Y_k \widehat{Q}_{g,h}^k\|_\infty \theta \|\pi_{\text{unif}} - \widehat{\pi}_h^k\|_1$$

$$\leq \frac{2\theta |K_e| C_5}{K^B} + 2\theta C_5.$$

Finally, we have for any policy $\pi$ and $s \in \mathcal{S}$,

$$\sum_{k \in K_e} \langle \widehat{Q}_{f,h}^k(s, \cdot) + Y_k \widehat{Q}_{g,h}^k(s, \cdot), \widehat{\pi}_h^k(\cdot \mid s) - \pi(\cdot \mid s) \rangle$$

$$\leq \frac{\theta |K_e| \log|\mathcal{A}|}{\alpha K^B} + \frac{(1 + \theta) \log|\mathcal{A}|}{\alpha} + \frac{\alpha |K_e| C_5^2}{2} + \frac{2\theta |K_e| C_5}{\alpha K^B} + 2\theta C_5 + 2C_5.$$

Let us take $\pi = \pi_h^*$ for each $h \in [H]$. Then, by taking $\sum_{h \in [H]}$ and $\mathbb{E}_{\bar{\mathbb{P}}^{k_e}, \pi^*}$, it follows that

$$\mathbb{E}_{\bar{\mathbb{P}}^{k_e}, \pi^*} \left[ \sum_{h \in [H]} \sum_{k \in K_e} \langle \widehat{Q}_{f,h}^k(s_h, \cdot) + Y_k \widehat{Q}_{g,h}^k(s_h, \cdot), \widehat{\pi}_h^k(\cdot \mid s_h) - \pi_h^*(\cdot \mid s_h) \rangle \right]$$

$$\leq \mathbb{E}_{\bar{\mathbb{P}}^{k_e}, \pi^*} \left[ H \left( \frac{\theta |K_e| \log |\mathcal{A}|}{\alpha K^B} + \frac{(1+\theta) \log |\mathcal{A}|}{\alpha} + \frac{\alpha |K_e| C_5^2}{2} + \frac{2\theta |K_e| C_5}{K^B} + 2\theta C_5 + 2C_5 \right) \right]$$

$$\leq H \left( \frac{\theta |K_e| \log |\mathcal{A}|}{\alpha K^B} + \frac{(1+\theta) \log |\mathcal{A}|}{\alpha} + \frac{\alpha |K_e| C_5^2}{2} + \frac{2\theta |K_e| C_5}{K^B} + 2\theta C_5 + 2C_5 \right).$$

Finally, by $E_g$ and Lemma 18, we have $C_5 = \widetilde{\mathcal{O}}(H^3/\gamma)$. Furthermore, by Lemma 33, the number of epochs is at most $\widetilde{\mathcal{O}}(dH)$. Then, by taking $\sum_{e \in E}$ to the above inequality, it follows that

$$\sum_{e \in E} \sum_{k \in K_e} \mathbb{E}_{\bar{\mathbb{P}}^{k_e}, \pi^*} \left[ \sum_{h \in [H]} \langle \widehat{Q}_{f,h}^k(s_h, \cdot) + Y_k \widehat{Q}_{g,h}^k(s_h, \cdot), \widehat{\pi}_h^k(\cdot \mid s_h) - \pi_h^*(\cdot \mid s_h) \rangle \right]$$

$$= \widetilde{\mathcal{O}} \left( dH^3 K^{3/4} + \frac{H^6}{\gamma^2} K^{1/4} + \frac{dH^5}{\gamma} \right).$$

$\square$

The next lemma claims that the regret terms associated with optimism are nonpositive, highlighting the effectiveness of our bonus terms. We closely follow the proof of Lemma 4 of Cassel & Rosenberg (2024).

**Lemma 20.** *Let $H^2 \leq K$. Suppose that $E_g$ holds. For all $(s, a, h, k, \ell) \in \mathcal{S} \times \mathcal{A} \times [H] \times [K] \times \{f, g\}$,*

$$\widehat{Q}_{\ell,h}^k(s, a) - \bar{\phi}_h^{k_e}(s, a)^\top (\theta_{\ell,h}^k + \psi_h \widehat{V}_{\ell,h+1}^k) \leq 0$$

*Proof.* By definition, we have

$$\widehat{Q}_{\ell,h}^k(s, a) - \bar{\phi}_h^{k_e}(s, a)^\top (\theta_{\ell,h}^k + \psi_h \widehat{V}_{\ell,h+1}^k)$$

$$= \bar{\phi}_h^{k_e}(s, a)^\top (\widehat{\theta}_{\ell,h}^k - \theta_{\ell,h}^k) + \bar{\phi}_h^{k_e}(s, a)^\top (\widehat{\psi}_h^k - \psi_h) \widehat{V}_{\ell,h+1}^k - \beta_b \|\bar{\phi}_h^{k_e}(s, a)\|_{(\Lambda_h^{k_e})^{-1}}$$

$$\leq \|\widehat{\theta}_{\ell,h}^k - \theta_{\ell,h}^k\|_{\Lambda_h^{k_e}} \|\bar{\phi}_h^{k_e}(s, a)\|_{(\Lambda_h^{k_e})^{-1}} + \|(\widehat{\psi}_h^k - \psi_h) \widehat{V}_{\ell,h+1}^k\|_{\Lambda_h^{k_e}} \|\bar{\phi}_h^{k_e}(s, a)\|_{(\Lambda_h^{k_e})^{-1}} - \beta_b \|\bar{\phi}_h^{k_e}(s, a)\|_{(\Lambda_h^{k_e})^{-1}}$$

$$\leq \|\widehat{\theta}_{\ell,h}^k - \theta_{\ell,h}^k\|_{\Lambda_h^k} \|\bar{\phi}_h^{k_e}(s, a)\|_{(\Lambda_h^{k_e})^{-1}} + \|(\widehat{\psi}_h^k - \psi_h) \widehat{V}_{\ell,h+1}^k\|_{\Lambda_h^k} \|\bar{\phi}_h^{k_e}(s, a)\|_{(\Lambda_h^{k_e})^{-1}} - \beta_b \|\bar{\phi}_h^{k_e}(s, a)\|_{(\Lambda_h^{k_e})^{-1}}$$

$$\leq \beta_r \|\bar{\phi}_h^{k_e}(s, a)\|_{(\Lambda_h^{k_e})^{-1}} + \beta_p \|\bar{\phi}_h^{k_e}(s, a)\|_{(\Lambda_h^{k_e})^{-1}} - \beta_b \|\bar{\phi}_h^{k_e}(s, a)\|_{(\Lambda_h^{k_e})^{-1}}$$

$$= 0$$

where the first inequality is due to the Cauchy-Schwarz inequality, the second inequality follows from $\Lambda_h^{k_e} \preceq \Lambda_h^k$, and the last equality is because $\beta_b = \beta_r + \beta_p$. $\square$

**Proof of Lemma 3** By Lemma 8,

$$V_{\ell,1}^{\pi^k}(s_1) - \widehat{V}_{\ell,1}^k(s_1) = \mathbb{E}_{\mathbb{P}, \widehat{\pi}^k} \left[ \sum_{h \in [H]} \phi(s_h^k, a_h^k)^\top (\theta_{\ell,h}^k + \psi_h \widehat{V}_{\ell,h+1}^k) - \widehat{Q}_{\ell,h}^k(s_h^k, a_h^k) \right]$$

$$= \underbrace{\mathbb{E}_{\mathbb{P}, \widehat{\pi}^k} \left[ \sum_{h \in [H]} \bar{\phi}_h^{k_e}(s_h^k, a_h^k)^\top (\theta_{\ell,h}^k + \psi_h \widehat{V}_{\ell,h+1}^k) - \widehat{Q}_{\ell,h}^k(s_h^k, a_h^k) \right]}_{\text{(I)}}$$

$$+ \underbrace{\mathbb{E}_{\mathbb{P}, \widehat{\pi}^k} \left[ \sum_{h \in [H]} (\phi(s_h^k, a_h^k) - \bar{\phi}_h^{k_e}(s_h^k, a_h^k))^\top (\theta_{\ell,h}^k + \psi_h \widehat{V}_{\ell,h+1}^k) \right]}_{\text{(II)}}.$$

To bound (I), we have

$$\bar{\phi}_h^{k_e}(s_h^k, a_h^k)^\top (\theta_{\ell,h}^k + \psi_h \widehat{V}_{\ell,h+1}^k) - \widehat{Q}_{\ell,h}^k(s_h^k, a_h^k)$$
$$= \bar{\phi}_h^{k_e}(s_h^k, a_h^k)^\top (\theta_{\ell,h}^k - \widehat{\theta}_{\ell,h}^k) + \bar{\phi}_h^{k_e}(s_h^k, a_h^k)^\top (\psi_h - \widehat{\psi}_h^k) \widehat{V}_{\ell,h+1}^k + \beta_b \|\bar{\phi}_h^{k_e}(s_h^k, a_h^k)\|_{(\Lambda_h^{k_e})^{-1}}$$
$$\leq \beta_b \|\bar{\phi}_h^{k_e}(s_h^k, a_h^k)\|_{(\Lambda_h^k)^{-1}} + \beta_b \|\bar{\phi}_h^{k_e}(s_h^k, a_h^k)\|_{(\Lambda_h^{k_e})^{-1}}$$

where the inequality is due to the Cauchy-Schwarz inequality. Furthermore, since $k \in K_e$, it must hold $\det(\Lambda_h^k) \leq 2 \det(\Lambda_h^{k_e})$, otherwise $k$ would belong to epoch $e + 1$. As it is obvious that $(\Lambda_h^k)^{-1} \preceq (\Lambda_h^{k_e})^{-1}$, we can apply Lemma 34 for nonzero $\bar{\phi}_h^{k_e}(s_h^k, a_h^k)$ as follows.

$$\frac{\|\bar{\phi}_h^{k_e}(s_h^k, a_h^k)\|_{(\Lambda_h^{k_e})^{-1}}^2}{\|\bar{\phi}_h^{k_e}(s_h^k, a_h^k)\|_{(\Lambda_h^k)^{-1}}^2} \leq \frac{\det((\Lambda_h^{k_e})^{-1})}{\det((\Lambda_h^k)^{-1})} \leq 2.$$

This implies that $\beta_b \|\bar{\phi}_h^{k_e}(s_h^k, a_h^k)\|_{(\Lambda_h^{k_e})^{-1}} \leq 2\beta_b \|\bar{\phi}_h^{k_e}(s_h^k, a_h^k)\|_{(\Lambda_h^k)^{-1}}$ for nonzero $\bar{\phi}_h^{k_e}(s_h^k, a_h^k)$. If $\bar{\phi}_h^{k_e}(s_h^k, a_h^k) = 0$, then the inequality is trivial. Then it follows that

$$\text{(I)} \leq \mathbb{E}_{\mathbb{P},\widehat{\pi}^k} \left[ \sum_{h \in [H]} 3\beta_b \|\bar{\phi}_h^{k_e}(s_h^k, a_h^k)\|_{(\Lambda_h^k)^{-1}} \right] \leq \mathbb{E}_{\mathbb{P},\widehat{\pi}^k} \left[ \sum_{h \in [H]} 3\beta_b \|\phi(s_h^k, a_h^k)\|_{(\Lambda_h^k)^{-1}} \right].$$

where the second inequality is due to $\|\bar{\phi}_h^{k_e}(s_h^k, a_h^k)\|_{(\Lambda_h^k)^{-1}} \leq \|\phi(s_h^k, a_h^k)\|_{(\Lambda_h^k)^{-1}}$. To bound (II), by Lemma 32, we have

$$(\phi(s_h^k, a_h^k) - \bar{\phi}_h^{k_e}(s_h^k, a_h^k))^\top (\theta_{\ell,h}^k + \psi_h \widehat{V}_{\ell,h+1}^k)$$
$$\leq (4\beta_w^2 \|\phi(s_h^k, a_h^k)\|_{(\Lambda_h^k)^{-1}}^2 + 2K^{-1}) |\phi(s_h^k, a_h^k)^\top (\theta_{\ell,h}^k + \psi_h \widehat{V}_{\ell,h+1}^k)|$$
$$\leq 16H\beta_w^2 \|\phi(s_h^k, a_h^k)\|_{(\Lambda_h^k)^{-1}}^2 + 8HK^{-1}$$

where the second inequality follows from the fact that for any $\ell \in \{f, g\}$,

$$\phi(s_h^k, a_h^k)^\top (\theta_{\ell,h}^k + \psi_h \widehat{V}_{\ell,h+1}^k) = \ell_h^k(s_h^k, a_h^k) + \sum_{s' \in \mathcal{S}} \mathbb{P}_h(s' \mid s_h^k, a_h^k) \widehat{V}_{\ell,h+1}^k(s') \leq 1 + \|\widehat{V}_{\ell,h+1}^k\|_\infty \leq 4H.$$

This implies that $\text{(II)} \leq 16H\beta_w^2 \|\phi(s_h^k, a_h^k)\|_{(\Lambda_h^k)^{-1}}^2 + 8HK^{-1}$. Finally, we have

$$V_{\ell,1}^{\pi^k}(s_1) - \widehat{V}_{\ell,1}^k(s_1) \leq \mathbb{E}_{\mathbb{P},\widehat{\pi}^k} \left[ \sum_{h \in [H]} 3\beta_b \|\phi(s_h^k, a_h^k)\|_{(\Lambda_h^k)^{-1}} + 16H\beta_w^2 \|\phi(s_h^k, a_h^k)\|_{(\Lambda_h^k)^{-1}}^2 \right] + 8H^2 K^{-1}.$$

Taking $\sum_{k=1}^K$ on both sides,

$$\sum_{k=1}^K (V_{\ell,1}^{\pi^k}(s_1) - \widehat{V}_{\ell,1}^k(s_1)) \leq \sum_{k=1}^K \mathbb{E}_{\mathbb{P},\widehat{\pi}^k} \left[ \sum_{h \in [H]} 3\beta_b \|\phi(s_h^k, a_h^k)\|_{(\Lambda_h^k)^{-1}} + 16H\beta_w^2 \|\phi(s_h^k, a_h^k)\|_{(\Lambda_h^k)^{-1}}^2 \right] + 8H^2$$
$$\leq \sum_{k=1}^K \sum_{h \in [H]} \left( 6\beta_b \|\phi(s_h^k, a_h^k)\|_{(\Lambda_h^k)^{-1}} + 32H\beta_w^2 \|\phi(s_h^k, a_h^k)\|_{(\Lambda_h^k)^{-1}}^2 \right)$$
$$+ 8H(3\beta_b + 16H\beta_w^2) \log \frac{6K}{\delta} + 8H^2$$

where the second inequality follows from $E_g$. By Lemma 35, we have

$$\sum_{k \in [K]} \|\phi(s_h^k, a_h^k)\|_{(\Lambda_h^k)^{-1}}^2 \leq 2 \log \frac{\det(\Lambda_h^{K+1})}{\det(\Lambda_h^1)} \leq 2d \log(K + 1)$$

where the second inequality follows from $\|\Lambda_h^{k+1}\|_2 = \|I + \sum_{\tau \in [k]} \phi(s_h^\tau, a_h^\tau)\phi(s_h^\tau, a_h^\tau)^\top\|_2 \le 1 + k$, and thus $\det(\Lambda_h^{K+1}) \le (K+1)^d$. Furthermore, the Cauchy-Schwarz inequality implies that $\sum_{k \in [K]} \|\phi(s_h^k, a_h^k)\|_{(\Lambda_h^k)^{-1}} \le \sqrt{2dK \log(K+1)}$. Then we deduce that

$$\sum_{k=1}^K (V_{\ell,1}^{\pi^k}(s_1) - \widehat{V}_{\ell,1}^k(s_1)) \le 6\beta_b H \sqrt{2dK \log(K+1)} + 64dH^2 \beta_w^2 \log(K+1)$$

$$+ 8H(3\beta_b + 16H\beta_w^2) \log \frac{6K}{\delta} + 8H^2$$

$$= \widetilde{\mathcal{O}}\left(\sqrt{d^3 H^4} K^{3/4} + d^3 H^4 K^{1/2}\right)$$

where the last equality follows from $\beta_b, \beta_w = \widetilde{\mathcal{O}}\left(K^{1/4}dH\right)$. □

**Proof of Lemma 4**  Given $e \in E$, for any $k \in K_e$, the dual variable $Y_k$ is updated as

$$Y_{k+1} = \begin{cases} 0 & \text{if } k+1 = k_e, \\ \left[(1 - 4\alpha\eta H^3)Y_k + \eta\left(\widehat{V}_{g,1}^k(s_1) - b - 4\alpha H^3 - 4\theta H^2\right)\right]_+ & \text{otherwise.} \end{cases}$$

Then it follows that

$$0 \le Y_{k_{e+1}-1}^2$$

$$= \sum_{k=k_e}^{k_{e+1}-2} \left(Y_{k+1}^2 - Y_k^2\right)$$

$$= \sum_{k=k_e}^{k_{e+1}-2} \left(\left[(1 - 4\alpha\eta H^3)Y_k + \eta\left(\widehat{V}_{g,1}^k(s_1) - b - 4\alpha H^3 - 4\theta H^2\right)\right]_+^2 - Y_k^2\right)$$

$$= \sum_{k=k_e}^{k_{e+1}-2} \left(\left[Y_k + \eta\left(\widehat{V}_{g,1}^k(s_1) - b - 4\alpha H^3 - 4\theta H^2 - 4\alpha H^3 Y_k\right)\right]_+^2 - Y_k^2\right)$$

$$\le \sum_{k=k_e}^{k_{e+1}-2} \left(2Y_k\eta\left(\widehat{V}_{g,1}^k(s_1) - b - 4\alpha H^3 - 4\theta H^2 - 4\alpha H^3 Y_k\right) + \eta^2\left(\widehat{V}_{g,1}^k(s_1) - b - 4\alpha H^3 - 4\theta H^2 - 4\alpha H^3 Y_k\right)^2\right)$$

where the first equality is due to $Y_{k_e} = 0$, and the last inequality is due to the fact that $\max\{0, z\}^2 \le z^2$ for any $z \in \mathbb{R}$. This can be rewritten as

$$\sum_{k=k_e}^{k_{e+1}-2} Y_k(b - \widehat{V}_{g,1}^k(s_1))$$

$$\le \sum_{k=k_e}^{k_{e+1}-2} Y_k(-4\alpha H^3 - 4\theta H^2 - 4\alpha H^3 Y_k) + \frac{\eta}{2} \sum_{k=k_e}^{k_{e+1}-2} \left(\widehat{V}_{g,1}^k(s_1) - b - 4\alpha H^3 - 4\theta H^2 - 4\alpha H^3 Y_k\right)^2.$$

(33)

Note that the first term is nonpositive. Furthermore, the second term can be bounded as

$$|\widehat{V}_{g,1}^k(s_1) - b - 4\alpha H^3 - 4\theta H^2 - 4\alpha H^3 Y_k| \le |\widehat{V}_{g,1}^k(s_1) - b| + 4\alpha H^3 + 4\theta H^2 + 4\alpha H^3 Y_k$$

$$\le 3H + 4H^2 K^{-3/4} + 4H^2 K^{-1} + 4H^2 K^{-3/4} Y_k$$

$$\le 3H + 4H^{1/2} + 4 + 4H^2 K^{-3/4} Y_k$$

$$\le 11H + 4H^2 K^{-3/4} Y_k$$

where the second inequality follows from $E_g$, the third inequality is because we assumed that $H^2 \le K$. Thus, (33) is bounded as

$$\sum_{k=k_e}^{k_{e+1}-2} Y_k(b - \widehat{V}_{g,1}^k(s_1)) \le \frac{\eta}{2} \sum_{k=k_e}^{k_{e+1}-2} \left(11H + 4H^2 K^{-3/4} Y_k\right)^2$$

$$\le \frac{\eta}{2} \sum_{k=k_e}^{k_{e+1}-2} 2(121H^2 + 16H^4 K^{-3/2} Y_k^2)$$

where the second inequality is due to the Cauchy-Schwarz inequality. Then we have

$$\sum_{k \in K_e} Y_k(b - \widehat{V}_{g,1}^k(s_1)) = \sum_{k=k_e}^{k_{e+1}-2} Y_k(b - \widehat{V}_{g,1}^k(s_1)) + Y_{k_{e+1}-1}(b - \widehat{V}_{g,1}^{k_{e+1}-1}(s_1))$$

$$\le \frac{\eta}{2} \sum_{k=k_e}^{k_{e+1}-2} 2(121H^2 + 16H^4 K^{-3/2} Y_k^2) + 3HY_{k_{e+1}-1}$$

$$\le 121\eta H^2 |K_e| + 16\eta H^4 K^{-3/2} \sum_{k=k_e}^{k_{e+1}-2} Y_k^2 + 3HY_{k_{e+1}-1}.$$

By Lemma 18, we have $Y_k = \widetilde{\mathcal{O}}(H^2/\gamma)$ for all $k \in [K]$. Furthermore, by Lemma 33, the number of epochs is at most $\widetilde{\mathcal{O}}(dH)$. By taking $\sum_{e \in E}$, it follows that

$$\sum_{k \in [K]} Y_k(b - \widehat{V}_{g,1}^k(s_1)) = \sum_{e \in E} \sum_{k \in K_e} Y_k(b - \widehat{V}_{g,1}^k(s_1))$$

$$= \widetilde{\mathcal{O}}\left(K^{1/4} + \frac{dH^6}{\gamma^2}\right).$$

$\square$

**Proof of Lemma 5**  Note that

$$\sum_{k=1}^{K} \left(\widehat{V}_{f,1}^k(s_1) + Y_k \widehat{V}_{g,1}^k(s_1) - V_{f^k,1}^{\pi^*}(s_1) - Y_k V_{g,1}^{\pi^*}(s_1)\right)$$

$$\le \sum_{e \in E} \sum_{k \in K_e} \left(\widehat{V}_{f,1}^k(s_1) + Y_k \widehat{V}_{g,1}^k(s_1) - \bar{V}_{f^k,1}^{\pi^*}(s_1) - Y_k \bar{V}_{g,1}^{\pi^*}(s_1)\right)$$

where $\bar{V}_{f^k,1}^{\pi^*}, \bar{V}_{g,1}^{\pi^*}$ are the value functions with respect to a contracted MDP. Furthermore, by Lemma 8, it follows that

$$\sum_{e \in E} \sum_{k \in K_e} \left(\widehat{V}_{f,1}^k(s_1) + Y_k \widehat{V}_{g,1}^k(s_1) - \bar{V}_{f^k,1}^{\pi^*}(s_1) - Y_k \bar{V}_{g,1}^{\pi^*}(s_1)\right)$$

$$= \sum_{e \in E} \sum_{k \in K_e} \mathbb{E}_{\bar{\mathbb{P}}^{k_e}, \pi^*}\left[\sum_{h \in [H]} \sum_{a \in \mathcal{A}} (\widehat{Q}_{f,h}^k(s_h^k, a) + Y_k \widehat{Q}_{g,h}^k(s_h^k, a))(\widehat{\pi}_h^k(a \mid s_h^k) - \pi_h^*(a \mid s_h^k))\right]$$

$$+ \sum_{e \in E} \sum_{k \in K_e} \mathbb{E}_{\bar{\mathbb{P}}^{k_e}, \pi^*}\left[\sum_{h \in [H]} \widehat{Q}_{f,h}^k(s_h^k, a_h^k) - \bar{f}_h^k(s_h^k, a_h^k) - \sum_{s' \in \mathcal{S}} \bar{\mathbb{P}}_h^{k_e}(s' \mid s_h^k, a_h^k) \widehat{V}_{f,h+1}^k(s')\right]$$

$$+ \sum_{e \in E} \sum_{k \in K_e} \mathbb{E}_{\bar{\mathbb{P}}^{k_e}, \pi^*}\left[\sum_{h \in [H]} Y_k \left(\widehat{Q}_{g,h}^k(s_h^k, a_h^k) - \bar{g}_h(s_h^k, a_h^k) - \sum_{s' \in \mathcal{S}} \bar{\mathbb{P}}_h^{k_e}(s' \mid s_h^k, a_h^k) \widehat{V}_{g,h+1}^k(s')\right)\right].$$

Note that by the definition of the contracted MDP, we have $\bar{f}_h^k(s,a) = \bar{\phi}_h^{k_e}(s,a)^\top \theta_{f,h}^k$, $\bar{g}_h(s,a) = \bar{\phi}_h^{k_e}(s,a)^\top \theta_{g,h}$, and $\bar{\mathbb{P}}_h^{k_e}(s' \mid s,a) = \bar{\phi}_h^{k_e}(s,a)^\top \psi_h(s')$. Then it follows that

$$\bar{f}_h(s_h^k, a_h^k) + \sum_{s' \in \mathcal{S}} \bar{\mathbb{P}}_h^{k_e}(s' \mid s_h^k, a_h^k) \widehat{V}_{f,h+1}^k(s') = \bar{\phi}_h^{k_e}(s_h^k, a_h^k)^\top (\theta_{f,h}^k + \psi_h \widehat{V}_{f,h+1}^k),$$

$$\bar{g}_h(s_h^k, a_h^k) + \sum_{s' \in \mathcal{S}} \bar{\mathbb{P}}_h^{k_e}(s' \mid s_h^k, a_h^k) \widehat{V}_{g,h+1}^k(s') = \bar{\phi}_h^{k_e}(s_h^k, a_h^k)^\top (\theta_{g,h} + \psi_h \widehat{V}_{g,h+1}^k).$$

We deduce that

$$\sum_{k=1}^K \left( \widehat{V}_{f,1}^k(s_1) + Y_k \widehat{V}_{g,1}^k(s_1) - V_{f^k,1}^{\pi^*}(s_1) - Y_k V_{g,1}^{\pi^*}(s_1) \right)$$

$$\leq \sum_{e \in E} \sum_{k \in K_e} \left( \widehat{V}_{f,1}^k(s_1) + Y_k \widehat{V}_{g,1}^k(s_1) - \bar{V}_{f^k,1}^{\pi^*}(s_1) - Y_k \bar{V}_{g,1}^{\pi^*}(s_1) \right)$$

$$= \underbrace{\sum_{e \in E} \sum_{k \in K_e} \mathbb{E}_{\bar{\mathbb{P}}^{k_e}, \pi^*} \left[ \sum_{h \in [H]} \sum_{a \in \mathcal{A}} (\widehat{Q}_{f,h}^k(s_h^k, a) + Y_k \widehat{Q}_{g,h}^k(s_h^k, a))(\widehat{\pi}_h^k(a \mid s_h^k) - \pi_h^*(a \mid s_h^k)) \right]}_{\text{(I)}}$$

$$+ \underbrace{\sum_{e \in E} \sum_{k \in K_e} \mathbb{E}_{\bar{\mathbb{P}}^{k_e}, \pi^*} \left[ \sum_{h \in [H]} \widehat{Q}_{f,h}^k(s_h^k, a_h^k) - \bar{\phi}_h^{k_e}(s_h^k, a_h^k)^\top (\theta_{f,h}^k + \psi_h \widehat{V}_{f,h+1}^k) \right]}_{\text{(II)}}$$

$$+ \underbrace{\sum_{e \in E} \sum_{k \in K_e} \mathbb{E}_{\bar{\mathbb{P}}^{k_e}, \pi^*} \left[ \sum_{h \in [H]} Y_k \left( \widehat{Q}_{g,h}^k(s_h^k, a_h^k) - \bar{\phi}_h^{k_e}(s_h^k, a_h^k)^\top (\theta_{g,h} + \psi_h \widehat{V}_{g,h+1}^k) \right) \right]}_{\text{(III)}}.$$

By Lemma 19,

$$\text{(I)} = \widetilde{\mathcal{O}} \left( dH^3 K^{3/4} + \frac{H^6}{\gamma^2} K^{1/4} + \frac{dH^5}{\gamma} \right).$$

Note that $Y_k \geq 0$ for all $k$. Then, by Lemma 20, for any $(s, a, h, k) \in \mathcal{S} \times \mathcal{A} \times [H] \times [K]$, we have

$$\widehat{Q}_{f,h}^k(s,a) - \bar{\phi}_h^{k_e}(s,a)^\top (\theta_{f,h}^k + \psi_h \widehat{V}_{f,h+1}^k) \leq 0,$$

$$Y_k \left( \widehat{Q}_{g,h}^k(s,a) - \bar{\phi}_h^{k_e}(s,a)^\top (\theta_{g,h} + \psi_h \widehat{V}_{g,h+1}^k) \right) \leq 0.$$

This implies that

$$\text{(II)}, \text{(III)} \leq 0.$$

Finally, we have

$$\sum_{k=1}^K \left( \widehat{V}_{f,1}^k(s_1) + Y_k \widehat{V}_{g,1}^k(s_1) - V_{f^k,1}^{\pi^*}(s_1) - Y_k V_{g,1}^{\pi^*}(s_1) \right) = \widetilde{\mathcal{O}} \left( dH^3 K^{3/4} + \frac{H^6}{\gamma^2} K^{1/4} + \frac{dH^5}{\gamma} \right).$$

$\square$

**Proof of Lemma 6**  Note that the dual update is

$$Y_{k+1} = \begin{cases} 0 & \text{if } k + 1 = k_e, \\ \left[ (1 - 4\alpha\eta H^3) Y_k + \eta \left( \widehat{V}_{g,1}^k(s_1) - b - 4\alpha H^3 - 4\theta H^2 \right) \right]_+ & \text{otherwise.} \end{cases}$$

Then it follows that for any $e \in E$,

$$
\begin{aligned}
Y_{k_{e+1}-1} &= \left[ (1 - 4\alpha\eta H^3)Y_{k_{e+1}-2} + \eta\left(\widehat{V}_{g,1}^{k_{e+1}-2}(s_1) - b - 4\alpha H^3 - 4\theta H^2\right)\right]_+ \\
&\geq (1 - 4\alpha\eta H^3)Y_{k_{e+1}-2} + \eta\left(\widehat{V}_{g,1}^{k_{e+1}-2}(s_1) - b - 4\alpha H^3 - 4\theta H^2\right) \\
&= Y_{k_{e+1}-2} + \eta\left(\widehat{V}_{g,1}^{k_{e+1}-2}(s_1) - b - 4\alpha H^3(1 + Y_{k_{e+1}-2}) - 4\theta H^2\right) \\
&\ \ \vdots \\
&\geq Y_{k_e} + \eta \sum_{k=k_e}^{k_{e+1}-2}\left(\widehat{V}_{g,1}^{k}(s_1) - b - 4\alpha H^3(1 + Y_k) - 4\theta H^2\right).
\end{aligned}
$$

Note that $Y_{k_e} = 0$. Then we have

$$
\begin{aligned}
\sum_{k=k_e}^{k_{e+1}-1}\left(\widehat{V}_{g,1}^{k}(s_1) - b\right) &= \sum_{k=k_e}^{k_{e+1}-2}\left(\widehat{V}_{g,1}^{k}(s_1) - b\right) + \left(\widehat{V}_{g,1}^{k_{e+1}-1}(s_1) - b\right) \\
&\leq \frac{Y_{k_{e+1}-1}}{\eta} + \sum_{k=k_e}^{k_{e+1}-2}\left(4\alpha H^3(1 + Y_k) + 4\theta H^2\right) + 3H.
\end{aligned}
$$

By Lemma 18, we have $Y_k = \widetilde{\mathcal{O}}(H^2/\gamma)$ for all $k \in [K]$. Then it follows that

$$
4\alpha H^3(1 + Y_k) + 4\theta H^2 = 4H^2K^{-3/4}(1 + Y_k) + 4H^2K^{-1} = \widetilde{\mathcal{O}}\left(\frac{H^4}{\gamma}K^{-3/4} + H^2K^{-1}\right).
$$

Furthermore, by Lemma 33, the number of epochs is at most $\widetilde{\mathcal{O}}(dH)$. By taking $\sum_{e \in E}$, it follows that

$$
\sum_{e \in E}\sum_{k \in K_e}\left(\widehat{V}_{g,1}^{k}(s_1) - b\right) = \widetilde{\mathcal{O}}\left(\frac{dH^5}{\gamma}K^{3/4} + \frac{H^4}{\gamma}K^{1/4}\right).
$$

$\square$

**Proof of Theorem 1** If $K < \beta_w$, we cannot use Lemma 16. Nevertheless, in this case, we have the following upper bounds for regret and violation.

$$
\text{Regret}(K) \leq HK < H\beta_w = \widetilde{\mathcal{O}}\left(dH^2K^{1/4}\right),
$$

$$
\text{Violation}(K) \leq HK < H\beta_w = \widetilde{\mathcal{O}}\left(dH^2K^{1/4}\right).
$$

Otherwise, it is trivial that the conditions of Lemma 16 hold, i.e., $E_g$ holds with probability at least $1 - \delta$. Furthermore, under $E_g$, we have the upper bound on $Y_k$ as in Lemma 18, i.e.,

$$
Y_k = \widetilde{\mathcal{O}}\left(\frac{H^2}{\gamma}\right) \tag{34}
$$

Thus, with probability at least $1 - \delta$, $E_g$ and (34) hold.

Now, we begin with the proof of the regret upper bound. Note that an optimal policy $\pi^*$ satisfies $V_{g,1}^{\pi^*}(s_1) \leq b$. Since $Y_k \geq 0$ for all $k$, it follows that

$\text{Regret}(K)$

$$
= \sum_{k=1}^{K}\left(V_{f^k,1}^{\pi^k}(s_1) - \widehat{V}_{f,1}^{k}(s_1)\right) + \sum_{k=1}^{K}Y_k(b - \widehat{V}_{g,1}^{k}(s_1)) + \sum_{k=1}^{K}\left(\widehat{V}_{f,1}^{k}(s_1) + Y_k\widehat{V}_{g,1}^{k}(s_1) - V_{f^k,1}^{\pi^*}(s_1) - Y_kb\right)
$$

$$
\leq \underbrace{\sum_{k=1}^{K}\left(V_{f^k,1}^{\pi^k}(s_1) - \widehat{V}_{f,1}^{k}(s_1)\right)}_{(I)} + \underbrace{\sum_{k=1}^{K}Y_k(b - \widehat{V}_{g,1}^{k}(s_1))}_{(II)} + \underbrace{\sum_{k=1}^{K}\left(\widehat{V}_{f,1}^{k}(s_1) + Y_k\widehat{V}_{g,1}^{k}(s_1) - V_{f^k,1}^{\pi^*}(s_1) - Y_kV_{g,1}^{\pi^*}(s_1)\right)}_{(III)}
$$

By Lemma 3,

$$(\text{I}) = \widetilde{\mathcal{O}}\left(\sqrt{d^3 H^4} K^{3/4} + d^3 H^4 K^{1/2}\right).$$

By Lemma 4,

$$(\text{II}) = \widetilde{\mathcal{O}}\left(K^{1/4} + \frac{dH^6}{\gamma^2}\right).$$

By Lemma 5,

$$(\text{III}) = \widetilde{\mathcal{O}}\left(dH^3 K^{3/4} + \frac{H^6}{\gamma^2} K^{1/4} + \frac{dH^5}{\gamma}\right).$$

Thus, we have the following regret upper bound.

$$\text{Regret}(K) = \widetilde{\mathcal{O}}\left(\sqrt{d^3 H^4} K^{3/4} + dH^3 K^{3/4} + d^3 H^4 K^{1/2} + \frac{H^6}{\gamma^2} K^{1/4} + \frac{dH^6}{\gamma^2}\right).$$

Next, we show the violation upper bound. If constraint violation is 0, the statement is trivial. Otherwise, we decompose it as

$$\text{Violation}(K) \leq \underbrace{\sum_{k=1}^{K}\left(V_{g,1}^{\pi^k}(s_1) - \widehat{V}_{g,1}^{k}(s_1)\right)}_{(\text{IV})} + \underbrace{\sum_{k=1}^{K}\left(\widehat{V}_{g,1}^{k}(s_1) - b\right)}_{(\text{V})}.$$

By Lemma 3,

$$(\text{IV}) = \widetilde{\mathcal{O}}\left(\sqrt{d^3 H^4} K^{3/4} + d^3 H^4 K^{1/2}\right).$$

By Lemma 6,

$$(\text{V}) = \widetilde{\mathcal{O}}\left(\frac{dH^5}{\gamma} K^{3/4} + \frac{H^4}{\gamma} K^{1/4}\right).$$

Thus, we have the following violation upper bound.

$$\text{Violation}(K) = \widetilde{\mathcal{O}}\left(\frac{dH^5}{\gamma} K^{3/4} + \sqrt{d^3 H^4} K^{3/4} + d^3 H^4 K^{1/2}\right).$$

$\square$

## K  AUXILIARY LEMMAS

**Lemma 21.** *Let $c > 0$. For $x \in [c, \infty)^d$, let $\log x = (\log x_1, \ldots, \log x_d)^\top$. For any $x, y \in [c, \infty)^d$, we have*

$$\|\log x - \log y\|_\infty \leq \frac{1}{c}\|x - y\|_1$$

*Proof.* Fix $x, y \in [c, \infty)^d$ and $i \in \{1, \ldots, d\}$. Consider the scalar function $p_i : [0, 1] \to \mathbb{R}$ defined as

$$p_i(t) := \log(y_i + t(x_i - y_i)).$$

Since $[c, \infty)^d$ is convex, we have $y_i + t(x_i - y_i) \geq \min\{x_i, y_i\} \geq c$ for all $t \in [0, 1]$, so $p_i$ is continuously differentiable and

$$p_i'(t) = \frac{x_i - y_i}{y_i + t(x_i - y_i)}.$$

Hence, for all $t \in [0, 1]$,

$$|p_i'(t)| = \frac{|x_i - y_i|}{y_i + t(x_i - y_i)} \leq \frac{|x_i - y_i|}{c}.$$

By the fundamental theorem of calculus,

$$|\log x_i - \log y_i| \;=\; |p_i(1) - p_i(0)| \;=\; |\int_0^1 p_i'(t)\,dt| \;\leq\; \int_0^1 |p_i'(t)|\,dt \;\leq\; \frac{|x_i - y_i|}{c}.$$

Taking the maximum over $i$ and using $\|z\|_\infty \leq \|z\|_1$ for all $z \in \mathbb{R}^d$ yields

$$\|\log x - \log y\|_\infty = \max_{1 \leq i \leq d} |\log x_i - \log y_i| \leq \frac{1}{c} \max_{1 \leq i \leq d} |x_i - y_i| \leq \frac{1}{c} \sum_{i=1}^d |x_i - y_i| = \frac{1}{c} \|x - y\|_1.$$

$\square$

**Lemma 22.** *Let $\widehat{\pi}_h^k : \mathcal{S} \to \Delta(\mathcal{A})$ be any policies. For $\theta \in [0,1]$, let $\widetilde{\pi}_h^k(\cdot|s) = (1 - \theta)\widehat{\pi}_h^k(\cdot|s) + \theta\pi_{\mathrm{unif}}(\cdot|s)$. For $\widehat{Q}_{f,h}^k, \widehat{Q}_{g,h}^k : \mathcal{S} \times \mathcal{A} \to [-2H, 2H]$, $Y_k \in \mathbb{R}_+$, and $\alpha > 0$, let $\widehat{\pi}^{k+1}(\cdot \mid s) \propto \widetilde{\pi}^k(\cdot \mid s) \exp(-\alpha(\widehat{Q}_{f,h}^k(s,\cdot) + Y_k\widehat{Q}_{g,h}^k(s,\cdot)))$. For any $s \in \mathcal{S}$, we have*

1. $\|\widehat{\pi}_h^{k+1}(\cdot \mid s) - \widetilde{\pi}_h^k(\cdot \mid s)\|_1 \leq 2\alpha H(1 + Y_k)$,

2. $\left|\langle \widehat{\pi}_h^{k+1}(\cdot|s) - \widehat{\pi}_h^k(\cdot|s), \widehat{Q}_{g,h}^k(s,\cdot)\rangle\right| \leq 4\alpha H^2(1 + Y_k) + 4\theta H$.

3. $\langle \widehat{\pi}_h^k(\cdot|s) - \widehat{\pi}_h^{k+1}(\cdot|s), \widehat{Q}_{f,h}^k(s,\cdot)\rangle - \frac{1}{\alpha}D(\widehat{\pi}_h^{k+1}(\cdot|s)\|\widetilde{\pi}_h^k(\cdot|s)) \leq 2\alpha H^2 + 4H\theta$.

*Proof.* **(Proof of the first statement)** We show the first statement. Given $s \in \mathcal{S}$, we omit $(\cdot \mid s)$ in notation for simplicity. Note that $\widehat{\pi}_h^{k+1}$ can be viewed as an optimal solution for $\min_\pi \langle \pi, \widehat{Q}_{f,h}^k + Y_k\widehat{Q}_{g,h}^k\rangle + (1/\alpha)D(\pi\|\widetilde{\pi}_h^k)$. Due to the pushback lemma (Lemma 23), by taking $z = \widetilde{\pi}^k$,

$$\langle \widehat{\pi}_h^{k+1}, \widehat{Q}_{f,h}^k + Y_k\widehat{Q}_{g,h}^k\rangle + \frac{1}{\alpha}D(\widehat{\pi}_h^{k+1}\|\widetilde{\pi}_h^k) \leq \langle \widetilde{\pi}_h^k, \widehat{Q}_{f,h}^k + Y_k\widehat{Q}_{g,h}^k\rangle + \frac{1}{\alpha}D(\widetilde{\pi}_h^k\|\widetilde{\pi}_h^k) - \frac{1}{\alpha}D(\widetilde{\pi}_h^k\|\widehat{\pi}_h^{k+1}).$$

Note that $D(\widetilde{\pi}_h^k\|\widetilde{\pi}_h^k) = 0$. This can be rewritten as

$$\frac{1}{\alpha}D(\widehat{\pi}_h^{k+1}\|\widetilde{\pi}_h^k) + \frac{1}{\alpha}D(\widetilde{\pi}_h^k\|\widehat{\pi}_h^{k+1}) \leq \langle \widetilde{\pi}_h^k - \widehat{\pi}_h^{k+1}, \widehat{Q}_{f,h}^k + Y_k\widehat{Q}_{g,h}^k\rangle.$$

To lower bound the left-hand side, Pinsker's inequality implies that

$$\frac{1}{2}\|\widehat{\pi}_h^{k+1} - \widetilde{\pi}_h^k\|_1^2 \leq D(\widehat{\pi}_h^{k+1}\|\widetilde{\pi}_h^k), \quad \frac{1}{2}\|\widehat{\pi}_h^{k+1} - \widetilde{\pi}_h^k\|_1^2 \leq D(\widetilde{\pi}_h^k\|\widehat{\pi}_h^{k+1}).$$

To upper bound the right-hand side, Hölder's inequality implies that

$$\langle \widetilde{\pi}_h^k - \widehat{\pi}_h^{k+1}, \widehat{Q}_{f,h}^k + Y_k\widehat{Q}_{g,h}^k\rangle \leq \|\widetilde{\pi}_h^k - \widehat{\pi}_h^{k+1}\|_1 \|\widehat{Q}_{f,h}^k + Y_k\widehat{Q}_{g,h}^k\|_\infty.$$

As a result, we deduce that

$$\frac{1}{\alpha}\|\widehat{\pi}_h^{k+1} - \widetilde{\pi}_h^k\|_1^2 \leq \|\widetilde{\pi}_h^k - \widehat{\pi}_h^{k+1}\|_1 \|\widehat{Q}_{f,h}^k + Y_k\widehat{Q}_{g,h}^k\|_\infty.$$

If $\|\widehat{\pi}_h^{k+1} - \widetilde{\pi}_h^k\|_1 = 0$, then the statement is trivial. Otherwise, it follows that

$$\|\widehat{\pi}_h^{k+1} - \widetilde{\pi}_h^k\|_1 \leq \alpha\|\widehat{Q}_{f,h}^k + Y_k\widehat{Q}_{g,h}^k\|_\infty.$$

Since $\|\widehat{Q}_{f,h}^k\|_\infty, \|\widehat{Q}_{g,h}^k\|_\infty \leq 2H$ and $Y_k \geq 0$, we have

$$\|\widehat{\pi}_h^{k+1} - \widetilde{\pi}_h^k\|_1 \leq 2\alpha H(1 + Y_k).$$

**(Proof of the second statement)** Now, we show the second statement. By Hölder's inequality and the triangle inequality,

$$\left|\langle \widehat{\pi}_h^{k+1} - \widehat{\pi}_h^k, \widehat{Q}_{g,h}^k\rangle\right| \leq \|\widehat{\pi}_h^{k+1} - \widehat{\pi}_h^k\|_1 \|\widehat{Q}_{g,h}^k\|_\infty \leq \|\widehat{\pi}_h^{k+1} - \widetilde{\pi}_h^k\|_1 \|\widehat{Q}_{g,h}^k\|_\infty + \|\widetilde{\pi}_h^k - \widehat{\pi}_h^k\|_1 \|\widehat{Q}_{g,h}^k\|_\infty.$$

By the first statement,

$$\|\widehat{\pi}_h^{k+1} - \widetilde{\pi}_h^k\|_1 \leq \alpha\|\widehat{Q}_{f,h}^k(s,\cdot) + Y_k\widehat{Q}_{g,h}^k(s,\cdot)\|_\infty.$$

Furthermore, we have

$$\|\widetilde{\pi}_h^k - \widehat{\pi}_h^k\|_1 = \|(1-\theta)\widehat{\pi}_h^k + \theta\pi_{\text{unif}} - \widehat{\pi}_h^k\|_1 = \theta\|-\widehat{\pi}_h^k + \pi_{\text{unif}}\|_1 \leq 2\theta.$$

Finally, we have

$$\left|\langle\widehat{\pi}_h^{k+1} - \widehat{\pi}_h^k, \widehat{Q}_{g,h}^k\rangle\right| \leq \alpha\|\widehat{Q}_{f,h}^k(s,\cdot) + Y_k\widehat{Q}_{g,h}^k(s,\cdot)\|_\infty\|\widehat{Q}_{g,h}^k\|_\infty + 2\theta\|\widehat{Q}_{g,h}^k\|_\infty.$$

Since $\|\widehat{Q}_{f,h}^k\|_\infty, \|\widehat{Q}_{g,h}^k\|_\infty \leq 2H$ and $Y_k \geq 0$,

$$\left|\langle\widehat{\pi}_h^{k+1} - \widehat{\pi}_h^k, \widehat{Q}_{g,h}^k\rangle\right| \leq 4\alpha H^2(1 + Y_k) + 4\theta H.$$

**(Proof of the third statement)** Note that

$$\langle\widehat{\pi}_h^k - \widehat{\pi}_h^{k+1}, \widehat{Q}_{f,h}^k\rangle - \frac{1}{\alpha}D(\widehat{\pi}_h^{k+1}||\widetilde{\pi}_h^k) \leq \|\widehat{\pi}_h^{k+1} - \widehat{\pi}_h^k\|_1\|\widehat{Q}_{f,h}^k\|_\infty - \frac{1}{2\alpha}\|\widehat{\pi}_h^{k+1} - \widetilde{\pi}_h^k\|_1^2$$

$$\leq 2H\|\widehat{\pi}_h^{k+1} - \widehat{\pi}_h^k\|_1 - \frac{1}{2\alpha}\|\widehat{\pi}_h^{k+1} - \widetilde{\pi}_h^k\|_1^2$$

$$\leq 2H\|\widehat{\pi}_h^{k+1} - \widetilde{\pi}_h^k\|_1 - \frac{1}{2\alpha}\|\widehat{\pi}_h^{k+1} - \widetilde{\pi}_h^k\|_1^2 + 2H\|\widetilde{\pi}_h^k - \widehat{\pi}_h^k\|_1$$

$$\leq 2\alpha H^2 + 2H\|\widetilde{\pi}_h^k - \widehat{\pi}_h^k\|_1$$

where the first inequality is due to Hölder's inequality and Pinsker's inequality, the third inequality is due to the triangle inequality, and the last inequality follows from the fact that $-ax^2 + bx \leq b^2/(4a)$ for $a, b, x > 0$, i.e., $a = 1/(2\alpha)$, $b = 2H$, $x = \|\widehat{\pi}_h^{k+1} - \widetilde{\pi}_h^k\|_1$. The following is true.

$$\|\widehat{\pi}_h^k - \widetilde{\pi}_h^k\|_1 \leq \theta\|\widehat{\pi}_h^k - \pi_{\text{unif}}\|_1 \leq \theta(\|\widehat{\pi}_h^k\|_1 + \|\pi_{\text{unif}}\|_1) \leq 2\theta.$$

Finally, we have

$$\langle\widehat{\pi}_h^k - \widehat{\pi}_h^{k+1}, \widehat{Q}_{f,h}^k\rangle - \frac{1}{\alpha}D(\widehat{\pi}_h^{k+1}||\widetilde{\pi}_h^k) \leq 2\alpha H^2 + 4H\theta$$

as desired. $\qquad\square$

**Lemma 23** (Lemma 1 of Wei et al. (2020), Lemma C.3 of Qiu et al. (2020) ). *Let $\Delta$, $\text{int}(\Delta)$ be the probability simplex and its interior, respectively, and let $f : \mathcal{C} \to \mathbb{R}$ be a convex function. Fix $\alpha > 0$, $y \in \text{int}(\Delta)$. Suppose $x^* \in \arg\min_{x\in\Delta} f(x) + (1/\alpha)D(x||y)$ and $x^* \in \text{int}(\Delta)$, then, for any $z \in \Delta$,*

$$f(x^*) + \frac{1}{\alpha}D(x^*||y) \leq f(z) + \frac{1}{\alpha}D(z||y) - \frac{1}{\alpha}D(z||x^*).$$

**Lemma 24** (Lemma 33 of Kitamura et al. (2025)). *Let $Q_1, Q_2 : \mathcal{A} \to \mathbb{R}$ be two functions. For $\alpha > 0$, let $\pi_1 \propto \exp(\alpha Q_1)$, $\pi_2 \propto \exp(\alpha Q_2)$. Then we have*

$$\|\pi_1 - \pi_2\|_1 \leq 8\alpha\|Q_1 - Q_2\|_\infty.$$

**Lemma 25** (Lemma 31 of Wei et al. (2020)). *Let $\pi_1, \pi_2$ be two probability distributions in $\Delta(\mathcal{A})$. Let $\tilde{\pi}_2 = (1-\theta)\pi_2 + \theta/|\mathcal{A}|$ where $\theta \in (0, 1)$. Then,*

$$D(\pi_1||\tilde{\pi}_2) - D(\pi_1||\pi_2) \leq \theta\log|\mathcal{A}|, \quad D(\pi_1||\tilde{\pi}_2) \leq \log(|\mathcal{A}|/\theta).$$

**Lemma 26** (Lemma 24 of Cassel & Rosenberg (2024)). *Let $\mathcal{V} = \{V(\cdot; \theta) : \|\theta\| \leq W\}$ denote a class of functions $V : \mathcal{S} \to \mathbb{R}$. Suppose that any $V \in \mathcal{V}$ is $L$-Lipschitz with respect to $\theta$ and the supremum distance, i.e.,*

$$\|V(\cdot; \theta_1) - V(\cdot; \theta_2)\|_\infty \leq L\|\theta_1 - \theta_2\|_1, \quad \|\theta_1\|_2, \|\theta_2\|_2 \leq W.$$

*Let $\mathcal{N}_\epsilon$ be the $\epsilon$-covering number of $\mathcal{V}$ with respect to the supremum distance. Then*

$$\log\mathcal{N}_\epsilon \leq d\log(1 + 2WL/\epsilon).$$

**Lemma 27** (Lemma 18 of Cassel & Rosenberg (2024)). *For any $K \geq 1$, $\beta > 0$, we have that*

$$\max_{y \geq 0} [y \cdot \sigma(-\beta y + \log K)] \leq \frac{2 \log K}{\beta}.$$

**Lemma 28** (Lemma D.4 of Rosenberg et al. (2020)). *Let $\{X_t\}_{t \geq 1}$ be a sequence of random variables with expectation adapted to a filtration $\mathcal{F}_t$. Suppose that $0 \leq X_t \leq C$ almost surely. Then with probability at least $1 - \delta$,*

$$\sum_{t=1}^{T} \mathbb{E}[X_t \mid \mathcal{F}_{t-1}] \leq 2 \sum_{t=1}^{T} X_t + 4C \log \frac{2T}{\delta}.$$

**Lemma 29** (Lemma 21 of Cassel & Rosenberg (2024)). *Let $\widehat{\theta}_{g,h}^k$ be as in line 14 of Algorithm 1. With probability at least $1 - \delta$, for all $k \geq 1$, $h \in [H]$,*

$$\|\theta_{g,h} - \widehat{\theta}_{g,h}^k\|_{\Lambda_h^k} \leq 2\sqrt{2d \log(2KH/\delta)}.$$

**Lemma 30** (Lemma 22 of Cassel & Rosenberg (2024)). *Let $\widehat{\psi}_h^k : \mathbb{R}^{|\mathcal{S}|} \to \mathbb{R}^d$ be the linear operator defined in line 16 of Algorithm 1. For all $h \in [H]$, let $\widehat{\mathcal{V}}_h \subset \mathbb{R}^{|\mathcal{S}|}$ be a set of mappings $\widehat{V} : \mathcal{S} \to \mathbb{R}$ such that $\|\widehat{V}\|_\infty \leq \beta_Q$ and $\beta_Q \geq 1$. With probability at least $1 - \delta$, for all $h \in [H]$, $\widehat{V} \in \widehat{\mathcal{V}}_{h+1}$, and $k \geq 1$,*

$$\|(\psi_h - \widehat{\psi}_h^k)\widehat{V}\|_{\Lambda_h^k} \leq 4\beta_Q \sqrt{d \log(K + 1) + 2 \log(H\mathcal{N}_\epsilon/\delta)},$$

*where $\epsilon \leq \beta_Q \sqrt{d}/(2K)$, $\mathcal{N}_\epsilon = \sum_{h \in [H]} \mathcal{N}_\epsilon(\widehat{\mathcal{V}}_h)$, and $\mathcal{N}_\epsilon(\widehat{\mathcal{V}}_h)$ is the $\epsilon$-covering number of $\widehat{\mathcal{V}}_h$ with respect to the $\ell_\infty$-norm.*

**Lemma 31** (Lemma 17 of Cassel & Rosenberg (2024)). *For any $\lambda > 0$ and matrices $\Lambda, \Lambda' \in \mathbb{R}^{d \times d}$ satisfying $\Lambda, \Lambda' \succeq \lambda I$, we have that*

$$\|\Lambda^{1/2} - (\Lambda')^{1/2}\|_2 \leq \frac{1}{2\sqrt{\lambda}}\|\Lambda - \Lambda'\|_2.$$

**Lemma 32** (Lemma 3 of Cassel & Rosenberg (2024)). *For any $e \in [E]$ and $v \in \mathbb{R}^d$, we have that*

$$\left(\phi(s_h, a_h) - \bar{\phi}_h^{k_e}(s_h, a_h)\right)^\top v \leq \left(4\beta_w^2 \|\phi(s_h, a_h)\|_{(\Lambda_h^k)^{-1}}^2 + 2K^{-1}\right) \left|\phi(s_h, a_h)^\top v\right|.$$

**Lemma 33** (Lemma 8 of Cassel & Rosenberg (2024)). *The number of epochs $|E|$ is bounded by $(3/2)dH \log(2K)$.*

**Lemma 34** (Lemma 12 of Abbasi-Yadkori et al. (2011)). *Let $A, B, C$ be positive semi-definite matrices such that $A = B + C$. Then, we have that*

$$\sup_{x \neq 0} \frac{x^\top A x}{x^\top B x} \leq \frac{\det(A)}{\det(B)}.$$

**Lemma 35** (Lemma D.2 of Jin et al. (2020)). *Let $\{\phi_t\}_{t \geq 0}$ be a bounded sequence in $\mathbb{R}^d$ satisfying $\sup_{t \geq 0} \|\phi_t\| \leq 1$. Let $\Lambda_0 \in \mathbb{R}^{d \times d}$ be a positive definite matrix. For any $t \geq 0$, define*

$$\Lambda_t = \Lambda_0 + \sum_{j=1}^{t} \phi_j \phi_j^\top.$$

*Then, if the smallest eigenvalue of $\Lambda_0$ satisfies $\lambda_{\min}(\Lambda_0) \geq 1$, we have*

$$\log \left[\frac{\det(\Lambda_t)}{\det(\Lambda_0)}\right] \leq \sum_{j=1}^{t} \phi_j^\top \Lambda_{j-1}^{-1} \phi_j \leq 2 \log \left[\frac{\det(\Lambda_t)}{\det(\Lambda_0)}\right].$$

## L  NUMERICAL EXPERIMENT

We evaluate Algorithm 1 on a finite-horizon job-scheduling CMDP closely following the setup of Ghosh et al. (2022) with modifications to incorporate adversarial losses. The number of episodes and the horizon are set to $K = 100{,}000$ and $H = 10$, respectively, and the state space is $\mathcal{S} = \{0, 1, \ldots, 9\}$. At step $h$, $s_h$ denotes the number of remaining jobs in the stack, and each episode begins with the initial state $s_1 = 9$. The agent chooses $a_h \in \mathcal{A} = \{0, 1\}$, where $a_h = 1$ corresponds to processing the current job and $a_h = 0$ corresponds to idling. Specifically, if $a_h = 1$, then $s_{h+1} = \max\{s_h - 2, 0\}$ with probability 0.8, $s_{h+1} = \max\{s_h - 1, 0\}$ with probability 0.1, and $s_{h+1} = s_h$ otherwise. If $a_h = 0$, then $s_{h+1} = s_h$.

The loss and cost functions are defined as follows. To simulate an adversarial setting, in each episode $k$, the loss is chosen between two functions, $f^{(1)}$ and $f^{(2)}$, with probabilities $0.9 - 0.9(k-1)/(K-1)$ and $0.1 + 0.9(k - 1)/(K - 1)$, respectively. These functions are defined as

$$f_h^{(1)}(a_h) = \begin{cases} 1 & a_h = 0, \\ 0.55 & a_h = 1 \text{ and } h \in \{3, 4, 5, 6\}, \\ 0.2 & a_h = 1 \text{ and } h \notin \{3, 4, 5, 6\}, \end{cases} \quad f_h^{(2)}(a_h) = \begin{cases} 1 & a_h = 0, \\ 0.6 & a_h = 1 \text{ and } h \in \{4, 5, 6\}, \\ 0.2 & a_h = 1 \text{ and } h \notin \{4, 5, 6\}. \end{cases}$$

The cost is defined as $g_h(s_h, a_h, s_{h+1}) = 1 - (s_h - s_{h+1})/2$ for all $h$, and the cost budget is set to $b = 5.6$.

Figure 1 summarizes the results of running Algorithm 1 for $K = 100{,}000$ episodes. To promote learning, we set the parameters as $\alpha = 0.1$, $\beta_b = K^{1/4}$, and $\beta_w = \beta_b \log K$, while keeping the other parameters same as in the setup of Algorithm 1. As shown in Figure 1a, the regret grows sublinearly in $K$, despite the fact that the losses are not sampled from a fixed distribution. Furthermore, Figure 1b shows that while the constraint violation grows rapidly in the early phase, it eventually converges to 0 approximately after episode 45,000. These results support our main claim that both regret and constraint violation are bounded by sublinear terms.

## M  THE USE OF LARGE LANGUAGE MODELS

Portions of the text were polished using ChatGPT-5, which was employed for grammar checking and sentence refinement.

