# OpenReview forum: "Primal-Dual Policy Optimization for Linear CMDPs with Adversarial Losses"
_ICLR.cc/2026/Conference — ICLR 2026 Poster_

### Official Review · Reviewer_Nuny · 2025-10-17

**Soundness:** 3
**Presentation:** 3
**Contribution:** 3
**Rating:** 6
**Confidence:** 3

**Summary:**

The paper initiates the study of adversarial linear CMDPs by proposing an efficient primal-dual policy optimization method for linear CMDPs with adversarial losses and stochastic constraints. The algorithm assumes full feedback on the loss and bandit feedback on the constraints. This method attains $\widetilde{O}(K^{3/4})$ regret and violation where $K$ is the number of episodes of the learning dynamic. Slater's condition is assumed, while the algorithm does not need the knowledge of the Slater's parameter $\gamma$.

**Strengths:**

The main contribution of the work is the setting studied. Indeed, the literature on adversarial CMDPs, up to now, mainly focuses on tabular CMDPs, while linear ones may be more useful in many real world applications. Moreover, while I am not an expert on linear MDPs settings, the algorithm seems to have sufficient technical novelties. To conclude, the overall contribution of the work seems sufficient for acceptance, even if there are many weaknesses that I will highlight in the following.

**Weaknesses:**

1. The different feedback for losses and constraints is not particularly reasonable in practice. I understand that the choice is made since bandit feedback and adversarial settings were too challenging while full feedback and stochastic settings are generally trivial. Still, I see it as a weakness of the work.
2. A similar reasoning can be done for the nature of losses and constraints. Nonetheless, given the well known impossibility result on learning with adversarial constraints, this is not the first work on CMDPs focusing on the adversarial losses and stochastic constraints setting.
3. No lower bounds are provided. Thus, the tightness of the bounds cannot be inferred.
4. In tabular CMDPs, $K^{3/4}$ regret and violation can be attained without the dependence on the Slater's parameter $\gamma$. Differently, in this work, the algorithm attains $K^{3/4}/\gamma$ violation.

**Questions:**

See weaknesses.

Moreover, I am curious about what would happen to the bounds of the algorithm assuming the knowledge of the Slater's parameter $\gamma$. Specifically, assuming $\gamma$ is known, I believe that the space of the dual variables che be optimally bonded to something of order $H/\gamma$, and no policy mixing/fixed shared approaches are needed. In such a case, can $\sqrt K$ regret and violation bounds be attained by the algorithm?

---

> ### Author Response · Authors · 2025-11-20
>
> We thank the reviewer for the insightful comments and address the questions as follows.
>
> > W1. The different feedback for losses and constraints is not particularly reasonable in practice. I understand that the choice is made since bandit feedback and adversarial settings were too challenging while full feedback and stochastic settings are generally trivial. Still, I see it as a weakness of the work.
>
> - We agree with the reviewer that assuming different feedback scenarios for loss and constraint functions is not entirely reflective of real-world applications. However, since our setting allows loss functions to be adversarially chosen, our algorithm inherently accommodates the case where stochastic bandit feedback is provided for both the loss and the constraint.
>
> - For the adversarial linear CMDP setting under bandit feedback, we regard this as an open problem. Notably, even in the unconstrained case, a computationally efficient algorithm with optimal regret is still unknown [Liu2024]. Since the constrained setting is strictly more challenging, it is natural to leave this direction for future work.
>
> - Furthermore, especially in the linear CMDP setting, we clarify that handling adversarial losses is non-trivial even under full feedback. For comparison, [Qiu2020] studied the same feedback regime but only in the tabular CMDP setting. Importantly, their approach does not require controlling the covering number, which is one of the main challenges in the linear CMDP setting and a key difficulty addressed in our work. In compensation for that, our algorithm can control large state spaces.
>
> > W2. A similar reasoning can be done for the nature of losses and constraints. Nonetheless, given the well known impossibility result on learning with adversarial constraints, this is not the first work on CMDPs focusing on the adversarial losses and stochastic constraints setting.
>
> - To clarify, our work is the first to address **linear** CMDPs under adversarial losses and stochastic constraints. As the reviewer correctly noted, several prior works have studied adversarial environments in the tabular CMDP setting (e.g., [Qiu2020, Stradi2025]). However, these results do not extend to settings where the number of states is large.
>
> - In contrast, our algorithms can scale effectively even when the state space is large. In particular, our regret and constraint violation bounds are independent of $|\mathcal{S}|$. Moreover, our algorithm is model-free because, at each step, it only requires computing the $Q$-function estimate and the policy only for the visited state $s_h^k$---as well as applying policy mixing---as in usual policy optimization algorithms in the linear MDP literature.
>
>
> > W3. No lower bounds are provided. Thus, the tightness of the bounds cannot be inferred.
>
> - We note that the regret lower bound of $\Omega(\sqrt{H^3 d^2 K})$ for stochastic linear unconstrained MDPs (e.g., see [Zhou2021] or [He2023]) also applies to our setting. This is because by taking the loss to be fixed across episodes and using a trivial constraint (i.e., taking $b = H$), our problem reduces to a stochastic linear unconstrained MDP. Therefore, we conjecture that there remains room for improving our regret bound by a factor of $\widetilde{\mathcal{O}}({K^{1/4}})$.
>
> - Additionally, we outline a promising direction toward achieving $\widetilde{\mathcal{O}}(\sqrt{K})$ regret and violation in our setting. The main challenge arises when analyzing the constraint violation without relying on mixing steps. In our current analysis, the mixing step is inevitable to mitigate KL divergence terms in the drift upper bound (Lemma 17); these KL divergence terms arise from mirror-descent updates to control adversarial losses. However, mixing becomes problematic for linear CMDPs because it enlarges the covering number. If one can design an approach that controls the dual variable without mixing, then achieving optimal bounds may become possible.
>
> - In the Additional Discussion section of the appendix, we added the discussion on the regret lower bound for our setting, along with future directions for achieving the optimal regret bound of $\widetilde{\mathcal{O}}(\sqrt{K})$.
>
> > References
>
> - [Liu2024] Towards Optimal Regret in Adversarial Linear MDPs with Bandit Feedback, ICLR 2024.
>
> - [Qiu2020] Upper Confidence Primal-Dual Reinforcement Learning for CMDP with Adversarial Loss, NeurIPS 2020.
>
> - [Stradi2025] Policy Optimization for CMDPs with Bandit Feedback: Learning Stochastic and Adversarial Constraints, ICML 2025.
>
> - [Zhou2021] Nearly minimax optimal reinforcement learning for linear mixture markov decision processes, COLT 2021.
>
> - [He2023] Nearly minimax optimal reinforcement learning for linear markov decision processes, ICML 2023.

---

> ### Author Response · Authors · 2025-11-20
>
> > W4. In tabular CMDPs, $K^{3/4}$ regret and violation can be attained without the dependence on the Slater's parameter $\gamma$. Differently, in this work, the algorithm attains $K^{3/4} / \gamma$ violation.
>
> - As the reviewer correctly pointed out, several works have achieved regret and constraint violation bounds without a multiplicative $1/\gamma$ factor. For example, in the tabular CMDP setting with adversarial constraints under bandit feedback, [Stradi2025] proved that their algorithm attains $\widetilde{\mathcal{O}}(K^{3/4})$ regret bounds even when $\gamma \approx 0$. One of their key techniques to achieve this is an additional clipping the dual variable at $K^{1/4}$.
>
> - In contrast, our regret and constraint violation bounds contain terms with a factor of  $\widetilde{\mathcal{O}}(1/\gamma)$, which become suboptimal as $\gamma \to 0$. At a high level, this dependence arises when we derive the negative drift term in the Lyapunov drift analysis (Lemma 17), and this phenomenon is common in drift-based analyses (e.g., [Wei2022]). Nevertheless, we believe that eliminating $1/\gamma$ factor in our setting would be meaningful and non-trivial. Therefore, we regard this as an interesting direction for future work.
>
> > Q1. Moreover, I am curious about what would happen to the bounds of the algorithm assuming the knowledge of the Slater's parameter $\gamma$. Specifically, assuming $\gamma$ is known, I believe that the space of the dual variables can be optimally bounded to something of order $H/\gamma$, and no policy mixing/fixed shared approaches are needed. In such a case, can $\sqrt{K}$ regret and violation bounds be attained by the algorithm?
>
> - Although we assume the knowledge of $\gamma$, it is non-trivial to achieve a $\widetilde{\mathcal{O}}(\sqrt{K})$ regret bound, because such techniques cannot be applied to the adversarial setting. The reason is as follows.
>
> - As the reviewer correctly pointed out, the strong duality of CMDPs implies that the optimal dual variable is bounded by $H/\gamma$. Building on this, [Ding2021, Ghosh2022] applied a dual clipping technique---cutting off the dual variable when it exceeds $2H/\gamma$---and this technique is key to achieving $\widetilde{\mathcal{O}}(\sqrt{K})$ bounds in their setting.
>
> - However, the analysis of [Ding2021, Ghosh2022] relies on the assumption that both the loss and constraint remain fixed across episodes. In particular, to leverage the strong duality of CMDPs, they reformulated the weighted sum of regret and violation into a simple Lagrangian form (e.g., Appendix D of [Ghosh2022]), i.e., there exists $\pi'$ such that
> $$
>     \frac{1}{K}\bigg(\sum_{k=1}^K (V_{f,1}^{\pi^k} - V_{f,1}^{\pi^\star}) + Y\sum_{k=1}^K (V_{g,1}^{\pi^k} - b)\bigg) = (V_{f,1}^{\pi'} - V_{f,1}^{\pi^\star}) + Y (V_{g,1}^{\pi'} - b).
> $$
> In our case, when losses are adversarially chosen in each episode, reformulating the sum into a simple Lagrangian form is not allowed, i.e., there does not exist $f'$ and $\pi'$ such that
> $$
>     \frac{1}{K}\bigg(\sum_{k=1}^K (V_{f^k,1}^{\pi^k} - V_{f^k,1}^{\pi^\star}) + Y\sum_{k=1}^K (V_{g,1}^{\pi^k} - b)\bigg) \neq (V_{f',1}^{\pi'} - V_{f',1}^{\pi^\star}) + Y (V_{g,1}^{\pi'} - b).
> $$
> In turn, leveraging strong duality in our setting is non-trivial.
>
> - Consequently, their analysis is restricted to the stochastic setting, in which the losses and constraints (or their expectations) are fixed. This limitation motivates our new algorithmic components---periodic policy mixing and a regularized dual update---which are specifically designed to control the dual variable in adversarial environments.
>
>
> > References
>
> - [Stradi2025] Policy Optimization for CMDPs with Bandit Feedback: Learning Stochastic and Adversarial Constraints, ICML 2025.
>
> - [Wei2022] Triple-q: A model-free algorithm for constrained reinforcement learning with sublinear regret and zero constraint violation, AISTATS 2022.
>
> - [Ding2021] Provably Efficient Safe Exploration via Primal-Dual Policy Optimization, AISTATS 2021.
>
> - [Ghosh2022] Provably Efficient Model-Free Constrained RL with Linear Function Approximation, NeurIPS 2022.

---

> > ### Comment · Reviewer_Nuny · 2025-11-24
> >
> > I thank the Authors for their responses. I will keep my positive evaluation of the work.

---

> > > ### Author Response · Authors · 2025-11-25
> > >
> > > Thank you for your positive evaluation of our work. We sincerely appreciate your careful feedback and the effort you put into reviewing our submission.

---

### Official Review · Reviewer_6Wqt · 2025-10-29

**Soundness:** 3
**Presentation:** 3
**Contribution:** 3
**Rating:** 6
**Confidence:** 2

**Summary:**

This paper presents a novel primal-dual policy optimization algorithm for learning in finite-horizon adversarial CMDPs with linear function approximation, where the reward (loss) functions may vary adversarially and the cost functions are stochastic but constrained. The proposed method is model-free and policy-based, operating directly in the parameter space of policies (e.g., softmax). The authors provide rigorous theoretical guarantees: the algorithm achieves $\widetilde{O}(K^{3/4})$ regret and $\widetilde{O}(K^{3/4})$ constraint violation against the best fixed feasible policy in hindsight. This work claims to be the first to provide sublinear regret and hard violation guarantees for adversarial CMDPs with generalization via linear features and policy-based learning.

**Strengths:**

Novel setting: The paper addresses a previously open setting—adversarial losses with linear CMDPs using policy optimization—for which no prior sublinear regret and violation guarantees existed.

Theoretical rigor: The analysis is sound and carefully done, with clear derivation of the regret and violation bounds under realistic assumptions (e.g., smoothness, realizability).

Policy-based algorithm: Unlike prior works that rely on occupancy measure optimization or LPs, this work develops a computationally efficient policy-based method applicable to large-scale problems.

No Slater assumption: The algorithm does not require prior knowledge of a strictly feasible policy, increasing practical applicability.

Hard constraint violation: The results hold for non-compensated violation metrics, making the guarantee stronger than many earlier results.

**Weaknesses:**

W1: Partial adversariality: While the rewards are adversarial, the costs are assumed to be i.i.d. stochastic. The title and claims could more explicitly reflect this partial adversariality to avoid potential misinterpretation.

W2: Gap to optimal rate: The $\widetilde{O}(K^{3/4})$ regret and violation are not optimal; prior works in simpler settings (e.g., tabular CMDPs) achieved $\widetilde{O}(\sqrt{K})$. A discussion of whether this gap is fundamental in the linear function setting is missing.

W3: Limited empirical evaluation (if any): The submission does not include experiments, which makes it difficult to assess the practical effectiveness and convergence behavior of the algorithm, though the paper focuses on theory.

W4: Suggestion for the preliminary: the structure that overviews the strength in the earlier part is quite understandable, however, there appears some terms without mathmatical definition such as dual variables and primal-dual algorithm. It would be better to explain them in the preliminary to enhance the readability of the non-experts.

**Questions:**

Q1: Could the proposed approach be extended to fully adversarial cost functions (not just stochastic)? What obstacles arise in doing so?

Q2: Is the $\widetilde{O}(K^{3/4})$ bound tight under your assumptions, or do you believe a $\widetilde{O}(\sqrt{K})$ rate is possible with more sophisticated analysis or algorithm design?

Q3: Would the method generalize to nonlinear function approximation (e.g., neural network policies) under similar assumptions?

Q4: Have you considered empirical validation to support the theoretical findings? other than synthetic numerical task?

---

> ### Author Response · Authors · 2025-11-20
>
> We thank the reviewer for the insightful comments and address the questions as follows.
>
> > W1: Partial adversariality: While the rewards are adversarial, the costs are assumed to be i.i.d. stochastic. The title and claims could more explicitly reflect this partial adversariality to avoid potential misinterpretation.
>
> - We fully agree with the reviewer’s comment regarding the title. Accordingly, we will update the title to “Primal–Dual Policy Optimization for Linear CMDPs with Adversarial Losses” in the camera-ready version to avoid potential misinterpretation.
>
> > W2: Gap to optimal rate: The $\widetilde{\mathcal{O}}(K^{3/4})$ regret and violation are not optimal; prior works in simpler settings (e.g., tabular CMDPs) achieved $\widetilde{\mathcal{O}}(\sqrt{K})$. A discussion of whether this gap is fundamental in the linear function setting is missing.
>
> - As the reviewer correctly pointed out, there is a gap between the regret lower bound of $\Omega(\sqrt{H^3 d^2 K})$ (e.g., see [Zhou2021] or [He2023]) and our regret upper bound. Our suboptimal regret and violation bounds arise from the mixing technique, which can be interpreted as the cost of handling the constraints in the adversarial setting.
>
> - Additionally, it is worth noting a promising direction to reduce this gap in our setting. The main challenge arises when analyzing the constraint violation without relying on mixing steps. In our current analysis, the mixing step is inevitable to mitigate KL divergence terms in the drift upper bound (Lemma 17); these KL divergence terms arise from mirror-descent updates to control adversarial losses. However, mixing becomes problematic for linear CMDPs because it enlarges the covering number. If one can design an approach that controls the dual variable without mixing, then achieving optimal bounds may become possible.
>
> - In the Additional Discussion section of the appendix, we added the discussion on the regret lower bound for our setting, along with future directions for achieving the optimal regret bound of $\widetilde{\mathcal{O}}(\sqrt{K})$.
>
> > W3: Limited empirical evaluation (if any): The submission does not include experiments, which makes it difficult to assess the practical effectiveness and convergence behavior of the algorithm, though the paper focuses on theory.
>
> - We agree with the reviewer that our synthetic numerical experiment does not demonstrate practical effectiveness. Its primary purpose is to support our theoretical results. Thus, we leave more extensive empirical evaluation in complex environments as future work.
>
> - Meanwhile, we emphasize that our algorithms can scale effectively even when the state space is large. In particular, our regret and constraint violation bounds are independent of $|\mathcal{S}|$. Moreover, our algorithm is model-free because, at each step, it only requires computing the $Q$-function estimate and the policy only for the visited state $s_h^k$---as well as applying policy mixing---as in usual policy optimization algorithms in the linear MDP literature.
>
>
> > W4: Suggestion for the preliminary: the structure that overviews the strength in the earlier part is quite understandable, however, there appears some terms without mathmatical definition such as dual variables and primal-dual algorithm. It would be better to explain them in the preliminary to enhance the readability of the non-experts.
>
> - We appreciate the reviewer's comments and fully agree with the suggestion. We added additional preliminaries to clarify the motivation behind our technical challenges and algorithmic choices. Due to page limitations, these additions are placed in the appendix. The new content includes a high-level intuition of our algorithm along with the following points: (i) why naive approaches (e.g., those for stochastic linear CMDPs) fail in the adversarial setting; (ii) why previous dual-update strategies fail; (iii) why the mixing step is necessary; and  (iv) how the drift analysis operates.
>
> > References
> - [Zhou2021] Nearly minimax optimal reinforcement learning for linear mixture markov decision processes, COLT 2021.
>
> - [He2023] Nearly minimax optimal reinforcement learning for linear markov decision processes, ICML 2023.

---

> ### Author Response · Authors · 2025-11-20
>
> > Q1: Could the proposed approach be extended to fully adversarial cost functions (not just stochastic)? What obstacles arise in doing so?
>
> - To clarify, it is known that no algorithm can achieve sublinear regret when both the loss and the constraint are adversarially chosen in every episode ([Mannor2009]). Therefore, to obtain theoretically meaningful results, it is natural to assume that either the loss is adversarial and the constraint is stochastic, or vice versa. This impossibility result directly implies that our approach cannot be extended to fully adversarial cost functions.
>
> - Nevertheless, we highlight two natural future directions involving other adversarial settings: (i) adversarial losses under bandit feedback  and stochastic constraints (ii) stochastic losses and adversarial constraints under full feedback.
>
> > Q2: Is the $\widetilde{\mathcal{O}}(K^{3/4})$ bound tight under your assumptions, or do you believe a $\widetilde{\mathcal{O}}(\sqrt{K})$ rate is possible with more sophisticated analysis or algorithm design?
>
> - Please refer to our response to W2, where we discuss in detail why the current $\widetilde{\mathcal{O}}(K^{3/4})$ bound arises from the mixing step and why achieving a $\widetilde{\mathcal{O}}(\sqrt{K})$ regret bound might be possible with improved analysis or alternative algorithmic techniques.
>
> > Q3: Would the method generalize to nonlinear function approximation (e.g., neural network policies) under similar assumptions?
>
>
> - In the RL literature, many prior works have studied regret minimization under nonlinear structures. For instance, [Hwang2023, Li2024, Cho2024] investigated RL with multinomial logit  models; [Yang2020] considered RL with Reproducing Kernel Hilbert space; and [Yang2020, Liu2022] studied RL with neural network approximation. While these results focused on the unconstrained setting, we expect that one could possibly generalize such nonlinear frameworks to our constrained setting. We regard this as an open and important direction for future work.
>
>
> > Q4: Have you considered empirical validation to support the theoretical findings? other than synthetic numerical task?
>
> - Please refer to our response to W3, where we discuss the scope of our empirical evaluation, the purpose of the synthetic numerical experiment, and directions for more extensive empirical validation.
>
>
> > References
>
> - [Mannor2009] Online Learning with Sample Path Constraints, JMLR 2009.
>
> - [Hwang2023] Model-based reinforcement learning with multinomial logistic function approximation, AAAI 2023.
>
> - [Li2024] Provably Efficient Reinforcement Learning with Multinomial Logit Function Approximation, NeurIPS 2024.
>
> - [Cho2024] Randomized Exploration for Reinforcement Learning with Multinomial Logistic Function Approximation, NeurIPS 2024.
>
> - [Yang2020] On Function Approximation in Reinforcement Learning: Optimism in the Face of Large State Spaces, NeurIPS 2020.
>
> - [Liu2022] Understanding Deep Neural Function Approximation in Reinforcement Learning via $\epsilon$-Greedy Exploration, NeurIPS 2022.

---

> ### Comment · Reviewer_6Wqt · 2025-11-24
>
> Thank you very much for your detailed reply.
> Those solved all the concerns I had.
> Especially, I appreciate your explanation on
> - why mixing strategy leads the current bound.
> - impossibility of achieving sublinear regret under adversary in both loss.
>
> and agreement on
> - fixing title to clarify the position.
> - adding preliminary to enhance the readability.
>
> Having those I will increase the confidence to 3 and maintain the score.

---

> > ### Author Response · Authors · 2025-11-25
> >
> > Thank you for your increased confidence and positive evaluation. We sincerely appreciate your careful feedback and the effort you put into reviewing our submission.

---

### Official Review · Reviewer_j4Sg · 2025-11-01

**Soundness:** 3
**Presentation:** 2
**Contribution:** 2
**Rating:** 6
**Confidence:** 2

**Summary:**

This paper studies adversarial linear CMDPs. The authors propose a primal-dual policy optimization algorithm that achieves $O(K^{3/4})$ regret bounds for both the objective and constraint violations. (The constraint-violation bound allows error cancellation, though.)

Section 3 presents three technical challenges and their solutions, primarily focused on controlling the dual variable.

Note: I am not an expert in adversarial linear CMDPs, although I am familiar with stochastic linear CMDPs. My comments below may therefore contain misunderstandings.

**Strengths:**

The paper obtains the first sublinear regret bounds for adversarial linear CMDPs. The authors provide a rigorous analysis of their algorithm; while I did not check every proof detail, the high-level arguments appear reasonable.

**Weaknesses:**

The writing has some room to be improved. Particularly, the motivation behind the technical challenges and the algorithm is not sufficiently clear. It would be nice if the paper briefly describes natural or "naive" approaches for linear CMDPs, and then explain why those approaches fail in the adversarial setting.

For example,
- It is nice to state what existing algorithms (e.g., stochastic linear CMDPs) can achieve and why they are inadequate here. Then, it becomes easier for readers to appreciate the contributions. Section 3 currently assumes familiarity with prior work and jumps into technical fixes.
- Novelty 1 introduces a policy-mixing technique to control the dual variable. It would be nice to explain in Section 3 why mixing is necessary instead of simply using the upper bound implied by a Slater condition (e.g., Ghosh 2022). This point becomes clearer later (around page 7), but until then it was unclear.
- It would be nice to briefly explain what "drift analysis" mentioned in Novelty 3 is. The authors refer to it as a "well-known method for bounding dual variables," but readers would appreciate if it is explained how the method can be applied here or why bounding the dual variable is essential for their regret bounds.

It appears that many technical challenges and the main contributions stem from the mixing technique. Because the paper does not provide an early, intuitive explanation of that technique, its significance is hard to assess.

Overall, due to the lack of clear motivation and context, I score "weak reject" for this paper. However, I'm happy to reconsider my score if the authors can address these concerns.

**Questions:**

1. Would you explain why controlling the dual variable is necessary for regret analysis in this setting?
2. Would you describe how naive approaches fail to control the dual variable in adversarial CMDPs?
3. Would you provide an intuitive explanation of why the mixing technique addresses these failures?

Would you provide a high-level motivation for the algorithm? For example, how it differs from standard primal-dual algorithms for linear CMDPs and which components are critical for handling adversarial environments?

---

> ### Author Response · Authors · 2025-11-20
>
> We thank the reviewer for the insightful comments and address the questions as follows.
>
> > W1: The writing has some room to be improved. Particularly, the motivation behind the technical challenges and the algorithm is not sufficiently clear. It would be nice if the paper briefly describes natural or "naive" approaches for linear CMDPs, and then explain why those approaches fail in the adversarial setting.
>
>
> - We appreciate the reviewer's comments and fully agree with the suggestion. We added additional preliminaries to clarify the motivation behind our technical challenges and algorithmic choices. Due to page limitations, these additions are placed in the appendix. The new content includes a high-level intuition of our algorithm along with the following points: (i) why naive approaches (e.g., those for stochastic linear CMDPs) fail in the adversarial setting; (ii) why previous dual-update strategies fail; (iii) why the mixing step is necessary; and  (iv) how the drift analysis operates.
>
> > Q1: Would you explain why controlling the dual variable is necessary for regret analysis in this setting?
>
> - At a high level, controlling the dual variable is necessary because it captures the balance between regret and constraint violation. In particular, the dual variable plays the role of a weight that trades off the loss and the constraint, e.g., when the constraint is violated, the dual variable increases, assigning more emphasis to the constraint. Hence, if the dual variable blows up, then we may think that the constraint is being violated. In this case, the algorithm fails to maintain the desired balance between regret and violation, meaning that the learning process is unsuccessful.
>
> - More technically, controlling the dual variable is crucial because the optimistically estimated violation (i.e., $\widehat V_{g,1}^k(s_1) - b$) accumulates in the dual variable. This is the result of our dual update in the form of $Y_{k+1} \leftarrow [Y_k + \eta(\widehat V_{g,1}^k (s_1) - b  + \ldots)]_+$. In this sense, the magnitude of the dual variable reflects the amount of accumulated constraint violation.
>
> > Q2: Would you describe how naive approaches fail to control the dual variable in adversarial CMDPs?
>
> - In the stochastic linear CMDP setting, [Ding2021, Ghosh2022] applied a dual clipping technique---cutting off the dual variable when it exceeds a predetermined threshold---but their analysis relies on the assumption that both the loss and the constraint remain fixed across episodes. In particular, to leverage the strong duality of CMDPs, they reformulated the weighted sum of regret and violation into a simple Lagrangian form (e.g., Appendix D of [Ghosh2022]).
>
> - However, when losses are adversarially chosen in each episode, reformulating the sum into a simple Lagrangian form is not allowed. Consequently, their analysis is restricted to the stochastic setting, in which the losses and constraints (or their expectations) are fixed. This limitation motivates our new algorithmic components—periodic policy mixing and a regularized dual update—which are specifically designed to control the dual variable in adversarial environments.
>
> > References
>
> - [Ding2021] Provably Efficient Safe Exploration via Primal-Dual Policy Optimization, AISTATS 2021.
>
> - [Ghosh2022] Provably Efficient Model-Free Constrained RL with Linear Function Approximation, NeurIPS 2022

---

> ### Author Response · Authors · 2025-11-20
>
> > Q3: Would you provide an intuitive explanation of why the mixing technique addresses these failures?
>
> - Intuitively, the purpose of the mixing step is to prevent the dual variable from blowing up by pushing the policy away from the simplex boundary [Wei2020]. Specifically, this step mitigates the KL divergence terms that appear in the drift term $\Delta(k):=(Y_{k+1}^2 - Y_k^2)/2$, making the dual variable more stable.
>
> - More technically, let us recall an upper bound on $\Delta(k)$ (Lemma 17). In policy optimization, since a KL divergence term acts as a regularizer, a typical bound on $\Delta(k)$ involves KL divergence terms as follows: for some $c_1,c_2>0$, let $\widehat\pi_h^{k+1}$ be a policy obtained by applying policy optimization based on $\widetilde\pi_h^k$. Then,
> \begin{align}
>     \Delta(k) \leq - c_1 Y_k + c_2 + D(\pi||\widetilde\pi_h^k) - D(\pi||\widehat\pi_h^{k+1}) %\tag{\Cref{lem:lyap 1}}.
> \end{align}
>
> - The key issue is that $D(\pi || \widetilde\pi_h^k)$ can become arbitrarily large when $\widetilde\pi_h^k(a) \approx 0$ for some $a\in\mathcal{A}$, because the KL divergence is unbounded near the simplex boundary. In contrast, when a mixing step is applied, we ensure that $\widetilde\pi_h^k(a) \geq \theta / |\mathcal{A}|$ for all $a$, where $\theta$ denotes the level of mixing. In this case, we can easily show that $D(\pi|\widetilde\pi_h^k|) \leq \log(|\mathcal{A}|/\theta)$ (Lemma 26). Hence, the mixing step guarantees that the KL term remains bounded, which in turn prevents the dual variable from blowing up.
>
> > Q4: Would you provide a high-level motivation for the algorithm? For example, how it differs from standard primal-dual algorithms for linear CMDPs and which components are critical for handling adversarial environments?
>
> - The high-level motivation behind our algorithm is to adapt policy optimization ideas to the standard primal–dual framework for linear CMDPs in order to handle adversarial environments. The main algorithmic differences compared with prior approaches (e.g., [Ghosh2022]) are: (i) policy optimization, (ii) periodic policy mixing, and (iii) a regularized dual update. Among these, (ii) and (iii) are the most critical, as they address the key challenges that arise in adversarial settings, whereas (i) is a natural modification when losses are adversarially chosen ([Cai2020]).
>
>
> > References
>
> - [Ding2021] Provably Efficient Safe Exploration via Primal-Dual Policy Optimization, AISTATS 2021.
>
> - [Ghosh2022] Provably Efficient Model-Free Constrained RL with Linear Function Approximation, NeurIPS 2022
>
> - [Wei2020] Online Primal-Dual Mirror Descent under Stochastic Constraints, Proceedings of the ACM on Measurement and Analysis of Computing Systems 2020.
>
> - [Cai2020] Provably Efficient Exploration in Policy Optimization, ICML 2020.

---

### Official Review · Reviewer_Zk7f · 2025-11-01

**Soundness:** 3
**Presentation:** 3
**Contribution:** 3
**Rating:** 6
**Confidence:** 3

**Summary:**

The paper studies online finite-horizon adversarial linear CMDPs where per-episode losses are chosen adversarially with full-information feedback, costs are i.i.d. stochastic with bandit feedback, and the dynamics, loss, and cost all admit a linear feature structure as in linear MDPs. The authors propose a policy-based primal–dual algorithm that, for the first time in this setting, attains simultaneous sublinear regret and sublinear constraint violation, both of order $O(K^{\frac{3}{4}})$ where $K$ is the number of episodes. To get there, they introduce a nonstandard weighted LogSumExp softmax policies, which arises because they must occasionally mix the current policy with a uniform policy to keep the dual variable under control, and that mixing destroys the simple softmax recursion. The main technical work is (1) proving a new covering-number bound for this mixed policy class, and (2) combining periodic policy mixing with a regularized dual update so that the policy class does not blow up too fast, and the dual variable stays $O(1)$ rates, which together leads to $O(K^{\frac{3}{4}})$ regret. Experiments on a job-scheduling CMDP with adversarial losses confirm sublinear growth of both metrics.

**Strengths:**

This paper gives the first sublinear result in this exact setting. Prior CMDP work with linear function approximation handled stochastic losses/costs, prior adversarial CMDP work handled tabular state spaces. This paper is the first to get both linear function approximation and adversarial losses and constraints with provable sublinear regret and violation.

Technically, the paper propose analysis to the covering number of weighted logSumExp softmax policy class and periodic mixing to control the dual. Through this way, it is possible to balance statistical complexity and constraint control.

A lot of adversarial/safe CMDP papers fall back to occupancy-measure mirror descent but this paper is policy-based.

**Weaknesses:**

The paper consider full-information losses, leaving more challenging bandit feedback setting unsolved.

The regret bound is still suboptimal due to the mixing technique but this is the cost for the additional constraint.

**Questions:**

Can you show in the synthetic CMDP that (1) removing periodic mixing makes the dual blow up, or (2) forcing mixing every episode makes the regret curve worse? Right now the experiment doesn’t validate the two key algorithmic ideas.

---

> ### Author Response · Authors · 2025-11-20
>
> We thank the reviewer for the insightful comments and address the questions as follows.
>
> > W1: The paper consider full-information losses, leaving more challenging bandit feedback setting unsolved.
>
> - As the reviewer correctly noted, our paper studies adversarial losses under full feedback and stochastic constraints under bandit feedback, and we leave more challenging problem settings open for future work, e.g., adversarial losses under bandit feedback.
>
> - To elaborate on this open problem, extending our setting to the bandit feedback case naturally raises the following question: can we extend the existing frameworks for learning adversarial linear MDPs to the constrained setting? Currently, the state-of-the-art results for adversarial linear MDPs under bandit feedback are due to [Liu2024]. Their first algorithm attains a $\widetilde{\mathcal{O}}(\sqrt{K})$ regret but is computationally inefficient, whereas their second algorithm is computationally efficient but achieves only a $\widetilde{\mathcal{O}}(K^{3/4})$ regret bound.
>
> - However, the question seems highly non-trivial. Even in our full feedback setting, the feature contraction technique developed for adversarial linear MDPs [Cassel2024] is not sufficient for handling constraints, which motivates our new analysis and algorithmic components. Similarly, designing an efficient algorithm for adversarial linear CMDPs under bandit feedback appears fundamentally challenging, as a regret-optimal algorithm is not characterized even for the unconstrained case. For these reasons, we view this direction as an open and promising research problem.
>
> > W2: The regret bound is still suboptimal due to the mixing technique but this is the cost for the additional constraint.
>
> - As the reviewer correctly pointed out, our suboptimal regret and violation bounds arise from the mixing technique, which can be interpreted as the cost of handling the constraints. Nevertheless, we would like to emphasize that our work provides the first provably efficient algorithm for constrained adversarial RL that simultaneously (i) accommodates large state spaces and (ii) operates under adversarial environments.
>
> - Additionally, we outline a promising direction toward achieving $\widetilde{\mathcal{O}}(\sqrt{K})$ regret and violation in our setting. The main challenge arises when analyzing the constraint violation without relying on mixing steps. In our current analysis, the mixing step is inevitable to mitigate KL divergence terms in the drift upper bound (Lemma 17); these KL divergence terms arise from mirror-descent updates to control adversarial losses. However, mixing becomes problematic for linear CMDPs because it enlarges the covering number. If one can design an approach that controls the dual variable without mixing, then achieving optimal bounds may become possible.
>
> - In the Additional Discussion section of the appendix, we added the discussion on the regret lower bound for our setting, along with future directions for achieving the optimal regret bound of $\widetilde{\mathcal{O}}(\sqrt{K})$.
>
> > References
>
> - [Liu2024] Towards Optimal Regret in Adversarial Linear MDPs with Bandit Feedback, ICLR 2024.
>
> - [Cassel2024] Warm-up Free Policy Optimization: Improved Regret in Linear Markov Decision Processes, NeurIPS 2024.

---

> ### Author Response · Authors · 2025-11-20
>
> > Q1: Can you show in the synthetic CMDP that (1) removing periodic mixing makes the dual blow up, or (2) forcing mixing every episode makes the regret curve worse? Right now the experiment doesn’t validate the two key algorithmic ideas.
>
> - As the reviewer correctly pointed out, our original experiment does not directly show the influence of periodic mixing on the regret and the dual variable, as it focuses on demonstrating that the algorithm can learn from adversarial environments. To examine these relationships more clearly, we conducted additional experiments on a simple three-states CMDP, and the results are presented below.
>
>
> > Mixing Period vs. Dual Variable
>
> - To observe how the frequency of mixing affects the dual variable, we compare the upper bound on the Lyapunov drift term (i.e., the right-hand side of Lemma 17). Since the theoretical worst-case bound on the dual variable heavily depends on these drift upper bounds, it is appropriate to compare them rather than the dual variable itself. Moreover, the dominant term in the drift upper bound is $\mathbb{E}_{\mathbb{\bar P}, \bar\pi}[\sum_h D(\bar\pi_h \| \widetilde{\pi}_h^k)]$, which can be blown up without mixing steps.
>
> - The table below compares $\mathbb{E}_{\mathbb{\bar P}, \bar\pi}[\sum_h D(\bar\pi_h \| \widetilde{\pi}_h^k)]$ under different mixing periods for episodes $100,\ldots,1000$. The table shows that the drift upper bound tends to decrease as the mixing period decreases, indicating that mixing steps are indeed effective for preventing the dual variable from blowing up.
>
> | Episode | 100   | 200   | 300   | 400   | 500   | 600   | 700   | 800   | 900   | 1000  |
> |---------------|--------|--------|--------|--------|--------|--------|--------|--------|--------|--------|
> | Mixing Period=$1000$      | 0.20   | 0.22   | 0.37   | 1.90   | 9.92   | 20.78  | 21.75  | 21.32  | 21.44  | 21.37  |
> | Mixing Period=$200$         | 0.20   | 0.28   | 0.61   | 2.24   | 9.37   | 20.15  | 13.58  | 20.11  | 12.63  | 19.67  |
> | Mixing Period=$100$         | 0.20   | 0.16   | 0.91   | 3.95   | 7.02   | 11.09  | 11.90  | 11.57  | 12.79  | 14.60  |
>
>
>
> > Mixing Period vs. Regret
>
> - Next, let us consider the question of how the frequency of mixing affects regret. Before presenting the comparison of the mixing period and regret, we note that the mixing period directly influences the bonus term. This is because, in the linear CMDP setting, an upper bound on the covering number is closely tied to the estimation error of the transition kernel, and this error determines the bonus term. This relationship can be roughly summarized as: $(\text{bonus term}) \propto K / (\text{mixing period})$, where $K$ is the number of episodes.
>
> - The table below shows the regret under different mixing periods for episodes $200,\ldots, 1000$. As noted above, we set the bonus term proportional to $K / (\text{mixing period})$. As shown in the table, the regret increases as the mixing period decreases.
>
> | Episode | 200     | 400     | 600     | 800     | 1000    |
> |----------------|---------|---------|---------|---------|---------|
> | Mixing Period=$1000$            | 266.7162 | 495.2860 | 553.3740 | 598.9484 | 619.3965 |
> | Mixing Period=$200$           | 348.5755 | 649.3384 | 977.2894 | 1153.4152 | 1558.5700 |
> | Mixing Period=$100$            | 345.8861 | 700.0715 | 1007.0875 | 1279.6341 | 1679.9006 |
>
>
> > Conclusion
>
> - These additional experiments demonstrate that as the frequency of mixing increases, i.e., the mixing period shortens, the drift term decreases while the regret increases. Since the constraint violation can be regarded as being proportional to the dual variable (see the proof of Lemma 6), these results exactly support our claim that the mixing period is a key factor to balance regret and constraint violation, highlighting the importance of our algorithmic design.

---

### Meta-Review · Area_Chair_RzmD · 2025-12-24

**Summary:**

This paper studies the linear CMDPs under the online and adversarial setting. Using the log-sum-exp policies, the authors   propose a primal–dual policy optimization algorithm and analyze its regret and constraint satisfaction. Sublinear regret bound has been established for their method.

The reviewers have raised several reasonable concerns. And the AC will summarize the un-resolved ones after the rebuttal and discussion period: (1) The bandit feedback setting is not considered in the adversarial setting; (2) The regret bound remains suboptimal; (3) Lack of numerical validation; (4) Non-optimal dependence on the margin in Slater's condition.

Overall, the authors have properly addressed part of the major issues. Despite some concerns are not fully addressed, the reviewers are already giving overall positive ratings, suggesting that the reviewers appreciate the results even with the current issues. Combined with initial scores from the reviewers, the AC suggest an accept to this paper.

**Reviewer Concerns:**

Addressed:
(1) j4Sg: Writing clarity.
(2) 6Wqt:  Suggestion for the preliminary.


Not Addressed:
(1) Zk7f: The bandit feedback setting is not considered in the adversarial setting.
(2) Zk7f & 6Wqt: The regret bound remains suboptimal.
(3) 6Wqt: Lack of numerical validation.
(4) Nuny: Non-optimal dependence on the margin in Slater's condition.

**Reviewer Scores:**

The reviewers are all giving positive scores. However, part of their concerns are only justified, yet not fixed. This indicate that they may not want to further increase the score. (One reviewer has already increased the score to 6 before the data leaking of ICLR, the AC has the increased score)

---

### Decision · Program_Chairs · 2026-01-26

Accept (Poster)